# THE HEAVY-TAIL PHENOMENON IN SGD

## ABSTRACT

In recent years, various notions of capacity and complexity have been proposed for characterizing the generalization properties of stochastic gradient descent (SGD) in deep learning. Some of the popular notions that correlate well with the performance on unseen data are (i) the 'flatness' of the local minimum found by SGD, which is related to the eigenvalues of the Hessian, (ii) the ratio of the stepsize $\eta$ to the batch size $b$, which essentially controls the magnitude of the stochastic gradient noise, and (iii) the 'tail-index', which measures the heaviness of the tails of the network weights at convergence. In this paper, we argue that these three seemingly unrelated perspectives for generalization are deeply linked to each other. We claim that depending on the structure of the Hessian of the loss at the minimum, and the choices of the algorithm parameters $\eta$ and $b$, the SGD iterates will converge to a *heavy-tailed* stationary distribution. We rigorously prove this claim in the setting of quadratic optimization: we show that even in a simple linear regression problem with independent and identically distributed Gaussian data, the iterates can be heavy-tailed with infinite variance. We further characterize the behavior of the tails with respect to algorithm parameters, the dimension, and the curvature. We then translate our results into insights about the behavior of SGD in deep learning. We finally support our theory with experiments conducted on both synthetic data and fully connected neural networks.

## 1 INTRODUCTION

The learning problem in neural networks can be expressed as an instance of the well-known *population risk minimization* problem in statistics, given as follows:

$$\min_{x \in \mathbb{R}^d} F(x) := \mathbb{E}_{z \sim \mathcal{D}}[f(x, z)], \tag{1.1}$$

where $z \in \mathbb{R}^p$ denotes a random data point, $\mathcal{D}$ is a probability distribution on $\mathbb{R}^p$ that denotes the law of the data points, $x \in \mathbb{R}^d$ denotes the parameters of the neural network to be optimized, and $f : \mathbb{R}^d \times \mathbb{R}^p \mapsto \mathbb{R}_+$ denotes a measurable cost function, which is often non-convex in $x$. While this problem cannot be attacked directly since $\mathcal{D}$ is typically unknown, if we have access to a *training dataset* $S = \{z_1, \ldots, z_n\}$ with $n$ independent and identically distributed (i.i.d.) observations, i.e., $z_i \sim_{\text{i.i.d.}} \mathcal{D}$ for $i = 1, \ldots, n$, we can use the *empirical risk minimization* strategy, which aims at solving the following optimization problem (Shalev-Shwartz & Ben-David, 2014):

$$\min_{x \in \mathbb{R}^d} f(x) := f(x, S) := (1/n) \sum_{i=1}^{n} f^{(i)}(x), \tag{1.2}$$

where $f^{(i)}$ denotes the cost induced by the data point $z_i$. The stochastic gradient descent (SGD) algorithm has been one of the most popular algorithms for addressing this problem:

$$x_k = x_{k-1} - \eta \nabla \tilde{f}_k(x_{k-1}), \quad \text{where} \quad \nabla \tilde{f}_k(x) := (1/b) \sum_{i \in \Omega_k} \nabla f^{(i)}(x). \tag{1.3}$$

Here, $k$ denotes the iterations, $\eta > 0$ is the stepsize (also called the learning-rate), $\nabla \tilde{f}$ is the stochastic gradient, $b$ is the batch-size, and $\Omega_k \subset \{1, \ldots, n\}$ is a random subset with $|\Omega_k| = b$ for all $k$.

Even though the practical success of SGD has been proven in many domains, the theory for its generalization properties is still in an early phase. Among others, one peculiar property of SGD that has not been theoretically well-grounded is that, depending on the choice of $\eta$ and $b$, the algorithm can exhibit significantly different behaviors in terms of the performance on unseen test data.

A common perspective over this phenomenon is based on the 'flat minima' argument that dates back to Hochreiter & Schmidhuber (1997), and associates the performance with the 'sharpness' or 'flatness' of the minimizers found by SGD, where these notions are often characterized by the magnitude of the eigenvalues of the Hessian, larger values corresponding to sharper local minima

(Keskar et al., 2016). Recently, Jastrzębski et al. (2017) focused on this phenomenon as well and empirically illustrated that the performance of SGD on unseen test data is mainly determined by the stepsize $\eta$ and the batch-size $b$, i.e., larger $\eta/b$ yields better generalization. Revisiting the flat-minima argument, they concluded that the ratio $\eta/b$ determines the flatness of the minima found by SGD; hence the difference in generalization. In the same context, Şimşekli et al. (2019b) focused on the statistical properties of the gradient noise $(\nabla \tilde{f}_k(x) - \nabla f(x))$ and illustrated that under an isotropic model, the gradient noise exhibits a heavy-tailed behavior, which was also confirmed in follow-up studies (Zhang et al., 2019). Based on this observation and a metastability argument (Pavlyukevich, 2007), they showed that SGD will 'prefer' wider basins under the heavy-tailed noise assumption, without an explicit mention of the cause of the heavy-tailed behavior.

In another recent study, Martin & Mahoney (2019) introduced a new approach for investigating the generalization properties of deep neural networks by invoking results from heavy-tailed random matrix theory. They empirically showed that the eigenvalues of the weight matrices in different layers exhibit a *heavy-tailed* behavior, which is an indication that the weight matrices themselves exhibit heavy tails as well (Ben Arous & Guionnet, 2008). Accordingly, they fitted a power law distribution to the empirical spectral density of individual layers and illustrated that heavier-tailed weight matrices indicate better generalization. Very recently, Şimşekli et al. (2020) formalized this argument in a mathematically rigorous framework and showed that such a heavy-tailed behavior diminishes the 'effective dimension' of the problem, which in turn results in improved generalization. While these studies form an important initial step towards establishing the connection between heavy tails and generalization, the *originating cause* of the observed heavy-tailed behavior is yet to be understood.

In this paper, we argue that these three seemingly unrelated perspectives for generalization are deeply linked to each other. We claim that, depending on the choice of the algorithm parameters $\eta$ and $b$, the dimension $d$, and the curvature of $f$ (to be precised in Section 3), SGD exhibits a 'heavy-tail phenomenon', meaning that the law of the iterates converges to a heavy-tailed distribution. We rigorously prove that, this phenomenon is not specific to deep learning and in fact it can be observed even in surprisingly simple settings: we show that when $f$ is chosen as a simple quadratic function and the data points are i.i.d. from an isotropic Gaussian distribution, the iterates can still converge to a heavy-tailed distribution with arbitrarily heavy tails, hence with infinite variance. We summarize our contributions as follows:

1. When $f$ is a quadratic, we prove that: (i) the tails become *monotonically heavier* for increasing curvature, increasing $\eta$, or decreasing $b$, hence relating the heavy-tails to the ratio $\eta/b$ and the curvature, (ii) the law of the iterates converges exponentially fast towards the stationary distribution in the Wasserstein metric, (iii) there exists a higher-order moment (e.g., variance) of the iterates that diverges *at most* polynomially-fast, depending on the heaviness of the tails at stationarity.
2. We support our theory with experiments conducted on both synthetic data and neural networks. Our experimental results confirm our theory on synthetic setups and also illustrate that the heavy-tail phenomenon is also observed in fully connected multi-layer neural networks.

To the best of our knowledge, these results are the first of their kind to rigorously characterize the empirically observed heavy-tailed behavior of SGD with respect to $\eta$, $b$, $d$, and the curvature, with explicit convergence rates. [1]

## 2 TECHNICAL BACKGROUND

**Heavy-tailed distributions with a power-law decay.** In probability theory, a real-valued random variable $X$ is said to be *heavy-tailed* if the right tail or the left tail of the distribution decays slower

---

[1] We note that in a concurrent work, which very recently appeared on arXiv, Hodgkinson & Mahoney (2020) showed that heavy tails with power laws arise in more general Lipschitz stochastic optimization algorithms that are contracting on average for strongly convex objectives near infinity with positive probability. Our Theorem 1 and Lemma 14 are more refined as we focus on the special case of SGD with Gaussian data, where we are able to provide constants which *explicitly* determine the tail index as an expectation over data and SGD parameters (see also eqn. (3.6)). Due to the generality of their framework, (Hodgkinson & Mahoney, 2020, Thm 1) is more implicit and it cannot provide such a characterization of these constants, however it can be applied to other algorithms beyond SGD. All our other results (including Theorem 2 – monotonicity of the tail-index and Corollary 9 – central limit theorem for the ergodic averages) are all specific to SGD and cannot be obtained under the framework of Hodgkinson & Mahoney (2020). We encourage the readers to refer to (Hodgkinson & Mahoney, 2020) for the treatment of more general stochastic recursions.

than any exponential distribution. We say $X$ has heavy (right) tail if $\lim_{x \to \infty} \mathbb{P}(X \geq x)e^{cx} = \infty$ for any $c > 0$. [2] Similarly, an $\mathbb{R}^d$-valued random vector $X$ has heavy tail if $u^T X$ has heavy right tail for some vector $u \in \mathbb{S}^{d-1}$, where $\mathbb{S}^{d-1} := \{u \in \mathbb{R}^d : \|u\| = 1\}$ is the unit sphere in $\mathbb{R}^d$.

Heavy tail distributions include $\alpha$-stable distributions, Pareto distribution, log-normal distribution and the Weilbull distribution. One important class of the heavy-tailed distributions is the distributions with *power-law* decay, which is the focus of our paper. That is, $\mathbb{P}(X \geq x) \sim c_0 x^{-\alpha}$ as $x \to \infty$ for some $c_0 > 0$ and $\alpha > 0$, where $\alpha > 0$ is known as the *tail-index*, which determines the tail thickness of the distribution. Similarly, we say that the random vector $X$ has power-law decay with tail-index $\alpha$ if for some $u \in \mathbb{S}^{d-1}$, we have $\mathbb{P}(u^T X \geq x) \sim c_0 x^{-\alpha}$, for some $c_0, \alpha > 0$.

**Stable distributions.** The class of $\alpha$-stable distributions are an important subclass of heavy-tailed distributions with a power-law decay, which appears as the limiting distribution of the generalized CLT for a sum of i.i.d. random variables with infinite variance (Lévy, 1937). A random variable $X$ follows a symmetric $\alpha$-stable distribution denoted as $X \sim \mathcal{S}\alpha\mathcal{S}(\sigma)$ if its characteristic function takes the form: $\mathbb{E}\left[e^{itX}\right] = \exp\left(-\sigma^\alpha |t|^\alpha\right)$, $t \in \mathbb{R}$, where $\sigma > 0$ is the scale parameter that measures the spread of $X$ around 0, and $\alpha \in (0, 2]$ is known as the tail-index, and $\mathcal{S}\alpha\mathcal{S}$ becomes heavier-tailed as $\alpha$ gets smaller. The probability density function of a symmetric $\alpha$-stable distribution, $\alpha \in (0, 2]$, does not yield closed-form expression in general except for a few special cases. When $\alpha = 1$ and $\alpha = 2$, $\mathcal{S}\alpha\mathcal{S}$ reduces to the Cauchy and the Gaussian distributions, respectively. When $0 < \alpha < 2$, $\alpha$-stable distributions have their moments being finite only up to the order $\alpha$ in the sense that $\mathbb{E}[|X|^p] < \infty$ if and only if $p < \alpha$, which implies infinite variance.

**Wasserstein metric.** For any $p \geq 1$, define $\mathcal{P}_p(\mathbb{R}^d)$ as the space consisting of all the Borel probability measures $\nu$ on $\mathbb{R}^d$ with the finite $p$-th moment (based on the Euclidean norm). For any two Borel probability measures $\nu_1, \nu_2 \in \mathcal{P}_p(\mathbb{R}^d)$, we define the standard $p$-Wasserstein metric (Villani, 2009): $\mathcal{W}_p(\nu_1, \nu_2) := (\inf \mathbb{E}[\|Z_1 - Z_2\|^p])^{1/p}$, where the infimum is taken over all joint distributions of the random variables $Z_1, Z_2$ with marginal distributions $\nu_1, \nu_2$ respectively.

## 3 SETUP AND MAIN THEORETICAL RESULTS

Before stating our theoretical results in detail, let us informally motivate our main method of analysis. Suppose that the initial SGD iterate $x_0$ is in the domain of attraction[3] of a local minimum $x_\star$ of $f$ and the function $f$ is smooth and well-approximated by a quadratic function in this basin. Under this assumption, by considering a first-order Taylor approximation of $\nabla f^{(i)}(x)$ around $x_\star$, we have $\nabla f^{(i)}(x) \approx \nabla f^{(i)}(x_\star) + \nabla^2 f^{(i)}(x_\star)(x - x_\star)$. By using this approximation, we can approximate the SGD recursion (1.3) as follows:

$$x_k \approx x_{k-1} - (\eta/b) \sum_{i \in \Omega_k} \nabla^2 f^{(i)}(x_\star) x_{k-1} + (\eta/b) \sum_{i \in \Omega_k} \left( \nabla^2 f^{(i)}(x_\star) x_\star - \nabla f^{(i)}(x_\star) \right)$$
$$=: (I - (\eta/b)H_k) x_{k-1} + q_k, \tag{3.1}$$

where $I$ denotes the identity matrix of appropriate size. Here, our main observation is that the SGD recursion can be approximated by a *linear stochastic recursion*, which gives us access to the tools from implicit renewal theory for investigating its statistical properties (Kesten, 1973; Goldie, 1991). In a renewal theoretic context, the object of interest would be the matrix $(I - \frac{\eta}{b}H_k)$, whose statistical properties determine the behavior of $x_k$: depending on the moments of this matrix, $x_k$ can have heavy or light tails, or might even diverge.

In this study, we focus on the tail behavior of the SGD dynamics by analyzing it through the lens of implicit renewal theory. As, the recursion (3.1) is obtained by a quadratic approximation of the component functions $f^{(i)}$, which arises naturally in linear regression, we will consider a simplified setting and rigorously study this dynamics in the case of linear regression.

We would like to underline that, in our analysis, the Taylor approximation (3.1) is not crucial. Indeed, we can easily extend our theory to more general *non-linear* recursions by imposing strict statistical assumptions on the loss function (which can be chosen non-convex) and the data distribution (Mirek, 2011). Unfortunately, such assumptions would either be trivially false for deep learning problems,

---

[2] A real-valued random variable $X$ has heavy (left) tail if $\lim_{x \to \infty} \mathbb{P}(X \leq -x)e^{c|x|} = \infty$ for any $c > 0$.

[3] We say $x_0$ is in the domain of attraction of a local minimum $x_\star$, if gradient descent iterations to minimize $f$ started at $x_0$ with sufficiently small stepsize converge to $x_\star$ as the number of iterations goes to infinity.

e.g., (Mirek, 2011) or cannot be verified in practice, e.g., (Hodgkinson & Mahoney, 2020). In order to be able to provide explicit results with clear assumptions, we limit our scope to quadratic optimization, which turns out to be already fairly technical. We leave the analysis of the general case as a natural next step of our work.

We now consider the case when $f$ is a quadratic, which arises in linear regression:

$$\min_{x \in \mathbb{R}^d} F(x) := (1/2)\mathbb{E}_{(a,y) \sim \mathcal{D}} \left[ \left( a^T x - y \right)^2 \right] , \qquad (3.2)$$

where the data $(a, y)$ comes from an unknown distribution $\mathcal{D}$ with support $\mathbb{R}^d \times \mathbb{R}$. Assume we have access to i.i.d. samples $(a_i, y_i)$ from the distribution $\mathcal{D}$ where $\nabla f^{(i)}(x) = a_i(a_i^T x - y_i)$ is an unbiased estimator of the true gradient $\nabla F(x)$. The curvature, i.e. the value of second partial derivatives, of this objective around a minimum is determined by the Hessian matrix $\mathbb{E}(aa^T)$ which depends on the distribution of $a$. In this setting, SGD with batch size $b$ leads to the iterations

$$x_k = M_k x_{k-1} + q_k \text{ with } M_k := I - (\eta/b)H_k, \ H_k := \sum\nolimits_{i \in \Omega_k} a_i a_i^T, \ q_k := (\eta/b) \sum\nolimits_{i \in \Omega_k} a_i y_i , \qquad (3.3)$$

where $\Omega_k := \{b(k-1) + 1, b(k-1) + 2, \ldots, bk\}$ with $|\Omega_k| = b$. Here, for simplicity, we assume that we are in the one-pass regime (also called the streaming setting (Frostig et al., 2015; Jain et al., 2017)) where each sample is used only once without being recycled. Our purpose in this paper is to show that heavy tails can arise in SGD even in simple settings such as when the input data $a_i$ is Gaussian, *without the necessity to have a heavy-tailed input data*[4]. Consequently, we make the following assumptions on the data throughout the paper:

(A1) $a_i \sim \mathcal{N}(0, \sigma^2 I_d)$ are i.i.d.

(A2) $y_i$ are i.i.d. with a continuous density whose support is $\mathbb{R}$ with all the moments finite.

Assumption (A2) would be satisfied in many cases, for instance when $y_i$ is normally distributed on $\mathbb{R}$. Note that by Assumption (A1), the matrices $M_k = I - \frac{\eta}{b} H_k$ are i.i.d. and the Hessian matrix of the objective (3.2) satisfies $\mathbb{E}(aa^T) = \sigma^2 I_d$ where the value of $\sigma^2$ determines the *curvature* around a minimum; smaller (larger) $\sigma^2$ implies the objective will grow slower (faster) around the minimum and the minimum will be flatter (sharper) (see e.g. Dinh et al. (2017)). We introduce

$$h(s) := \lim_{k \to \infty} \left( \mathbb{E} \| M_k M_{k-1} \ldots M_1 \|^s \right)^{1/k} , \qquad (3.4)$$

which arises in stochastic matrix recursions (see e.g. Buraczewski et al. (2014)) where $\| \cdot \|$ denotes the matrix 2-norm (i.e. largest singular value of a matrix). Since $\mathbb{E}\|M_k\|^s < \infty$ for all $k$ and $s > 0$, we have $h(s) < \infty$. Let us also define

$$\Pi_k := M_k M_{k-1} \ldots M_1 \quad \text{and} \quad \rho := \lim_{k \to \infty} (2k)^{-1} \log \left( \text{largest eigenvalue of } \Pi_k^T \Pi_k \right) . \quad (3.5)$$

The latter quantity is called the top Lyapunov exponent of the stochastic recursion (3.3). Furthermore, if $\rho$ exists and is negative, it can be shown that a stationary distribution of the recursion (3.3) exists. In the Appendix (see Lemma 14), we show that under our assumptions,

$$\rho = \mathbb{E} \log \| (I - (\eta/b)H) e_1 \| , \quad h(s) = \mathbb{E} \left[ \| (I - (\eta/b)H) e_1 \|^s \right] \text{ for } \rho < 0, \qquad (3.6)$$

where $H$ is a matrix with the same distribution as $H_k$, and $e_1$ is the first basis vector.

In the following, we show that the limit density has a polynomial tail with a tail-index given precisely by $\alpha$, the unique critical value such that $h(\alpha) = 1$. The result builds on adapting the techniques developed in stochastic matrix recursions (Alsmeyer & Mentemeier, 2012; Buraczewski et al., 2016) to our setting. Our result shows that even in the simplest setting when the input data is Gaussian without any heavy tail, SGD iterates can lead to a heavy-tailed stationary distribution with an infinite variance. To our knowledge, this is the first time such a phenomenon is proven in the linear regression setting.

**Theorem 1.** *Consider the SGD iterations (3.3). If $\rho < 0$, then SGD iterations admit a unique stationary distribution $x_\infty$ which satisfy*

$$\lim_{t \to \infty} t^\alpha \mathbb{P} \left( u^T x_\infty > t \right) = e_\alpha(u), \quad u \in \mathbb{S}^{d-1} , \qquad (3.7)$$

*for some positive and continuous function $e_\alpha$ on the unit sphere $\mathbb{S}^{d-1}$, where $\alpha$ is the unique positive value such that $h(\alpha) = 1$.*

---

[4]Note that if the input data is heavy-tailed, the stationary distribution of SGD automatically becomes heavy-tailed, see Buraczewski et al. (2012) for details. In our context, the challenge is to identify the occurrence of the heavy tails when the distribution of the input data is light-tailed, such as a simple Gaussian distribution.

As Martin & Mahoney (2019); Şimşekli et al. (2020) provide both numerical and theoretical evidence showing that the tail-index of the density of the network weights is closely related to the generalization performance, where smaller tail-index indicates better generalization, a natural question of practical importance is *how the tail-index depends on the parameters of the problem including the batch size, dimension and the stepsize*. We prove that larger batch sizes lead to a lighter tail (i.e. larger $\alpha$), which links the heavy tails to the observation that smaller $b$ yields improved generalization in a variety of settings in deep learning (Keskar et al., 2016; Panigrahi et al., 2019; Martin & Mahoney, 2019). We also prove that smaller stepsizes lead to larger $\alpha$, hence lighter tails, which agrees with the fact that the existing literature for linear regression often choose $\eta$ small enough to guarantee that variance of the iterates stay bounded (Dieuleveut et al., 2017b; Jain et al., 2017).

**Theorem 2.** *The tail-index $\alpha$ is strictly increasing in batch size $b$ and strictly decreasing in stepsize $\eta$ and variance $\sigma^2$ provided that $\alpha \geq 1$. Moreover, the tail-index $\alpha$ is strictly decreasing in dimension $d$.*

Next result characterizes the tail-index $\alpha$ depending on the choice of the batch size $b$, the variance $\sigma^2$, which determines the curvature around the minimum and the stepsize; in particular we show that if the stepsize exceeds an explicit threshold, the stationary distribution will become heavy tailed with an infinite variance.

**Proposition 3.** *Let $\eta_{crit} = \frac{2b}{\sigma^2(d+b+1)}$. The following holds: (i) There exists $\eta_{max} > \eta_{crit}$ such that for any $\eta_{crit} < \eta < \eta_{max}$, Theorem 1 holds with tail index $0 < \alpha < 2$. (ii) If $\eta = \eta_{crit}$, Theorem 1 holds with tail index $\alpha = 2$. (iii) If $\eta \in (0, \eta_{crit})$, then Theorem 1 holds with tail index $\alpha > 2$.*

**Relation to first exit times.** Proposition 3 implies that, for fixed $\eta$ and $b$, the tail-index $\alpha$ will be decreasing with increasing $\sigma$. Combined with the first-exit-time analyses of Şimşekli et al. (2019b); Nguyen et al. (2019), which state that the escape probability from a basin becomes higher for smaller $\alpha$, our result implies that the probability of SGD escaping from a basin gets larger with increasing curvature; hence providing an alternative view for the argument that SGD prefers flat minima.

**Three regimes for stepsize.** Theorems 1-2 and Proposition 3 identify three regimes: (I) convergence to a limit with a finite variance if $\rho < 0$ and $\alpha > 2$; (II) convergence to a heavy-tailed limit with infinite variance if $\rho < 0$ and $\alpha < 2$; (III) $\rho > 0$ when convergence cannot be guaranteed. For Gaussian input, if the stepsize is small enough, smaller than $\eta_{crit}$, by Proposition 3, $\rho < 0$ and $\alpha > 2$, therefore regime (I) applies. As we increase the stepsize, there is a critical stepsize level $\eta_{crit}$ for which $\eta > \eta_{crit}$ leads to $\alpha < 2$ as long as $\eta < \eta_{max}$ where $\eta_{max}$ is the maximum allowed stepsize for ensuring convergence (corresponds to $\rho = 0$). A similar behavior with three (learning rate) stepsize regimes was reported in Lewkowycz et al. (2020) and derived analytically for one hidden layer linear networks with a large width. The large stepsize choices that avoids divergence, so called the *catapult phase* for the stepsize, yielded the best generalization performance empirically, driving the iterates to a flatter minima in practice. We suspect that the catapult phase in Lewkowycz et al. (2020) corresponds to regime (II) in our case, where the iterates are heavy-tailed, which might cause convergence to flatter minima as the first-exit-time discussions suggest (Şimşekli et al., 2019a).

**Moment Bounds and Convergence Speed.** Theorem 1 is of asymptotic nature which characterizes the stationary distribution $x_\infty$ of SGD iterations with a tail-index $\alpha$. Next, we provide non-asymptotic moment bounds for $x_k$ at each $k$-th iterate, and also for the limit $x_\infty$.

**Theorem 4.** *(i) If the tail-index $\alpha \leq 1$, then for any $p \in (0, \alpha)$, we have $h(p) < 1$ and*

$$\mathbb{E}\|x_k\|^p \leq (h(p))^k \mathbb{E}\|x_0\|^p + \frac{1 - (h(p))^k}{1 - h(p)} \mathbb{E}\|q_1\|^p. \tag{3.8}$$

*(ii) If the tail-index $\alpha > 1$, then for any $p \in (1, \alpha)$, we have $h(p) < 1$ and for any $\epsilon > 0$ such that $(1 + \epsilon)h(p) < 1$, we have*

$$\mathbb{E}\|x_k\|^p \leq ((1 + \epsilon)h(p))^k \mathbb{E}\|x_0\|^p + \frac{1 - ((1 + \epsilon)h(p))^k}{1 - (1 + \epsilon)h(p)} \frac{(1 + \epsilon)^{\frac{p}{p-1}} - (1 + \epsilon)}{((1 + \epsilon)^{\frac{1}{p-1}} - 1)^p} \mathbb{E}\|q_1\|^p. \tag{3.9}$$

Theorem 4 shows that when $p < \alpha$ the upper bound on the $p$-th moment of the iterates converges exponentially to the $p$-the moment of $q_1$ when $\alpha \leq 1$ and a neighborhood of the $p$-moment of $q_1$ when $\alpha > 1$, where $q_1$ is defined in (3.3). By letting $k \to \infty$ and applying Fatou's lemma, we can also characterize the moments of the stationary distribution.

**Corollary 5.** *(i) If the tail-index $\alpha \leq 1$, then for any $p \in (0, \alpha)$, $\mathbb{E}\|x_\infty\|^p \leq \frac{1}{1-h(p)}\mathbb{E}\|q_1\|^p$, where $h(p) < 1$. (ii) If the tail-index $\alpha > 1$, then for any $p \in (1, \alpha)$, we have $h(p) < 1$ and for any $\epsilon > 0$ such that $(1+\epsilon)h(p) < 1$, we have $\mathbb{E}\|x_\infty\|^p \leq \frac{1}{1-(1+\epsilon)h(p)} \frac{(1+\epsilon)^{\frac{p}{p-1}}-(1+\epsilon)}{((1+\epsilon)^{\frac{1}{p-1}}-1)^p}\mathbb{E}\|q_1\|^p$.*

Next, we will study the speed of convergence of the $k$-th iterate $x_k$ to its stationary distribution $x_\infty$ in the Wasserstein metric $\mathcal{W}_p$ for any $1 \leq p < \alpha$.

**Theorem 6.** *Assume $\alpha > 1$. Let $\nu_k$, $\nu_\infty$ denote the probability laws of $x_k$ and $x_\infty$ respectively. Then $\mathcal{W}_p(\nu_k, \nu_\infty) \leq (h(p))^{k/p}\mathcal{W}_p(\nu_0, \nu_\infty)$, for any $1 \leq p < \alpha$, where the convergence rate $(h(p))^{1/p} \in (0, 1)$.*

Theorem 6 shows that in case $\alpha < 2$ the convergence to a heavy tailed distribution occurs relatively fast, i.e. with a linear convergence in the $p$-Wasserstein metric. We can also characterize the constant $h(p)$ in Theorem 6 which controls the convergence rate as follows:

**Corollary 7.** *When $\eta < \eta_{crit} = \frac{2b}{\sigma^2(d+b+1)}$, the tail-index $\alpha > 2$,*

$$\mathcal{W}_2(\nu_k, \nu_\infty) \leq \left(1 - 2\eta\sigma^2\left(1 - \eta/\eta_{crit}\right)\right)^{k/2}\mathcal{W}_2(\nu_0, \nu_\infty). \tag{3.10}$$

Theorem 6 works for any $p < \alpha$. At the critical $p = \alpha$, Theorem 1 indicates that $\mathbb{E}\|x_\infty\|^\alpha = \infty$, and therefore has $\mathbb{E}\|x_k\|^\alpha \to \infty$ as $k \to \infty$, [5] which serves as an evidence that the tail gets heavier as the number of iterates $k$ increases. By adapting the proof of Theorem 4, we have the following result stating that the moments of the iterates of order $\alpha$ go to infinity but this speed can only be polynomially fast.

**Proposition 8.** *Given the tail-index $\alpha$, we have $\mathbb{E}\|x_\infty\|^\alpha = \infty$. Moreover, $\mathbb{E}\|x_k\|^\alpha = O(k)$ if $\alpha \leq 1$, and $\mathbb{E}\|x_k\|^\alpha = O(k^\alpha)$ if $\alpha > 1$.*

It may be possible to leverage recent results on the concentration of products of i.i.d. random matrices (Huang et al., 2020; Henriksen & Ward, 2020) to study the tail of $x_k$ for finite $k$, which can be a future research direction.

**Extension to non-Gaussian data.** Our main purpose in Theorem 1 is to show that heavy tails can arise even in the simplest setting when the input is Gaussian. However, Proposition 1 extends naturally if the input $a_i$ is not necessarily Gaussian. For example, if we assume that the distribution of $a_i$ has support of $\mathbb{R}^d$ and has a finite second moment, it can be checked that our proof technique for Theorem 1 will be still applicable and Theorem 1 will hold with $h(s)$ defined by (3.4). The only difference is that when input is not Gaussian, the explicit formula (3.6) for $h(s)$ will not hold as an equality but it will become an inequality, i.e.

$$h(s) \leq \tilde{h}(s) := \mathbb{E}\left[\|(I - (\eta/b)H)\,e_1\|^s\right], \tag{3.11}$$

where $h(s)$ is defined by (3.4). This inequality is just a consequence of sub-multiplicativity of matrix products appearing in (3.4). If $\tilde{\alpha}$ is such that $\tilde{h}(\alpha) = 1$, then by (3.11), $\tilde{\alpha}$ is a lower bound on the tail index $\alpha$ that satisfies $h(\alpha) = 1$ where $h$ is defined as in (3.4). In other words, when the input is not Gaussian, we have $\tilde{\alpha} \leq \alpha$ and therefore $\tilde{\alpha}$ serves as a lower bound on the tail index. Furthermore, Theorem 2 will also apply in the sense that $\tilde{\alpha}$. $\tilde{\alpha}$ will be strictly increasing in batch size $b$ and strictly increasing in stepsize $\eta$ and variance $\sigma^2$ provided that $\tilde{\alpha} \geq 1$.

**Extension to non-quadratic optimization.** We note that extending our results beyond quadratic optimization is possible, if the gradients have asymptotic linear growth. For example, consider the cost $F(x) = \mathbb{E}[\ell(a^Tx - y)]$ with loss function $\ell$. The choice of $\ell(z) = \|z\|^2/2$ is the standard linear regression setting where the gradient of $F$ is an affine function of $x$ and in this case $\|\nabla F(x) - \Sigma x\|$ is bounded if we choose $\Sigma = \mathbb{E}[aa^T]$. Theorem 1 will hold as long as there exists a matrix $\Sigma$ such that $\|\nabla F(x) - \Sigma x\|$ stays bounded even if the function $\ell$ is not a quadratic function, the proof is straightforward and would be based on verifying that the conditions of (Mirek, 2011, Theorem 1.4) hold in the setting of Theorem 1. This type of optimization problems arises for instance in robust regression where the objective is $F(x) = \mathbb{E}[(a^Tx - b)^2] + \lambda g(x)$ with a penalty function $g(x)$ whose gradient is bounded and a tunable parameter $\lambda$. The boundedness of the gradient of $g(x)$ results in at-most linear growth of $g(x)$ and allows robustness to outliers where the parameter $\lambda$ can be used to adjust the robustness level desired. Examples for the choice of $g(x)$ include the smoothly clipped

---

[5]Otherwise, one can construct a subsequence $x_{n_k}$ that is bounded in the space $L^\alpha$ converging to $x_\infty$ which would be a contradiction.

absolute deviation (SCAD) penalty (Loh & Wainwright (2015)), Huber loss (Huber, 1992) or the exponential squared loss when $g(x) = 1 - \exp(-\|x\|^2/c)$ where $c$ is a tuning parameter.

**Generalized Central Limit Theorem for Ergodic Averages.** When $\alpha > 2$, by Corollary 5, second moment of the iterates $x_k$ are finite, in which case central limit theorem (CLT) says that if the cumulative sum of the iterates $S_K = \sum_{k=1}^{K} x_k$ is scaled properly, the resulting distribution is Gaussian. In the case where $\alpha < 2$, the variance of the iterates is not finite; however in this case, we derive the following generalized CLT (GCLT) which says that if the iterates are properly scaled, the limit will be an $\alpha$-stable distribution. This is stated in a more precise manner as follows.

**Corollary 9.** *Assume $\rho < 0$ so that Theorem 1 holds. Then, we have the following:*

*(i) If $\alpha \in (0,1) \cup (1,2)$, then there is a sequence $d_K = d_K(\alpha)$ and a function $C_\alpha : \mathbb{S}^{d-1} \mapsto \mathbb{C}$ such that as $K \to \infty$ the random variables $K^{-\frac{1}{\alpha}}(S_K - d_K)$ converge in law to the $\alpha$-stable random variable with characteristic function $\Upsilon_\alpha(tv) = \exp(t^\alpha C_\alpha(v))$, for $t > 0$ and $v \in \mathbb{S}^{d-1}$.*

*(ii) If $\alpha = 1$, then there are functions $\xi, \tau : (0,\infty) \mapsto \mathbb{R}$ and $C_1 : \mathbb{S}^{d-1} \mapsto \mathbb{C}$ such that as $K \to \infty$ the random variables $K^{-1}S_K - K\xi(K^{-1})$ converge in law to the random variable with characteristic function $\Upsilon_1(tv) = \exp(tC_1(v) + it\langle v, \tau(t)\rangle)$, for $t > 0$ and $v \in \mathbb{S}^{d-1}$.*

*(iii) If $\alpha = 2$, then there is a sequence $d_K = d_K(2)$ and a function $C_2 : \mathbb{S}^{d-1} \mapsto \mathbb{R}$ such that as $K \to \infty$ the random variables $(K \log K)^{-\frac{1}{2}}(S_K - d_K)$ converge in law to the random variable with characteristic function $\Upsilon_2(tv) = \exp(t^2 C_2(v))$, for $t > 0$ and $v \in \mathbb{S}^{d-1}$.*

*(iv) If $\alpha \in (0,1)$, then $d_K = 0$, and if $\alpha \in (1,2]$, then $d_K = K\bar{x}$, where $\bar{x} = \int_{\mathbb{R}^d} x\nu_\infty(dx)$.*

In addition to its evident theoretical interest, Corollary 9 has also an important practical implication: estimating the tail-index of a *generic* heavy-tailed distribution is a challenging problem (see e.g. Clauset et al. (2009); Goldstein et al. (2004); Bauke (2007)); however, for the specific case of $\alpha$-stable distributions, accurate and computationally efficient estimators, which *do not* require the knowledge of the functions $C_\alpha, \tau, \xi$, have been proposed (Mohammadi et al., 2015). Thanks to Corollary 9, we will be able to use such estimators in our numerical experiments, as we will detail in the next section.

We finally note that the gradient noise in SGD is actually both multiplicative and additive (Dieuleveut et al., 2017b;a); a fact that is often discarded for simplifying the mathematical analysis. In the linear regression setting, we have shown that the multiplicative noise $M_k$ is the main source of heavy-tails, where a deterministic $M_k$ would not lead to heavy tails.[6] In the light of our theory, in Appendix A, we discuss in detail the recently proposed stochastic differential equation (SDE) representations of SGD in continuous-time and argue that, compared to classical SDEs driven by a Brownian motion (e.g., (Jastrzębski et al., 2017; Cheng et al., 2019), SDEs driven by heavy-tailed $\alpha$-stable Lévy processes (e.g., (Şimşekli et al., 2019b)) are more adequate when $\alpha < 2$.

## 4 EXPERIMENTS

In this section, we present our experimental results on both synthetic and real data, in order to illustrate that our theory also holds in finite-sum problems (besides the streaming setting). Our main goal will be to illustrate the tail behavior of SGD by varying the algorithm parameters: depending on the choice of the stepsize $\eta$ and the batch-size $b$, the iterates do converge to a heavy-tailed distribution (Theorem 1) and the behavior of the tail-index obeys Theorem 2.

**Synthetic experiments.** In our first setting, we consider a simple synthetical setup, where we assume that the data points follow a Gaussian distribution. We will illustrate that the SGD iterates can become heavy-tailed even in this simplistic setting where the problem is a simple linear regression with all the variables being Gaussian. More precisely, we will consider the following model:

$$x_0 \sim \mathcal{N}(0, \sigma_x^2 I), \quad a_i \sim \mathcal{N}(0, \sigma^2 I), \quad y_i | a_i, x_0 \sim \mathcal{N}(a_i^\top x_0, \sigma_y^2), \tag{4.1}$$

where $x_0, a_i \in \mathbb{R}^d, y_i \in \mathbb{R}$ for $i = 1, \ldots, n$, and $\sigma, \sigma_x, \sigma_y > 0$.

In our experiments, we will need to estimate the tail-index $\alpha$ of the stationary distribution $\nu_\infty$. Even though several tail-index estimators have been proposed for generic heavy-tailed distributions in the literature (Paulauskas & Vaičiulis, 2011), we observed that, even for small $d$, these estimators can yield inaccurate estimations and require tuning hyper-parameters, which is non-trivial. We

---

[6]E.g., if $M_k$ is deterministic and $q_k$ is Gaussian, then $x_k$ is Gaussian for all $k$, and so is $x_\infty$ if the limit exists.

circumvent this issue thanks to the GCLT in Corollary 9: since the average of the iterates is guaranteed to converge to a multivariate $\alpha$-stable random variable, we can use the tail-index estimators that are specifically designed for stable distributions. By following Tzagkarakis et al. (2018); Şimşekli et al. (2019b), we use the estimator proposed by Mohammadi et al. (2015), which is fortunately agnostic to the scaling function $C_\alpha$. The details of this estimator are given in Appendix B.

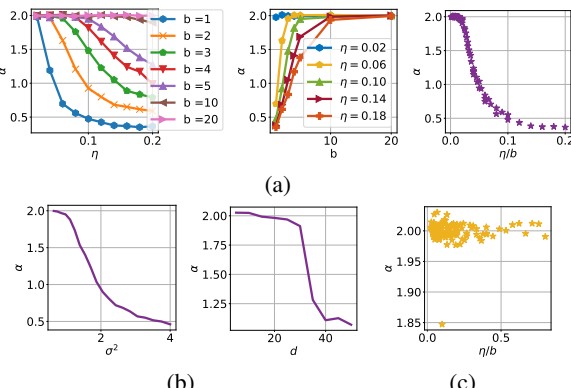

To be able to benefit from the CLT, we are required to compute the average of the 'centered' iterates: $\frac{1}{K-K_0} \sum_{k=K-K_0+1}^{K} (x_k - \bar{x})$, where $K_0$ is a 'burn-in' period aiming to discard the initial phase of SGD, and the mean of $\nu_\infty$ is given by $\bar{x} = \int_{\mathbb{R}^d} x \nu_\infty(dx) = (A^\top A)^{-1} A^\top y$ as long as $\alpha > 1$[7], where the $i$-th row of $A \in \mathbb{R}^{n \times d}$ contains $a_i^\top$ and $y = [y_1, \ldots, y_n] \in \mathbb{R}^n$. We then repeat this procedure 1600 times for different initial points and obtain 1600 different random vectors, whose distributions are supposedly close to an $\alpha$-stable distribution. Finally, we run the tail-index estimator of Mohammadi et al. (2015) on these random vectors to estimate $\alpha$.

Figure 1: Behavior of $\alpha$ with (a) varying stepsize $\eta$ and batch-size $b$, (b) $d$ and $\sigma$, (c) under RMSProp.

In our first experiment, we investigate the tail-index $\alpha$ of the stationary measure $\nu_\infty$ for varying stepsize $\eta$ and batch-size $b$. We set $d = 100$ first fix the variances $\sigma = 1$, $\sigma_x = \sigma_y = 3$, and generate $\{a_i, y_i\}_{i=1}^n$ by simulating the statistical model. Then, by fixing this dataset, we run the SGD recursion (3.3) for a large number of iterations and vary $\eta$ from 0.02 to 0.2 and $b$ from 1 to 20. We also set $K = 1000$ and $K_0 = 500$. Figure 1(a) illustrates the results. We can observe that, increasing $\eta$ and decreasing $b$ both result in decreasing $\alpha$, where the tail-index can be prohibitively small (i.e., $\alpha < 1$, hence even the mean of $\nu_\infty$ is not defined) for large $\eta$. Besides, we can also observe that the tail-index is in strong correlation with the ratio $\eta/b$.

In our second experiment, we investigate the effect of $d$ and $\sigma$ on $\alpha$. In Figure 1(b) (left), we set $d = 100$, $\eta = 0.1$ and $b = 5$ and vary $\sigma$ from 0.8 to 2. For each value of $\sigma$, we simulate a new dataset from by using the generative model and run SGD with $K, K_0$. We again repeat each experiment 1600 times. We follow a similar route for Figure 1(b) (right): we fix $\sigma = 1.75$ and repeat the previous procedure for each value of $d$ ranging from 5 to 50. The results confirm our theory: $\alpha$ decreases for increasing $\sigma$ and $d$, and we can observe that for a fixed $b$ and $\eta$ the change in $d$ can abruptly alter $\alpha$.

In our final synthetic data experiment, we investigate how the tails behave under adaptive optimization algorithms. We replicate the setting of our first experiment, with the only difference that we replace SGD with RMSProp (Hinton et al., 2012). As shown in Figure 1(c), the 'clipping' effect of RMSProp as reported in Zhang et al. (2019) prevents the iterates become heavy-tailed and the vast majority of the estimated tail-indices is around 2, indicating a Gaussian behavior. On the other hand, we repeated the same experiment with the variance-reduced optimization algorithm SVRG (Johnson & Zhang, 2013), and observed that for almost all choices of $\eta$ and $b$ the algorithm converges near the minimizer (with an error in the order of $10^{-6}$), hence the stationary distribution $\nu_\infty$ seems to be a degenerate distribution, which does not admit a heavy-tailed behavior. Regarding the observed link between heavy-tails and generalization (Martin & Mahoney, 2019; Şimşekli et al., 2020), this behavior of RMSProp and SVRG might be related to their ineffective generalization performance as reported in Keskar & Socher (2017); Defazio & Bottou (2019).

**Experiments on fully connected neural networks.** In the second set of experiments, we investigate the applicability of our theory beyond the quadratic optimization problems. Here, we follow the setup of Şimşekli et al. (2019a) and consider a fully connected neural network with the cross entropy

---

[7]The form of $\bar{x}$ can be verified by noticing that $\mathbb{E}[x_k]$ converges to the minimizer of the problem by the law of total expectation. Besides, our GCLT requires the sum of the iterates to be normalized by $\frac{1}{(K-K_0)^{1/\alpha}}$; however, for a finite $K$, normalizing by $\frac{1}{K-K_0}$ results in a scale difference, to which our tail-index estimator is agnostic.

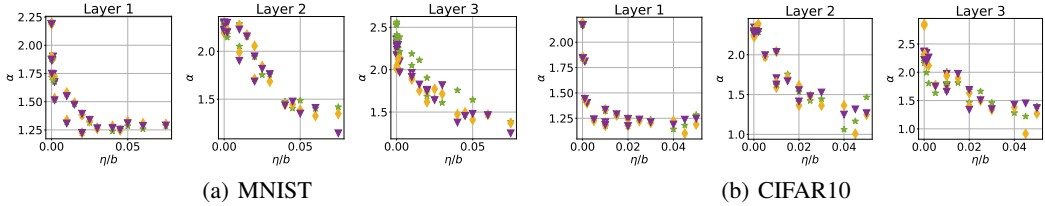

(a) MNIST        (b) CIFAR10

Figure 2: Results on FCNs. Different markers represent different initializations with the same $\eta$, $b$.

loss and ReLU activation functions on the MNIST and CIFAR10 datasets. We train the models by using SGD for 10K iterations and we range $\eta$ from $10^{-4}$ to $10^{-1}$ and $b$ from 1 to 10. Since it would be computationally infeasible to repeat each run thousands of times as we did in the synthetic data experiments, in this setting we follow a different approach based on (i) (Şimşekli et al., 2019a) that suggests that the tail behavior can differ in different layers of a neural network, and (ii) (De Bortoli et al., 2020) that shows that in the infinite width limit, the different components of a given layer of a two-layer fully connected network (FCN) becomes independent. Accordingly, we first compute the average of the last 1K SGD iterates, whose distribution should be close an $\alpha$-stable distribution by the GCLT. We then treat each layer as a collection of i.i.d. $\alpha$-stable random variables and measure the tail-index of each individual layer separately by using the the estimator from Mohammadi et al. (2015). Figure 2 shows the results for a three-layer network (with 128 hidden units at each layer) , whereas we obtained very similar results with a two-layer network as well. We observe that, while the dependence of $\alpha$ on $\eta/b$ differs from layer to layer, in each layer the measured $\alpha$ correlate very-well with the ratio $\eta/b$ in both datasets.

**Experiments on VGG networks.** In our last set of experiments, we evaluate our theory on VGG networks (Simonyan & Zisserman, 2015) with 11 layers (10 convolutional layers with max-pooling and ReLU units, followed by a final linear layer), which contains 10M parameters. We follow the same procedure as we used for the fully connected networks, where we vary $\eta$ from $10^{-4}$ to $1.7 \times 10^{-3}$ and $b$ from 1 to 10.

The results are shown in Figure 3. Similar to the previous experiments, we observe that $\alpha$ depends on the layer. For the intermediate layers (Layers 2-8), the tail index correlates well with the ratio $\eta/b$, whereas the first and last two convolutional layers (Layers 9 and 10)

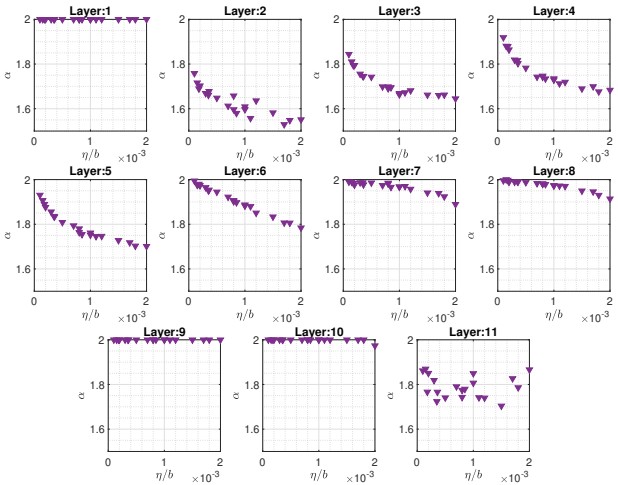

Figure 3: Results on VGG networks. The values of $\alpha$ that exceeded 2 is truncated to 2 for visualization purposes.

exhibit a Gaussian behavior ($\alpha \approx 2$). On the other hand, the tail-index of the last layer (which is linear) does not correlate with either $\eta$ or $b$. These observations provide further support for our theory and show that the heavy-tail phenomenon also occurs in neural networks, whereas $\alpha$ is potentially related to $\eta$ and $b$ in a more complicated way.

## 5   Conclusion and Future Directions

We studied the tail behavior of the SGD in a quadratic optimization problem and showed that depending on the curvature, $\eta$, and $b$, the iterates can converge to a *heavy-tailed* random variable. We further supported our theory with experiments conducted on fully connected neural networks and illustrated that our results would also apply to more general settings and hence provide new insights about the behavior of SGD in deep learning. This study also brings up a number of future directions. (i) Our proof techniques are for the streaming setting, where each sample is used only once. However, in practice SGD is typically implemented on the finite-sum problem (1.2) with multiple passes over the data. Extending our results to this scenario and investigating the effects of finite-sample size on the tail index and generalization would be an interesting future research direction. (ii) We suspect that the tail index of the SGD iterates may have an impact on the time required to escape a saddle point and this can be investigated further as another future research direction.

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

## A A NOTE ON STOCHASTIC DIFFERENTIAL EQUATION REPRESENTATIONS FOR SGD

In recent years, a popular approach for analyzing the behavior of SGD has been viewing it as a discretization of a continuous-time stochastic process that can be represented via a stochastic differential equation (SDE) (Mandt et al., 2016; Jastrzębski et al., 2017; Li et al., 2017; Hu et al., 2019; Zhu et al., 2018; Chaudhari & Soatto, 2018; Şimşekli et al., 2019b). While these SDEs have been useful for understanding different properties of SGD, their differences and functionalities have not been clearly understood. In this section, in the light of our theoretical results, we will discuss in which situation their choice would be more appropriate. We will restrict ourselves to the case where $f(x)$ is a quadratic function; however, the discussion can be extended to more general $f$.

The SDE approximations are often motivated by first rewriting the SGD recursion as follows:

$$x_{k+1} = x_k - \eta \nabla \tilde{f}_{k+1}(x_k) = x_k - \eta \nabla f(x_k) + \eta U_{k+1}(x_k), \tag{A.1}$$

where $U_k(x) := \nabla \tilde{f}_k(x) - \nabla f(x)$ is called the 'stochastic gradient noise'. Then, based on certain statistical assumptions on $U_k$, we can view (A.1) as a discretization of an SDE. For instance, if we assume that the gradient noise follows a Gaussian distribution, whose covariance does not depend on the iterate $x_k$, i.e., $\eta U_k \approx \sqrt{\eta} Z_k$ where $Z_k \sim \mathcal{N}(0, \sigma_z \eta I)$ for some constant $\sigma_z > 0$, we can see (A.1) as the Euler-Maruyama discretization of the following SDE with stepsize $\eta$ (Mandt et al., 2016):

$$\mathrm{d}x_t = -\nabla f(x_t)\mathrm{d}t + \sqrt{\eta \sigma_z}\mathrm{d}\mathrm{B}_t, \tag{A.2}$$

where $\mathrm{B}_t$ denotes the $d$-dimensional standard Brownian motion. This process is called the Ornstein-Uhlenbeck (OU) process (see e.g. Øksendal (2013)), whose invariant measure is a Gaussian distribution. We argue that this process can be a good proxy to (3.3) only when $\alpha \geq 2$, since otherwise the SGD iterates will exhibit heavy-tails, whose behavior cannot be captured by a Gaussian distribution. As we illustrated in Section 4, to obtain large $\alpha$, the step-size $\eta$ needs to be small and/or the batch-size $b$ needs to be large. However, it is clear that this approximation will fall short when the system exhibits heavy tails, i.e., $\alpha < 2$. Therefore, for the large $\eta/b$ regime, which appears to be more interesting since it often yields improved test performance (Jastrzębski et al., 2017), this approximation would be inaccurate for understanding the behavior of SGD. This problem mainly stems from the fact that the additive isotropic noise assumption results in a deterministic $M_k$ matrix for all $k$. Since there is no *multiplicative noise* term, this representation cannot capture a potential heavy-tailed behavior.

A natural extension of the state-independent Gaussian noise assumption is to incorporate the covariance structure of $U_k$. In our linear regression problem, we can easily see that the covariance matrix of the gradient noise has the following form:

$$\Sigma_U(x) = \mathrm{Cov}(U_k|x) = \frac{\sigma^2}{b}\mathrm{diag}(x \circ x), \tag{A.3}$$

where $\circ$ denotes element-wise multiplication and $\sigma^2$ is the variance of the data points. Therefore, we can extend the previous assumption by assuming $Z_k|x \sim \mathcal{N}(0, \eta \Sigma_U(x))$. It has been observed that this approximation yields a more accurate representation (Cheng et al., 2019; Ali et al., 2020; Jastrzębski et al., 2017). Using this assumption in (A.1), the SGD recursion coincides with the Euler-Maruyama discretization of the following SDE:

$$dx_t = -\nabla f(x_t)dt + \sqrt{\eta \Sigma_U(x_t)}d\mathrm{B}_t$$
$$\overset{\mathrm{d}}{=} -\left(A^\top A x_t - A^\top y\right)dt + \sqrt{\frac{\sigma^2 \eta}{b}}\mathrm{diag}(x_t)d\mathrm{B}_t, \tag{A.4}$$

where $\overset{\mathrm{d}}{=}$ denotes equality in distribution. The stochasticity in such SDEs is called often called *multiplicative*. Let us illustrate this property by discretizing this process and by using the definition of the gradient and the covariance matrix, we observe that (noting that $N_k \sim \mathcal{N}(0, I)$)

$$x_{k+1} = x_k - \eta\left(A^\top A x_k - A^\top y\right) + \sqrt{\frac{\sigma^2 \eta^2}{b}}\mathrm{diag}(x_k)N_{k+1}$$
$$= \left(I - \eta A^\top A + \sqrt{\sigma^2 \eta^2/b}\,\mathrm{diag}(N_{k+1})\right)x_k - \eta A^\top y, \tag{A.5}$$

where we can clearly see the multiplicative effect of the noise, as indicated by its name. On the other hand, we can observe that, thanks to the multiplicative structure, this process would be able to capture the potential heavy-tailed structure of SGD. However, there are two caveats. The first one is that, in the case of linear regression, the process is called a geometric (or modified) Ornstein-Uhlenbeck process which is an extension of geometric Brownian motion. One can show that the distribution of the process at any time $t$ will have lognormal tails. Hence it will be accurate only when the tail-index $\alpha$ is close to the one of the lognormal distribution. The second caveat is that, for a more general cost function $f$, the covariance matrix is more complicated and hence the invariant measure of the process cannot be found analytically, hence analyzing these processes for a general $f$ can be as challenging as directly analyzing the behavior of SGD.

The third way of modeling the gradient noise is based on assuming that it is heavy-tailed. In particular, we can assume that $\eta U_k \approx \eta^{1/\alpha} L_k$ where $[L_k]_i \sim \mathcal{S}\alpha\mathcal{S}(\sigma_L \eta^{(\alpha-1)/\alpha})$ for all $i = 1, \ldots, d$. Under this assumption the SGD recursion coincides with the Euler discretization of the following Lévy-driven SDE (Şimşekli et al., 2019b):

$$dx_t = -\nabla f(x_t)dt + \sigma_L \eta^{(\alpha-1)/\alpha} d\mathrm{L}_t^\alpha, \tag{A.6}$$

where $\mathrm{L}_t^\alpha$ denotes the $\alpha$-stable Lévy process with independent components (see Section A.1 for technical background on Lévy processes and in particular $\alpha$-stable Lévy processes). In the case of linear regression, this processes is called a fractional OU process (Fink & Klüppelberg, 2011), whose invariant measure is also an $\alpha$-stable distribution with the same tail-index $\alpha$. Hence, even though it is based on an isotropic, state-independent noise assumption, in the case of large $\eta/b$ regime, this approach can mimic the heavy-tailed behavior of the system with the exact tail-index $\alpha$. On the other hand, Buraczewski et al. (2016) (Theorem 1.7 and 1.16) showed that if $U_k$ is assumed to heavy tailed with index $\alpha$ (not necessarily $\mathcal{S}\alpha\mathcal{S}$) then the process $x_k$ will inherit the same tails and the ergodic averages will still converge to an $\mathcal{S}\alpha\mathcal{S}$ random variable, hence generalizing the conclusions of the $\mathcal{S}\alpha\mathcal{S}$ assumption to the case where $U_k$ follows an arbitrary heavy-tailed distribution.

## A.1 TECHNICAL BACKGROUND: LÉVY PROCESSES

Lévy motions (processes) are stochastic processes with independent and stationary increments, which include Brownian motions as a special case, and in general may have heavy-tailed distributions (see e.g. Bertoin (1996) for a survey). Symmetric $\alpha$-stable Lévy motion is a Lévy motion whose time increments are symmetric $\alpha$-stable distributed. We define $\mathrm{L}_t^\alpha$, a $d$-dimensional symmetric $\alpha$-stable Lévy motion as follows. Each component of $\mathrm{L}_t^\alpha$ is an independent scalar $\alpha$-stable Lévy process defined as follows:

(i) $\mathrm{L}_0^\alpha = 0$ almost surely;

(ii) For any $t_0 < t_1 < \cdots < t_N$, the increments $\mathrm{L}_{t_n}^\alpha - \mathrm{L}_{t_{n-1}}^\alpha$ are independent, $n = 1, 2, \ldots, N$;

(iii) The difference $\mathrm{L}_t^\alpha - \mathrm{L}_s^\alpha$ and $\mathrm{L}_{t-s}^\alpha$ have the same distribution: $\mathcal{S}\alpha\mathcal{S}((t-s)^{1/\alpha})$ for $s < t$;

(iv) $\mathrm{L}_t^\alpha$ has stochastically continuous sample paths, i.e. for any $\delta > 0$ and $s \geq 0$, $\mathbb{P}(|\mathrm{L}_t^\alpha - \mathrm{L}_s^\alpha| > \delta) \to 0$ as $t \to s$.

When $\alpha = 2$, we obtain a scaled Brownian motion as a special case, i.e. $\mathrm{L}_t^\alpha = \sqrt{2}\mathrm{B}_t$, so that the difference $\mathrm{L}_t^\alpha - \mathrm{L}_s^\alpha$ follows a Gaussian distribution $\mathcal{N}(0, 2(t-s))$.

## B  TAIL-INDEX ESTIMATION

In this study, we follow Tzgkarakis et al. (2018); Şimşekli et al. (2019b), and make use of the recent estimator proposed by Mohammadi et al. (2015).

**Theorem 10** (Mohammadi et al. (2015) Corollary 2.4). *Let $\{X_i\}_{i=1}^K$ be a collection of strictly stable random variables in $\mathbb{R}^d$ with tail-index $\alpha \in (0, 2]$ and $K = K_1 \times K_2$. Define $Y_i = \sum_{j=1}^{K_1} X_{j+(i-1)K_1}$ for $i \in [\![1, K_2]\!]$. Then, the estimator*

$$\widehat{\frac{1}{\alpha}} \triangleq \frac{1}{\log K_1}\left(\frac{1}{K_2}\sum_{i=1}^{K_2}\log\|Y_i\| - \frac{1}{K}\sum_{i=1}^{K}\log\|X_i\|\right), \tag{B.1}$$

*converges to $1/\alpha$ almost surely, as $K_2 \to \infty$.*

As this estimator requires a hyperparameter $K_1$, at each tail-index estimation, we used several values for $K_1$ and we used the median of the estimators obtained with different values of $K_1$. We provide the codes in the supplement, where the implementation details can be found. For the neural network experiments, we used the same setup as provided in the repository of Şimşekli et al. (2019b).

## C PROOFS OF MAIN RESULTS

### C.1 PROOF OF THEOREM 1

*Proof of Theorem 1.* The proof follows from (Buraczewski et al., 2016, Thm 4.4.15) which goes back to (Alsmeyer & Mentemeier, 2012, Theorem 1.1) and Kesten (Kesten, 1973, Theorem 6). See also (Goldie, 1991; Buraczewski et al., 2015). We recall that we have the stochastic recursion:

$$x_k = M_k x_{k-1} + q_k, \tag{C.1}$$

where the sequence $(M_k, q_k)$ are i.i.d. distributed as $(M, q)$ and for each $k$, $(M_k, q_k)$ is independent of $x_{k-1}$. To apply (Buraczewski et al., 2016, Thm 4.4.15), it suffices to have the following conditions being satisfied:

1. $M$ is invertible with probability 1.

2. The matrix $M$ has a continuous Lebesgue density that is positive in a neighborhood of the identity matrix.

3. $\rho < 0$ and $h(\alpha) = 1$.

4. $\mathbb{P}(Mx + q = x) < 1$ for every $x$.

5. $\mathbb{E}\left[\|M\|^\alpha (\log^+ \|M\| + \log^+ \|M^{-1}\|)\right] < \infty$.

6. $0 < \mathbb{E}\|q\|^\alpha < \infty$.

All the conditions are satisfied under our assumptions. In particular, Condition 1 and Condition 5 are proved in Lemma 18, and Condition 2 and Condition 4 follow from the fact that $M$ and $q$ have continuous distributions. If $\rho < 0$, then by Lemma 15, we have $h(0) = 1$, $h'(0) = \rho < 0$ and $h(s)$ is convex in $s$, and moreover by Lemma 16, we have $\liminf_{s \to \infty} h(s) > 1$. Therefore, there exists some $\alpha \in (0, \infty)$ such that $h(\alpha) = 1$, which gives Condition 3. Finally, Condition 6 is satisfied by the definition of $q$ and by the Assumptions **(A1)–(A2)**. □

### C.2 PROOF OF THEOREM 2

*Proof of Theorem 2.* We will split the proof of Theorem 2 into two parts:

(I) We will show that the tail-index $\alpha$ is strictly decreasing in stepsize $\eta$ and variance $\sigma^2$ provided that $\alpha \geq 1$.

(II) We will show that the tail-index $\alpha$ is strictly increasing in batch size $b$ provided that $\alpha \geq 1$.

(III) We will show that the tail-index $\alpha$ is strictly decreasing in dimension $d$.

First, let us prove (I). Let $a := \eta\sigma^2 > 0$ be given. Consider the tail-index $\alpha$ as a function of $a$, i.e.

$$\alpha(a) := \min\{s : h(a, s) = 1\},$$

where $h(a, s) = h(s)$ with emphasis on dependence on $a$.

By assumption, $\alpha(a) \geq 1$. The function $h(a, s)$ is convex function of $a$ (see Lemma 19 for $s \geq 1$ and a strictly convex function of $s$ for $s \geq 0$). Furthermore, it satisfies $h(a, 0) = 1$ for every $a \geq 0$ and $h(0, s) = 1$ for every $s \geq 0$. We consider the curve

$$\mathcal{C} := \{(a, s) \in (0, \infty) \times [1, \infty] : h(a, s) = 1\}.$$

This is the set of the choice of $a$, which leads to a tail-index $s$ where $s \geq 1$. Since $h$ is smooth in both $a$ and $s$, we can represent $s$ as a smooth function of $a$, i.e. on the curve

$$h(a, s(a)) = 0,$$

where $s(a)$ is a smooth function of $a$. We will show that $s'(a) < 0$; i.e. if we increase $a$; the tail-index $s(a)$ will drop. Pick any $(a_*, s_*) \in \mathcal{C}$, it will satisfy $h(a_*, s_*) = 1$. We have the following facts:

(i) The function $h(a, s) = 1$ for either $a = 0$ or $s = 0$. This is illustrated in Figure 4 with a blue marker.

(ii) $h(a_*, s) < 1$ for $s < s_*$. This follows from the convexity of $h(a_*, s)$ function and the fact that $h(a_*, 0) = 1$, $h(a_*, s_*) = 1$. From here, we see that the function $h(a_*, s)$ is increasing at $s = s_*$ and we have its derivative

$$\frac{\partial h}{\partial s}(a_*, s_*) > 0.$$

(iii) The function $h(a, s_*)$ is convex as a function of $a$ by Lemma 19, it satisfies $h(0, s_*) = h(a_*, s_*) = 1$. Therefore, by convexity $h(a, s_*) < 1$ for a $\in (0, s_*)$; otherwise the function $h(a, s_*)$ would be a constant function. We have therefore necessarily.

$$\frac{\partial h}{\partial a}(a_*, s_*) > 0.$$

By convexity of the function $h(a, s_*)$, we have also $h(a, s_*) \geq h(a_*, s_*) + \frac{\partial h}{\partial a}(a_*, s_*)(a - a_*) > h(a_*, s_*) = 1$. Therefore, $h(a, s_*) > 1$ for $a > a_*$. Then, it also follows that $h(a, s) > 1$ for $a > a_*$ and $s > s_*$ (otherwise if $h(a, s) \leq 1$, we get a contradiction because $h(0, s) = 1$, $h(a_*, s) > 1$ and $h(a, s) \leq 1$ is impossible due to convexity). This is illustrated in Figure 4 where we mark this region as a rectangular box where $h > 1$.

(iv) By similar arguments we can show that the function $h(a, s) < 1$ if $(s, a) \in (0, a_*) \times [1, s_*)$. Indeed, if $h(a, s) \geq 1$ for some $(s, a) \in [1, s_*) \times (0, a_*)$, this contradicts the fact that $h(0, s) = 1$ and $h(a_*, s) < 1$ proven in part $(ii)$. This is illustrated in Figure 4 where inside the rectangular box on the left-hand side, we have $h < 1$.

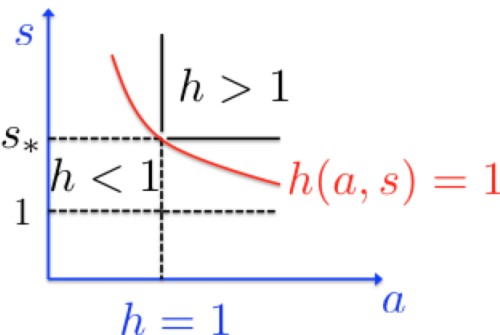

Figure 4: The curve $h(a, s) = 1$ in the $(a, s)$ plane

Geometrically, we see from Figure 4 that the curve $s(a)$ as a function of $a$, is sandwiched between two rectangular boxes and has necessarily $s'(a) < 0$. This can also be directly obtained rigorously from the implicit function theorem; if we differentiate the implicit equation $h(a, s(a)) = 0$ with respect to $a$, we obtain

$$\frac{\partial h}{\partial a}(a_*, s_*) + \frac{\partial h}{\partial s}(a_*, s_*)s'(a_*) = 0.$$

From parts $(ii) - (iii)$, we have $\frac{\partial h}{\partial a}(a_*, s_*)$ and $\frac{\partial h}{\partial s}(a_*, s_*) > 0$. Therefore, we have

$$s'(a_*) = -\frac{\frac{\partial h}{\partial a}(a_*, s_*)}{\frac{\partial h}{\partial s}(a_*, s_*)} < 0, \tag{C.2}$$

which completes the proof for $s_* \geq 1$.

Next, let us prove (II). With slight abuse of notation, we define the function $h(b,s) = h(s)$ to emphasize the dependence on $b$. We have

$$h(b,s) = \mathbb{E}\left\|\left(I - \frac{\eta}{b}\sum_{i=1}^b a_i a_i^T\right)e_1\right\|^s. \tag{C.3}$$

where we used Lemma 14. When $s \geq 1$, the function $x \mapsto \|x\|^s$ is convex, and by Jensen's inequality, we get for any $b \geq 2$ and $b \in \mathbb{N}$,

$$
\begin{aligned}
h(b,s) &= \mathbb{E}\left\|\frac{1}{b}\sum_{i=1}^b \left(I - \frac{\eta}{b-1}\sum_{j\neq i} a_j a_j^T\right)e_1\right\|^s \\
&\leq \mathbb{E}\left[\frac{1}{b}\sum_{i=1}^b \left\|\left(I - \frac{\eta}{b-1}\sum_{j\neq i} a_j a_j^T\right)e_1\right\|^s\right] \\
&= \frac{1}{b}\sum_{i=1}^b \mathbb{E}\left[\left\|\left(I - \frac{\eta}{b-1}\sum_{j\neq i} a_j a_j^T\right)e_1\right\|^s\right] = h(b-1,s),
\end{aligned}
$$

where we used the fact that $a_i$ are i.i.d. Indeed, from the condition for equality to hold in Jensen's inequality, and the fact that $a_i$ are i.i.d. random, the inequality above is a strict inequality. Hence when $d \in \mathbb{N}$ for any $s \geq 1$, $h(b,s)$ is strictly decreasing in $b$. By following the same argument as in the proof of (I), we conclude that the tail-index $\alpha$ is strictly increasing in batch size $b$.

Finally, let us prove (III). Let us show the tail-index $\alpha$ is strictly decreasing in dimension $d$. Since $a_i$ are i.i.d. and $a_i \sim \mathcal{N}(0, \sigma^2 I_d)$, by Lemma 25,

$$h(s) = \mathbb{E}\left[\left(1 - \frac{2a}{b}X + \frac{a^2}{b^2}X^2 + \frac{a^2}{b^2}XY\right)^{s/2}\right], \tag{C.4}$$

where $X, Y$ are independent chi-square random variables with degree of freedom $b$ and $d-1$ respectively. Notice that $h(s)$ is strictly increasing in $d$ since the only dependence of $h(s)$ on $d$ is via $Y$, which is a chi-square distribution with degree of freedom $(d-1)$. By writing $Y = Z_1^2 + \cdots + Z_{d-1}^2$, where $Z_i \sim N(0,1)$ i.i.d., it follows that $h(s)$ is strictly increasing in $d$. Hence, by similar argument as in (I), we conclude that $\alpha$ is strictly decreasing in dimension $d$. $\square$

**Remark 11.** *When $d = 1$ and $a_i$ are i.i.d. $N(0, \sigma^2)$, we can provide an alternative proof that the tail-index $\alpha$ is strictly increasing in batch size $b$. It suffices to show that for any $s \geq 1$, $h(s)$ is strictly decreasing in the batch size $b$. By Lemma 25 when $d = 1$,*

$$h(b,s) = \mathbb{E}\left[\left(1 - \frac{2\eta\sigma^2}{b}X + \frac{\eta^2\sigma^4}{b^2}X^2 + \frac{\eta^2\sigma^4}{b^2}XY\right)^{s/2}\right], \tag{C.5}$$

*where $h(b,s)$ is as in (C.3) and $X, Y$ are independent chi-square random variables with degree of freedom $b$ and $d-1$ respectively. When $d = 1$, we have $Y \equiv 0$, and*

$$h(b,s) = \mathbb{E}\left[\left(1 - \frac{2\eta\sigma^2}{b}X + \frac{\eta^2\sigma^4}{b^2}X^2\right)^{s/2}\right] = \mathbb{E}\left[\left|1 - \frac{\eta\sigma^2}{b}X\right|^s\right]. \tag{C.6}$$

*Since $X$ is a chi-square random variable with degree of freedom $b$, we have*

$$h(b,s) = \mathbb{E}\left[\left|1 - \frac{\eta\sigma^2}{b}\sum_{i=1}^b Z_i\right|^s\right], \tag{C.7}$$

*where $Z_i$ are i.i.d. $N(0,1)$ random variables. When $s \geq 1$, the function $x \mapsto |x|^s$ is convex, and by Jensen's inequality, we get for any $b \geq 2$ and $b \in \mathbb{N}$*

$$h(b,s) = \mathbb{E}\left[\left|\frac{1}{b}\sum_{i=1}^{b}\left(1 - \frac{\eta\sigma^2}{b-1}\sum_{j\neq i}Z_j\right)\right|^s\right]$$

$$\leq \mathbb{E}\left[\frac{1}{b}\sum_{i=1}^{b}\left|1 - \frac{\eta\sigma^2}{b-1}\sum_{j\neq i}Z_j\right|^s\right]$$

$$= \frac{1}{b}\sum_{i=1}^{b}\mathbb{E}\left[\left|1 - \frac{\eta\sigma^2}{b-1}\sum_{j\neq i}Z_j\right|^s\right] = h(b-1,s),$$

*where we used the fact that $Z_i$ are i.i.d. Indeed, from the condition for equality to hold in Jensen's inequality, and the fact that $Z_i$ are i.i.d. $N(0,1)$ distributed, the inequality above is a strict inequality. Hence when $d = 1$ for any $s \geq 1$, $h(b,s)$ is strictly decreasing in $b$.*

### C.3 PROOF OF PROPOSITION 3

*Proof of Proposition 3.* We next prove (i). When $\eta = \eta_{crit} = \frac{2b}{\sigma^2(d+b+1)}$, that is $\eta\sigma^2(d+b+1) = 2b$, it follows from the proof of Proposition 23 that

$$\rho \leq \frac{1}{2}\log\mathbb{E}\left[1 - \frac{2\eta\sigma^2}{b}\sum_{i=1}^{b}z_{i1}^2 + \frac{\eta^2\sigma^4}{b^2}\sum_{i=1}^{b}\sum_{j=1}^{b}(z_{i1}z_{j1} + \cdots + z_{id}z_{jd})z_{i1}z_{j1}\right] = 0. \quad \text{(C.8)}$$

Note that since $1 - \frac{2\eta\sigma^2}{b}\sum_{i=1}^{b}z_{i1}^2 + \frac{\eta^2\sigma^4}{b^2}\sum_{i=1}^{b}\sum_{j=1}^{b}(z_{i1}z_{j1} + \cdots + z_{id}z_{jd})z_{i1}z_{j1}$ is random, the inequality above is a strict inequality from Jensen's inequality. Thus, when $\eta = \eta_{crit}$, i.e. $\eta\sigma^2(d+b+1) = 2b$, $\rho < 0$. By continuity, there exists some $\delta > 0$ such that for any $2b < \eta\sigma^2(d+b+1) < 2b+\delta$, i.e. $\eta_{crit} < \eta < \eta_{max}$, where $\eta_{max} := \eta_{crit} + \frac{\delta}{\sigma^2(d+b+1)}$, we have $\rho < 0$. Moreover, when $\eta\sigma^2(d+b+1) > 2b$, i.e. $\eta > \eta_{crit}$, we have

$$h(2) = \mathbb{E}\left[\left(1 - \frac{2\eta\sigma^2}{b}\sum_{i=1}^{b}z_{i1}^2 + \frac{\eta^2\sigma^4}{b^2}\sum_{i=1}^{b}\sum_{j=1}^{b}(z_{i1}z_{j1} + \cdots + z_{id}z_{jd})z_{i1}z_{j1}\right)\right]$$

$$= 1 - 2\eta\sigma^2 + \frac{\eta^2\sigma^4}{b}(d+b+1) \geq 1,$$

which implies that there exists some $0 < \alpha < 2$ such that $h(\alpha) = 1$.

Finally, let us prove (ii) and (iii). When $\eta\sigma^2(d+b+1) \leq 2b$, i.e. $\eta \leq \eta_{crit}$, we have $h(2) \leq 1$, which implies that $\alpha > 2$. In particular, when $\eta\sigma^2(d+b+1) = 2b$, i.e. $\eta = \eta_{crit}$, the tail-index $\alpha = 2$. $\qquad\square$

### C.4 PROOF OF THEOREM 4 AND COROLLARY 5

*Proof of Theorem 4.* We recall that

$$x_k = M_k x_{k-1} + q_k, \quad \text{(C.9)}$$

which implies that

$$\|x_k\| \leq \|M_k x_{k-1}\| + \|q_k\|. \quad \text{(C.10)}$$

(i) If the tail-index $\alpha \leq 1$, then for any $0 < p < \alpha$, we have $h(p) = \mathbb{E}\|M_k e_1\|^p < 1$ and moreover by Lemma 20,

$$\|x_k\|^p \leq \|M_k x_{k-1}\|^p + \|q_k\|^p. \quad \text{(C.11)}$$

Due to spherical symmetry of the isotropic Gaussian distribution, the distribution of $\frac{\|M_k x\|}{\|x\|}$ does not depend on the choice of $x \in \mathbb{R}^d\backslash\{0\}$. Therefore, $\frac{\|M_k x_{k-1}\|}{\|x_{k-1}\|}$ and $\|x_{k-1}\|$ are independent, and

$\frac{\|M_k x_{k-1}\|}{\|x_{k-1}\|}$ has the same distribution as $\|M_k e_1\|$, where $e_1$ is the first basis vector. It follows that

$$\mathbb{E}\|x_k\|^p \leq \mathbb{E}\|M_k e_1\|^p \mathbb{E}\|x_{k-1}\|^p + \mathbb{E}\|q_k\|^p, \tag{C.12}$$

so that

$$\mathbb{E}\|x_k\|^p \leq h(p)\mathbb{E}\|x_{k-1}\|^p + \mathbb{E}\|q_1\|^p, \tag{C.13}$$

where $h(p) \in (0, 1)$. By iterating over $k$, we get

$$\mathbb{E}\|x_k\|^p \leq (h(p))^k \mathbb{E}\|x_0\|^p + \frac{1 - (h(p))^k}{1 - h(p)} \mathbb{E}\|q_1\|^p. \tag{C.14}$$

(ii) If the tail-index $\alpha > 1$, then for any $1 < p < \alpha$, by Lemma 20, for any $\epsilon > 0$, we have

$$\|x_k\|^p \leq (1 + \epsilon)\|M_k x_{k-1}\|^p + \frac{(1 + \epsilon)^{\frac{p}{p-1}} - (1 + \epsilon)}{\left((1 + \epsilon)^{\frac{1}{p-1}} - 1\right)^p}\|q_k\|^p, \tag{C.15}$$

which (similar as in (i)) implies that

$$\mathbb{E}\|x_k\|^p \leq (1 + \epsilon)\mathbb{E}\|M_k e_1\|^p \mathbb{E}\|x_{k-1}\|^p + \frac{(1 + \epsilon)^{\frac{p}{p-1}} - (1 + \epsilon)}{\left((1 + \epsilon)^{\frac{1}{p-1}} - 1\right)^p}\mathbb{E}\|q_k\|^p, \tag{C.16}$$

so that

$$\mathbb{E}\|x_k\|^p \leq (1 + \epsilon)h(p)\mathbb{E}\|x_{k-1}\|^p + \frac{(1 + \epsilon)^{\frac{p}{p-1}} - (1 + \epsilon)}{\left((1 + \epsilon)^{\frac{1}{p-1}} - 1\right)^p}\mathbb{E}\|q_1\|^p. \tag{C.17}$$

We choose $\epsilon > 0$ so that $(1 + \epsilon)h(p) < 1$. By iterating over $k$, we get

$$\mathbb{E}\|x_k\|^p \leq ((1 + \epsilon)h(p))^k \mathbb{E}\|x_0\|^p + \frac{1 - ((1 + \epsilon)h(p))^k}{1 - (1 + \epsilon)h(p)} \frac{(1 + \epsilon)^{\frac{p}{p-1}} - (1 + \epsilon)}{\left((1 + \epsilon)^{\frac{1}{p-1}} - 1\right)^p}\mathbb{E}\|q_1\|^p. \tag{C.18}$$

The proof is complete. $\qquad\square$

**Remark 12.** *In general, there is no closed-form expression for $\mathbb{E}\|q_1\|^p$ in Theorem 4. We provide an upper bound as follows. When $p > 1$, by Jensen's inequality, we can compute that*

$$\mathbb{E}\|q_1\|^p = \eta^p \mathbb{E}\left\|\frac{1}{b}\sum_{i=1}^{b} a_i y_i\right\|^p \leq \frac{\eta^p}{b}\sum_{i=1}^{b}\mathbb{E}\|a_i y_i\|^p = \eta^p \mathbb{E}\left[|y_1|^p \|a_1\|^p\right], \tag{C.19}$$

*and when $p \leq 1$, by Lemma 20, we can compute that*

$$\mathbb{E}\|q_1\|^p = \frac{\eta^p}{b^p}\mathbb{E}\left\|\sum_{i=1}^{b} a_i y_i\right\|^p \leq \frac{\eta^p}{b^p}\mathbb{E}\left[\left(\sum_{i=1}^{b}\|a_i y_i\|\right)^p\right] \leq \frac{\eta^p}{b^p}\sum_{i=1}^{b}\mathbb{E}\|a_i y_i\|^p = \eta^p \mathbb{E}\left[|y_1|^p \|a_1\|^p\right]. \tag{C.20}$$

*Proof of Corollary 5.* It follows from Theorem 4 by letting $k \to \infty$ and applying Fatou's lemma. $\quad\square$

### C.5 PROOF OF THEOREM 6, COROLLARY 7, PROPOSITION 8 AND COROLLARY 9

*Proof of Theorem 6.* For any $\nu_0, \tilde{\nu}_0 \in \mathcal{P}_p(\mathbb{R}^d)$, there exists a couple $x_0 \sim \nu_0$ and $\tilde{x}_0 \sim \tilde{\nu}_0$ independent of $(M_k, q_k)_{k \in \mathbb{N}}$ and $\mathcal{W}_p^p(\nu_0, \tilde{\nu}_0) = \mathbb{E}\|x_0 - \tilde{x}_0\|^p$. We define $x_k$ and $\tilde{x}_k$ starting from $x_0$ and $\tilde{x}_0$ respectively, via the iterates

$$x_k = M_k x_{k-1} + q_k, \tag{C.21}$$
$$\tilde{x}_k = M_k \tilde{x}_{k-1} + q_k, \tag{C.22}$$

and let $\nu_k$ and $\tilde{\nu}_k$ denote the probability laws of $x_k$ and $\tilde{x}_k$ respectively. For any $p < \alpha$, since $\mathbb{E}\|M_k\|^\alpha = 1$ and $\mathbb{E}\|q_k\|^\alpha < \infty$, we have $\nu_k, \tilde{\nu}_k \in \mathcal{P}_p(\mathbb{R}^d)$ for any $k$. Moreover, we have

$$x_k - \tilde{x}_k = M_k(x_{k-1} - \tilde{x}_{k-1}), \tag{C.23}$$

Due to spherical symmetry of the isotropic Gaussian distribution, the distribution of $\frac{\|M_k x\|}{\|x\|}$ does not depend on the choice of $x \in \mathbb{R}^d \backslash \{0\}$. Therefore, $\frac{\|M_k(x_{k-1} - \tilde{x}_{k-1})\|}{\|x_{k-1} - \tilde{x}_{k-1}\|}$ and $\|x_{k-1} - \tilde{x}_{k-1}\|$ are independent, and $\frac{\|M_k(x_{k-1} - \tilde{x}_{k-1})\|}{\|x_{k-1} - \tilde{x}_{k-1}\|}$ has the same distribution as $\|M_k e_1\|$, where $e_1$ is the first basis vector. It follows from (C.23) that

$$
\begin{aligned}
\mathbb{E}\|x_k - \tilde{x}_k\|^p &\leq \mathbb{E}\left[\|M_k(x_{k-1} - \tilde{x}_{k-1})\|^p\right] \\
&= \mathbb{E}\left[\|M_k e_1\|^p\right] \mathbb{E}\left[\|x_{k-1} - \tilde{x}_{k-1}\|^p\right] \\
&= h(p)\mathbb{E}\left[\|x_{k-1} - \tilde{x}_{k-1}\|^p\right],
\end{aligned}
$$

which by iterating implies that

$$
\mathcal{W}_p^p(\nu_k, \tilde{\nu}_k) \leq \mathbb{E}\|x_k - \tilde{x}_k\|^p \leq (h(p))^k \mathbb{E}\|x_0 - \tilde{x}_0\|^p = (h(p))^k \mathcal{W}_p^p(\nu_0, \tilde{\nu}_0). \tag{C.24}
$$

By letting $\tilde{\nu}_0 = \nu_\infty$, the probability law of the stationary distribution $x_\infty$, we conclude that

$$
\mathcal{W}_p(\nu_k, \nu_\infty) \leq \left((h(p))^{1/q}\right)^k \mathcal{W}_p(\nu_0, \nu_\infty). \tag{C.25}
$$

Finally, notice that $1 \leq p < \alpha$, and therefore $h(p) < 1$. The proof is complete. $\qquad\square$

*Proof of Corollary 7.* When $\eta\sigma^2 < \frac{2b}{d+b+1}$, by Proposition 3, the tail-index $\alpha > 2$, by taking $p = 2$, and using $h(2) = 1 - 2\eta\sigma^2 + \frac{\eta^2\sigma^4}{b}(d + b + 1) < 1$ (see Proposition 3), it follows from Theorem 6 that

$$
\mathcal{W}_2(\nu_k, \nu_\infty) \leq \left(1 - 2\eta\sigma^2\left(1 - \frac{\eta\sigma^2}{2b}(d + b + 1)\right)\right)^{k/2} \mathcal{W}_2(\nu_0, \nu_\infty). \tag{C.26}
$$

$\qquad\square$

**Remark 13.** *Consider the case $a_i$ are i.i.d. $\mathcal{N}(0, \sigma^2 I_d)$. In Theorem 4, Corollary 5 and Theorem 6, the key quantity is $h(p) \in (0, 1)$, where $p < \alpha$. We recall that*

$$
h(p) = \mathbb{E}\left[\left(1 - \frac{2a}{b}X + \frac{a^2}{b^2}X^2 + \frac{a^2}{b^2}XY\right)^{p/2}\right], \tag{C.27}
$$

*where $a = \eta\sigma^2$, $X, Y$ are independent chi-square random variables with degree of freedom $b$ and $d - 1$ respectively. The first-order approximation of $h(p)$ is given by*

$$
h(p) \sim 1 + \frac{p}{2}\mathbb{E}\left[-\frac{2a}{b}X + \frac{a^2}{b^2}X^2 + \frac{a^2}{b^2}XY\right] = 1 + \frac{p}{2}\left[-2a + \frac{a^2}{b}(b + 2) + \frac{a^2}{b}(d - 1)\right] < 1, \tag{C.28}
$$

*provided that $a = \eta\sigma^2 < \frac{2b}{d+b+1}$ which occurs if and only if $\alpha > 2$. In other words, when $\eta\sigma^2 < \frac{2b}{d+b+1}$, $\alpha > 2$ and*

$$
h(p) \sim 1 - p\eta\sigma^2\left(1 - \frac{\eta\sigma^2(b + d + 1)}{2b}\right) < 1. \tag{C.29}
$$

*On the other hand, when $\eta\sigma^2 \geq \frac{2b}{d+b+1}$, $p < \alpha \leq 2$, and the second-order approximation of $h(p)$ is given by*

$$
\begin{aligned}
h(p) &\sim 1 + \frac{p}{2}\mathbb{E}\left[-\frac{2a}{b}X + \frac{a^2}{b^2}X^2 + \frac{a^2}{b^2}XY\right] + \frac{\frac{p}{2}(\frac{p}{2} - 1)}{2}\mathbb{E}\left[\left(-\frac{2a}{b}X + \frac{a^2}{b^2}X^2 + \frac{a^2}{b^2}XY\right)^2\right] \\
&= 1 + qa\left(\frac{a(b + d + 1)}{2b} - 1\right) - \frac{2 - p}{8}\mathbb{E}\left[\left(-\frac{2a}{b}X + \frac{a^2}{b^2}X^2 + \frac{a^2}{b^2}XY\right)^2\right],
\end{aligned}
$$

*and we computed before in (E.55) that for small $a = \eta\sigma^2$ and large $d$,*

$$
\mathbb{E}\left[\left(-\frac{2a}{b}X + \frac{a^2}{b^2}X^2 + \frac{a^2}{b^2}XY\right)^2\right] \sim \frac{4a^2}{b}(b + 2) + \frac{a^4}{b^3}(b + 2)d^2 - \frac{4a^3}{b^2}(b + 2)d, \tag{C.30}
$$

and therefore with $a = \eta \sigma^2$,

$$h(p) \sim 1 - pa \left( \frac{-a(b+d+1)}{2b} + 1 + \frac{(2-p)a(b+2)}{2qb} \left( 1 + \frac{a^2}{4b^2} d^2 - \frac{a}{b} d \right) \right) < 1, \quad \text{(C.31)}$$

provided that $1 \leq \frac{a(b+d+1)}{2b} < 1 + \frac{(2-p)a(b+2)}{2qb} \left( 1 + \frac{a^2}{4b^2} d^2 - \frac{a}{b} d \right)$.

*Proof of Proposition 8.* First, we notice that it follows from Theorem 1 that $\mathbb{E}\|x_\infty\|^\alpha = \infty$. To see this, notice that $\lim_{t \to \infty} t^\alpha \mathbb{P}(e_1^T x_\infty > t) = e_\alpha(e_1)$, where $e_1$ is the first basis vector in $\mathbb{R}^d$, and $\mathbb{P}(\|x_\infty\| \geq t) \geq \mathbb{P}(e_1^T x_\infty \geq t)$, and thus

$$\mathbb{E}\|x_\infty\|^\alpha = \int_0^\infty t\mathbb{P}(\|x_\infty\|^\alpha \geq t)dt = \int_0^\infty t\mathbb{P}(\|x_\infty\| \geq t^{1/\alpha})dt = \infty. \quad \text{(C.32)}$$

By following the proof of Theorem 4 by letting $q = \alpha$ in the proof, one can show the following.

(i) If the tail-index $\alpha \leq 1$, then we have

$$\mathbb{E}\|x_\infty\|^\alpha \leq \mathbb{E}\|x_0\|^\alpha + k\mathbb{E}\|q_1\|^\alpha, \quad \text{(C.33)}$$

which grows linearly in $k$.

(ii) If the tail-index $\alpha > 1$, then for any $\epsilon > 0$, we have

$$\mathbb{E}\|x_k\|^\alpha \leq (1+\epsilon)^k \mathbb{E}\|x_0\|^\alpha + \frac{(1+\epsilon)^k - 1}{\epsilon} \frac{(1+\epsilon)^{\frac{\alpha}{\alpha-1}} - (1+\epsilon)}{\left( (1+\epsilon)^{\frac{1}{\alpha-1}} - 1 \right)^\alpha} \mathbb{E}\|q_1\|^\alpha = O(k), \quad \text{(C.34)}$$

which grows exponentially in $k$ for any fixed $\epsilon > 0$. By letting $\epsilon \to 0$, we have

$$\mathbb{E}\|x_k\|^\alpha = (1+\epsilon)^k \mathbb{E}\|x_0\|^\alpha + (1 + O(\epsilon))((1+\epsilon)^k - 1) \frac{(\alpha-1)^{\alpha-1}}{\epsilon^\alpha} \mathbb{E}\|q_1\|^\alpha.$$

Therefore, it holds for any sufficiently small $\epsilon > 0$ that,

$$\mathbb{E}\|x_k\|^\alpha \leq \frac{(1+\epsilon)^k}{\epsilon^\alpha} \left( \mathbb{E}\|x_0\|^\alpha + (\alpha-1)^{\alpha-1} \mathbb{E}\|q_1\|^\alpha \right).$$

We can optimize $\frac{(1+\epsilon)^k}{\epsilon^\alpha}$ over the choice of $\epsilon > 0$, and by choosing $\epsilon = \frac{\alpha}{k-\alpha}$, which goes to zero as $k$ goes to $\infty$, we have $\frac{(1+\epsilon)^k}{\epsilon^\alpha} = (1 + \frac{\alpha}{k-\alpha})^k (\frac{k-\alpha}{\alpha})^\alpha = O(k^\alpha)$, and hence

$$\mathbb{E}\|x_k\|^\alpha = O(k^\alpha), \quad \text{(C.35)}$$

which grows polynomially in $k$. The proof is complete. $\square$

*Proof of Corollary 9.* The result is obtained by a direct application of (Mirek, 2011, Theorem 1.15) to the recursions (3.3) where it can be checked in a straightforward manner that the conditions for this theorem hold. $\square$

# D SUPPORTING LEMMAS

In this section, we present a few supporting lemmas that are used in the proofs of the main results of the paper as well as the additional results in the Appendix.

First, we recall that the iterates are given by $x_k = M_k x_{k-1} + q_k$, where $(M_k, q_k)$ are i.i.d. and $M_k$ is distributed as $I - \frac{\eta}{b} H$, where $H = \sum_{i=1}^b a_i a_i^T$ and $q_k$ is distributed as $\frac{\eta}{b} \sum_{i=1}^b a_i y_i$, where $a_i$ and $y_i$ are i.i.d. satisfying the Assumptions **(A1)–(A2)**.

We can compute $\rho$ and $h(s)$ as follows where $\rho$ and $h(s)$ are defined by (3.5) and (3.4).

**Lemma 14.** *$\rho$ can be characterized as:*

$$\rho = \mathbb{E} \log \left\| \left( I - \frac{\eta}{b} H \right) e_1 \right\|, \quad \text{(D.1)}$$

and $h(s)$ can be characterized as:

$$h(s) = \mathbb{E} \left[ \left\| \left( I - \frac{\eta}{b} H \right) e_1 \right\|^s \right], \quad \text{(D.2)}$$

provided that $\rho < 0$.

*Proof.* It is known that the Lyapunov exponent defined in (3.5) admits the alternative representation

$$\rho := \lim_{k \to \infty} \frac{1}{k} \log \|\tilde{x}_k\|, \tag{D.3}$$

where $\tilde{x}_k := \Pi_k \tilde{x}_0$ with $\Pi_k := M_k M_{k-1} \dots M_1$ and $\tilde{x}_0 := x_0$ (see (Newman, 1986, eqn. (2))). We will compute the limit on the right-hand side of (D.3). First, we observe that due to spherical symmetry of the isotropic Gaussian distribution, the distribution of $\frac{\|M_k x\|}{\|x\|}$ does not depend on the choice of $x \in \mathbb{R}^d \backslash \{0\}$ and is i.i.d. over $k$ with the expectation $\mathbb{E}(\|Me_1\|) = \mathbb{E}(\|(I - \frac{\eta}{b} H) e_1\|)$ where we chose $x = e_1$. Therefore,

$$\frac{1}{k} \log \|\tilde{x}_k\| - \frac{1}{k} \log \|\tilde{x}_0\| = \frac{1}{k} \sum_{i=1}^{k} \log \frac{\|\tilde{x}_i\|}{\|\tilde{x}_{i-1}\|} = \frac{1}{k} \sum_{i=1}^{k} \log \frac{\|M_i \tilde{x}_{i-1}\|}{\|\tilde{x}_{i-1}\|}$$

is an average of i.i.d. random variables and by the law of large numbers we obtain

$$\rho = \lim_{k \to \infty} \frac{1}{k} \log \|\tilde{x}_k\| = \mathbb{E} \left\| \left( I - \frac{\eta}{b} H \right) e_1 \right\|.$$

From (D.3), we conclude that this proves (D.1). It remains to prove (D.2). We consider the function

$$\tilde{h}(s) = \lim_{k \to \infty} \left( \mathbb{E} \frac{\|\tilde{x}_k\|^s}{\|\tilde{x}_0\|^s} \right)^{1/k},$$

where the initial point $\tilde{x}_0 = x_0$ is deterministic. In the rest of the proof, we will show that for $\rho < 0$, $h(s) = \tilde{h}(s)$ where $h(s)$ is given by (3.4) and $\tilde{h}(s)$ is equal to the right-hand side of (D.2); our proof is inspired by the approach of Newman (1986). We will first compute $\tilde{h}(s)$ and show that it is equal to the right-hand side of (D.2). Note that we can write

$$\frac{\|x_k\|^s}{\|x_0\|^s} = \prod_{i=1}^{k} \frac{\|M_i x_{i-1}\|^s}{\|x_{i-1}\|^s}.$$

This is a product of i.i.d. random variables with the same distribution as that of $\|Me_1\|^s$ due to the spherical symmetry of the input $a_i$. Therefore, we can write

$$\tilde{h}(s) = \lim_{k \to \infty} \left( \mathbb{E} \frac{\|x_k\|^s}{\|x_0\|^s} \right)^{1/k} = \lim_{k \to \infty} \left( \mathbb{E} \prod_{i=1}^{k} \|M_i e_1\|^s \right)^{1/k}$$

$$= \mathbb{E} \left[ \|Me_1\|^s \right] = \mathbb{E} \left[ \left\| \left( I - \frac{\eta}{b} H \right) e_1 \right\|^s \right], \tag{D.4}$$

where we used the fact that $M_i e_1$ are i.i.d. over $i$. It remains to show that $h(s) = \tilde{h}(s)$ for $\rho < 0$. Note that $\frac{\|\tilde{x}_k\|^s}{\|\tilde{x}_0\|^s} \leq \|\Pi_k\|^s$, and therefore from the definition of $h(s)$ and $\tilde{h}(s)$, we have immediately

$$h(s) \geq \tilde{h}(s) \tag{D.5}$$

for any $s > 0$. We will show that $h(s) \leq \tilde{h}(s)$ when $\rho < 0$. We assume $\rho < 0$. Then, Theorem 1 is applicable and there exists a stationary distribution $x_\infty$ with a tail index $\alpha$ such that $h(\alpha) = 1$. We will show that $\tilde{h}(\alpha) = 1$. First, the tail density admits the characterization (3.7), and therefore $x_\infty \in L_s$ for $s < \alpha$, i.e. the $s$-th moment of $x_\infty$ is finite. Similarly due to (3.7), $x_\infty \notin L_s$ for $s > \alpha$. Since $h(\alpha) = 1$, it follows from (D.5) that we have $\tilde{h}(\alpha) \leq 1$. However if $\tilde{h}(\alpha) < 1$, then by the continuity of the $\tilde{h}$ function there exists $\varepsilon$ such that $h(s) < 1$ for every $s \in (\alpha - \varepsilon, \alpha + \varepsilon) \subset (0, 1)$. From the definition of $\tilde{h}(s)$ then this would imply that $\mathbb{E}(\|x_k\|^s) \to 0$ for every $s \in (\alpha - \varepsilon, \alpha + \varepsilon)$. On the other hand, by following a similar argument to the proof technique of Corollary 5, it can be shown that the $s$-th moment of $x_\infty$ has to be bounded,[8] which would be a contradiction with the fact that $x_\infty \notin L_s$ for $s > \alpha$. Therefore, $\tilde{h}(\alpha) \geq 1$. Since $h(\alpha) = 1$, (D.5) leads to

$$h(\alpha) = \tilde{h}(\alpha) = 1. \tag{D.6}$$

---

[8] Note that the proof of Corollary 5 establishes first that $x_\infty$ has a bounded $s$-th moment provided that $\tilde{h}(s) = \mathbb{E} \left[ \|Me_1\|^s \right] < 1$ and then cites Lemma 14 regarding the equivalence $h(s) = \tilde{h}(s)$.

We observe that the function $h$ is homogeneous in the sense that if the iterations matrices $M_i$ are replaced by $cM_i$ where $c > 0$ is a real scalar, $h(s)$ will be replaced by $h_c(s) := c^s h(s)$. In other words, the function

$$h_c(s) := \lim_{k \to \infty} \left( \mathbb{E} \| (cM_k)(cM_{k-1}) \ldots (cM_1) \|^s \right)^{1/k} \tag{D.7}$$

clearly satisfies $h_c(s) = c^s h(s)$ by definition. A similar homogeneity property holds for $\tilde{h}(s)$: If the iterations matrices $M_i$ are replaced by $cM_i$, then $\tilde{h}(s)$ will be replaced by $\tilde{h}_c(s) := c^s \tilde{h}(s)$. We will show that this homogeneity property combined with the fact that $h(\alpha) = \tilde{h}(\alpha) = 1$ will force $h(s) = \tilde{h}(s)$ for any $s > 0$. For this purpose, given $s > 0$, we choose $c = 1/\sqrt[s]{h(s)}$. Then, by considering input matrix $cM_i$ instead of $M_i$ and by following a similar argument which led to the identity (D.6), we can show that $h_c(s) = c^s h(s) = 1$. Therefore, $\tilde{h}_c(s) = \tilde{h}_c(s) = 1$. This implies directly $\tilde{h}(s) = h(s)$.

$\qquad\qquad\qquad\qquad\qquad\qquad\qquad\qquad\qquad\qquad\qquad\qquad\qquad\qquad\qquad\qquad\square$

Next, we show the following property for the function $h$.

**Lemma 15.** *We have $h(0) = 1$, $h'(0) = \rho$ and $h(s)$ is strictly convex in $s$.*

*Proof.* By the expression of $h(s)$ from Lemma 14, it is easy to check that $h(0) = 1$. Moreover, we can compute that

$$h'(s) = \mathbb{E}\left[ \log \left( \left\| \left( I - \frac{\eta}{b} H \right) e_1 \right\| \right) \left\| \left( I - \frac{\eta}{b} H \right) e_1 \right\|^s \right], \tag{D.8}$$

and thus

$$h'(0) = \rho. \tag{D.9}$$

Moreover, we can compute that

$$h''(s) = \mathbb{E}\left[ \left( \log \left( \left\| \left( I - \frac{\eta}{b} H \right) e_1 \right\| \right) \right)^2 \left\| \left( I - \frac{\eta}{b} H \right) e_1 \right\|^s \right] > 0, \tag{D.10}$$

which implies that $h(s)$ is strictly convex in $s$. $\qquad\qquad\qquad\qquad\qquad\qquad\qquad\square$

In the next result, we show that $\liminf_{s \to \infty} h(s) > 1$. This property, together with Lemma 15 implies that if $\rho < 0$, then there exists some $\alpha \in (0, \infty)$ such that $h(\alpha) = 1$. Indeed, in the proof of Lemma 16, we will show that $\liminf_{s \to \infty} h(s) = \infty$.

**Lemma 16.** *We have $\liminf_{s \to \infty} h(s) > 1$.*

*Proof.* We recall from Lemma 14 that

$$h(s) = \mathbb{E} \left\| \left( I - \frac{\eta}{b} H \right) e_1 \right\|^s, \tag{D.11}$$

where $e_1$ is the first basis vector in $\mathbb{R}^d$ and $H = \sum_{i=1}^{b} a_i a_i^T$, and $a_i = (a_{i1}, \ldots, a_{id})$ are i.i.d. distributed as $\mathcal{N}(0, \sigma^2 I_d)$. We can compute that

$$\mathbb{E}\left\|\left(I - \frac{\eta}{b}H\right)e_1\right\|^s = \mathbb{E}\left(\left\|\left(I - \frac{\eta}{b}H\right)e_1\right\|^2\right)^{s/2}$$

$$= \mathbb{E}\left[\left(e_1^T\left(I - \frac{\eta}{b}\sum_{i=1}^{b}a_i a_i^T\right)\left(I - \frac{\eta}{b}\sum_{i=1}^{b}a_i a_i^T\right)e_1\right)^{s/2}\right]$$

$$= \mathbb{E}\left[\left(1 - \frac{2\eta}{b}e_1^T\sum_{i=1}^{b}a_i a_i^T e_1 + \frac{\eta^2}{b^2}e_1^T\sum_{i=1}^{b}a_i a_i^T\sum_{i=1}^{b}a_i a_i^T e_1\right)^{s/2}\right]$$

$$= \mathbb{E}\left[\left(1 - \frac{2\eta}{b}\sum_{i=1}^{b}a_{i1}^2 + \frac{\eta^2}{b^2}\sum_{i=1}^{b}\sum_{j=1}^{b}(a_{i1}a_{j1} + \cdots + a_{id}a_{jd})a_{i1}a_{j1}\right)^{s/2}\right]$$

$$= \mathbb{E}\left[\left(\left(1 - \frac{\eta}{b}\sum_{i=1}^{b}a_{i1}^2\right)^2 + \frac{\eta^2}{b^2}\sum_{i=1}^{b}\sum_{j=1}^{b}(a_{i2}a_{j2} + \cdots + a_{id}a_{jd})a_{i1}a_{j1}\right)^{s/2}\right]$$

$$\geq \mathbb{E}\left[2^{s/2}\mathbf{1}_{\frac{\eta^2}{b^2}\sum_{i=1}^{b}\sum_{j=1}^{b}(a_{i2}a_{j2}+\cdots+a_{id}a_{jd})a_{i1}a_{j1}\geq 2}\right]$$

$$= 2^{s/2}\mathbb{P}\left(\frac{\eta^2}{b^2}\sum_{i=1}^{b}\sum_{j=1}^{b}(a_{i2}a_{j2} + \cdots + a_{id}a_{jd})a_{i1}a_{j1} \geq 2\right) \to \infty,$$

as $s \to \infty$. $\qquad\square$

In the next result, we show that the inverse of $M$ exists with probability 1, and provide an upper bound result, which will be used to prove Lemma 18.

**Lemma 17.** $M^{-1}$ *exists with probability* 1. *Moreover, we have*

$$\mathbb{E}\left[\left(\log^+\|M^{-1}\|\right)^2\right] \leq 8.$$

*Proof.* Note that $M$ is a continuous random matrix, by the assumption on the distribution of $a_i$. Therefore,

$$\mathbb{P}(M^{-1} \text{ does not exist}) = \mathbb{P}(\det M = 0) = 0. \tag{D.12}$$

Note that the singular values of $M^{-1}$ are of the form $|1 - \frac{\eta}{b}\sigma_H|^{-1}$ where $\sigma_H$ is a singular value of $H$ and we have

$$(\log^+\|M^{-1}\|)^2 = \begin{cases} 0 & \text{if } \frac{\eta}{b}H \succ 2I, \\ (\|(I - \frac{\eta}{b}H)^{-1}\|)^2 & \text{if } 0 \preceq \frac{\eta}{b}H \preceq 2I. \end{cases} \tag{D.13}$$

We consider two cases $0 \preceq \frac{\eta}{b}H \preceq I$ and $I \preceq \frac{\eta}{b}H \preceq 2I$. We compute the conditional expectations for each case:

$$\mathbb{E}\left[\left(\log^+\|M^{-1}\|\right)^2 \mid 0 \preceq \frac{\eta}{b}H \preceq I\right] = \mathbb{E}\left[\left(\log\left\|\left(I - \frac{\eta}{b}H\right)^{-1}\right\|\right)^2 \mid 0 \preceq \frac{\eta}{b}H \prec I\right] \tag{D.14}$$

$$\leq \mathbb{E}\left[\left(2\frac{\eta}{b}\|H\|\right)^2 \mid 0 \preceq \frac{\eta}{b}H \preceq I\right] \tag{D.15}$$

$$\leq 4, \tag{D.16}$$

where in the first inequality we used the fact that

$$\log(I - X)^{-1} \preceq 2X \tag{D.17}$$

for a symmetric positive semi-definite matrix $X$ satisfying $0 \preceq X \prec I$ (the proof of this fact is analogous to the proof of the scalar inequality $\log(\frac{1}{1-x}) \le 2x$ for $0 \le x < 1$). By a similar computation,

$$\mathbb{E}\left[(\log^+\|M^{-1}\|)^2 \mid I \preceq \frac{\eta}{b}H \preceq 2I\right]$$

$$= \mathbb{E}\left[\log\left\|\left(I - \frac{\eta}{b}H\right)^{-1}\right\| \mid I \preceq \frac{\eta}{b}H \prec 2I\right]$$

$$= \mathbb{E}\left[\log^2\left\|\left(\frac{\eta}{b}H\right)^{-1}\left[I - \left(\frac{\eta}{b}H\right)^{-1}\right]^{-1}\right\| \mid I \preceq \frac{\eta}{b}H \prec 2I\right]$$

$$\le \mathbb{E}\left[\log^2\left(\left\|\left(\frac{\eta}{b}H\right)^{-1}\right\| \cdot \left\|\left[I - \left(\frac{\eta}{b}H\right)^{-1}\right]^{-1}\right\|\right) \mid I \preceq \frac{\eta}{b}H \prec 2I\right]$$

$$\le \mathbb{E}\left[\log^2\left(\left\|\left[I - \left(\frac{\eta}{b}H\right)^{-1}\right]^{-1}\right\|\right) \mid I \preceq \frac{\eta}{b}H \prec 2I\right]$$

$$= \mathbb{E}\left[\log^2\left(\left\|\left[I - \left(\frac{\eta}{b}H\right)^{-1}\right]^{-1}\right\|\right) \mid \frac{1}{2}I \preceq \left(\frac{\eta}{b}H\right)^{-1} \prec I\right],$$

where in the last inequality we used the fact that $(\frac{\eta}{b}H)^{-1} \preceq I$ for $I \preceq \frac{\eta}{b}H \prec 2I$. If we apply the inequality (D.17) to the last inequality for the choice of $X = (\frac{\eta}{b}H)^{-1}$, we obtain

$$\mathbb{E}\left[\log^2\left\|\left[I - \left(\frac{\eta}{b}H\right)^{-1}\right]^{-1}\right\| \mid \frac{1}{2}I \preceq \left(\frac{\eta}{b}H\right)^{-1} \prec I\right]$$

$$\le \mathbb{E}\left[\left\|2\left(\frac{\eta}{b}H\right)^{-1}\right\|^2 \mid \frac{1}{2}I \preceq \left(\frac{\eta}{b}H\right)^{-1} \prec I\right] \le 4. \tag{D.18}$$

Combining (D.16) and (D.18), it follows from (D.13) that $\mathbb{E}\log^+\|M^{-1}\| \le 8$. $\qquad\square$

In the next result, we show that a certain expected value that involves the moments and logarithm of $\|M\|$, and logarithm of $\|M^{-1}\|$ is finite, which is used in the proof of Theorem 1.

**Lemma 18.**
$$\mathbb{E}\left[\|M\|^\alpha\left(\log^+\|M\| + \log^+\|M^{-1}\|\right)\right] < \infty.$$

*Proof.* Note that $M = I - \frac{\eta}{b}H$, where $H = \sum_i^b a_i a_i^T$ in distribution. Therefore for any $s > 0$,

$$\mathbb{E}[\|M\|^s] = \mathbb{E}\left[\left\|I - \frac{\eta}{b}\sum_{i=1}^b a_i a_i^T\right\|^s\right] \le \mathbb{E}\left[\left(1 + \frac{\eta}{b}\sum_{i=1}^b \|a_i\|^2\right)^s\right] < \infty, \tag{D.19}$$

since all the moments of $a_i$ are finite by the Assumption **(A1)**. This implies that

$$\mathbb{E}\left[\|M\|^\alpha\left(\log^+\|M\|\right)\right] < \infty.$$

By Cauchy-Schwarz inequality,

$$\mathbb{E}\left[\|M\|^\alpha\left(\log^+\|M^{-1}\|\right)\right] \le \left(\mathbb{E}\left[\|M\|^{2\alpha}\right]\mathbb{E}\left[\left(\log^+\|M^{-1}\|\right)^2\right]\right)^{1/2} < \infty,$$

where we used Lemma 17. $\qquad\square$

In the next result, we show a convexity result, which is used in the proof of Theorem 2 to show that the tail-index $\alpha$ is strictly decreasing in stepsize $\eta$ and variance $\sigma^2$.

**Lemma 19.** *For any given positive semi-definite symmetric matrix $H$ fixed, the function $F_H : [0, \infty) \to \mathbb{R}$ defined as*
$$F_H(a) := \|(I - aH)e_1\|^s$$

is convex for $s \geq 1$. It follows that for given $b$ and $d$ with $\tilde{H} := \frac{1}{b} \sum_{i=1}^{b} a_i a_i^T$, the function

$$h(a, s) := \mathbb{E}\left[F_{\tilde{H}}(a)\right] = \mathbb{E}\left\|\left(I - a\tilde{H}\right) e_1\right\|^s \tag{D.20}$$

is a convex function of $a$ for a fixed $s \geq 1$.

*Proof.* We consider the case $s \geq 1$ and consider the function

$$G_H(a) := \|(I - aH) e_1\|,$$

and show that it is convex for $H \succeq 0$ and it is strongly convex for $H \succ 0$ over the interval $[0, \infty)$. Let $a_1, a_2 \in [0, \infty)$ be different points, i.e. $a_1 \neq a_2$. It follows from the subadditivity of the norm that

$$\begin{aligned}
G_H\left(\frac{a_1 + a_2}{2}\right) &= \left\|\left(I - \frac{a_1 + a_2}{2}H\right) e_1\right\| \\
&\leq \left\|\left(\frac{I}{2} - \frac{a_1}{2}H\right) e_1\right\| + \left\|\left(\frac{I}{2} - \frac{a_2}{2}H\right) e_1\right\| \\
&= \frac{1}{2}G_H(a_1) + \frac{1}{2}G_H(a_2),
\end{aligned}$$

which implies that $G_H(a)$ is a convex function. On the other hand, the function $g(x) = x^s$ is convex for $s \geq 1$ on the positive real axis, therefore the composition $g(G_H(a))$ is also convex for any $H$ fixed. Since the expectation of random convex functions is also convex, we conclude that $h(s)$ is also convex. □

The next result is used in the proof of Theorem 4 to bound the moments of the iterates.

**Lemma 20.** *(i) Given $0 < p \leq 1$, for any $x, y \geq 0$,*

$$(x + y)^p \leq x^p + y^p. \tag{D.21}$$

*(ii) Given $p > 1$, for any $x, y \geq 0$, and any $\epsilon > 0$,*

$$(x + y)^p \leq (1 + \epsilon)x^p + \frac{(1 + \epsilon)^{\frac{p}{p-1}} - (1 + \epsilon)}{\left((1 + \epsilon)^{\frac{1}{p-1}} - 1\right)^p} y^p. \tag{D.22}$$

*Proof.* (i) If $y = 0$, then $(x + y)^p \leq x^p + y^p$ trivially holds. If $y > 0$, it is equivalent to show that

$$\left(\frac{x}{y} + 1\right)^p \leq \left(\frac{x}{y}\right)^p + 1, \tag{D.23}$$

which is equivalent to show that

$$(x + 1)^p \leq x^p + 1, \qquad \text{for any } x \geq 0. \tag{D.24}$$

Let $F(x) := (x + 1)^p - x^p - 1$ and $F(0) = 0$ and $F'(x) = p(x + 1)^{p-1} - px^{p-1} \leq 0$ since $p \leq 1$, which shows that $F(x) \leq 0$ for every $x \geq 0$.

(ii) If $y = 0$, then the inequality trivially holds. If $y > 0$, by doing the transform $x \mapsto x/y$ and $y \mapsto 1$, it is equivalent to show that for any $x \geq 0$,

$$(1 + x)^p \leq (1 + \epsilon)x^p + \frac{(1 + \epsilon)^{\frac{p}{p-1}} - (1 + \epsilon)}{\left((1 + \epsilon)^{\frac{1}{p-1}} - 1\right)^p}. \tag{D.25}$$

To show this, we define

$$F(x) := (1 + x)^p - (1 + \epsilon)x^p, \qquad x \geq 0. \tag{D.26}$$

Then $F'(x) = p(1 + x)^{p-1} - p(1 + \epsilon)x^{p-1}$ so that $F'(x) \geq 0$ if $x \leq ((1 + \epsilon)^{\frac{1}{p-1}} - 1)^{-1}$, and $F'(x) \leq 0$ if $x \geq ((1 + \epsilon)^{\frac{1}{p-1}} - 1)^{-1}$. Thus,

$$\max_{x \geq 0} F(x) = F\left(\frac{1}{(1 + \epsilon)^{\frac{1}{p-1}} - 1}\right) = \frac{(1 + \epsilon)^{\frac{p}{p-1}} - (1 + \epsilon)}{\left((1 + \epsilon)^{\frac{1}{p-1}} - 1\right)^p}. \tag{D.27}$$

The proof is complete. □

# E   ADDITIONAL TECHNICAL RESULTS

We recall that the iterates are given by $x_k = M_k x_{k-1} + q_k$, where $(M_k, q_k)$ are i.i.d. copies of $(M, q)$ where $M = I - \frac{\eta}{b} H$ with $H = \sum_{i=1}^{b} a_i a_i^T$ in distribution and $q = \frac{\eta}{b} \sum_{i=1}^{b} a_i y_i$, where $a_i$ and $y_i$ are i.i.d. satisfying the Assumptions **(A1)–(A2)**.

We first obtain more explicit expressions for $\rho$ and $h(s)$ under the Assumption **(A1)**.

**Proposition 21.** *We have*

$$\rho = \frac{1}{2} \mathbb{E} \log \left[ 1 - \frac{2\eta\sigma^2}{b} \sum_{i=1}^{b} z_{i1}^2 + \frac{\eta^2\sigma^4}{b^2} \sum_{i=1}^{b} \sum_{j=1}^{b} (z_{i1}z_{j1} + \cdots + z_{id}z_{jd}) z_{i1}z_{j1} \right], \quad \text{(E.1)}$$

*and for any $s \geq 0$,*

$$h(s) = \mathbb{E} \left[ \left( 1 - \frac{2\eta\sigma^2}{b} \sum_{i=1}^{b} z_{i1}^2 + \frac{\eta^2\sigma^4}{b^2} \sum_{i=1}^{b} \sum_{j=1}^{b} (z_{i1}z_{j1} + \cdots + z_{id}z_{jd}) z_{i1}z_{j1} \right)^{s/2} \right], \quad \text{(E.2)}$$

*where $z_i := (z_{i1}, z_{i2}, \ldots, z_{id}) \sim \mathcal{N}(0, I_d)$, $1 \leq i \leq b$ are i.i.d.*

*Proof.* By the expression of $\rho$ from Lemma 14, we can compute that

$$\rho = \mathbb{E} \log \left\| \left( I - \frac{\eta}{b} H \right) e_1 \right\|$$

$$= \frac{1}{2} \mathbb{E} \log \left\| \left( I - \frac{\eta}{b} H \right) e_1 \right\|^2$$

$$= \frac{1}{2} \mathbb{E} \log \left[ e_1^T \left( I - \frac{\eta}{b} \sum_{i=1}^{b} a_i a_i^T \right) \left( I - \frac{\eta}{b} \sum_{i=1}^{b} a_i a_i^T \right) e_1 \right]$$

$$= \frac{1}{2} \mathbb{E} \log \left[ 1 - \frac{2\eta}{b} e_1^T \sum_{i=1}^{b} a_i a_i^T e_1 + \frac{\eta^2}{b^2} e_1^T \sum_{i=1}^{b} a_i a_i^T \sum_{i=1}^{b} a_i a_i^T e_1 \right]$$

$$= \frac{1}{2} \mathbb{E} \log \left[ 1 - \frac{2\eta\sigma^2}{b} \sum_{i=1}^{b} z_{i1}^2 + \frac{\eta^2\sigma^4}{b^2} \sum_{i=1}^{b} \sum_{j=1}^{b} (z_{i1}z_{j1} + \cdots + z_{id}z_{jd}) z_{i1}z_{j1} \right],$$

where $z_i = (z_{i1}, z_{i2}, \ldots, z_{id}) \sim \mathcal{N}(0, I_d)$, $1 \leq i \leq b$ are i.i.d. Similarly, by the expression of $h(s)$ from Lemma 14, we have

$$h(s) = \mathbb{E} \left\| \left( I - \frac{\eta}{b} H \right) e_1 \right\|^s$$

$$= \mathbb{E} \left[ \left( 1 - \frac{2\eta\sigma^2}{b} \sum_{i=1}^{b} z_{i1}^2 + \frac{\eta^2\sigma^4}{b^2} \sum_{i=1}^{b} \sum_{j=1}^{b} (z_{i1}z_{j1} + \cdots + z_{id}z_{jd}) z_{i1}z_{j1} \right)^{s/2} \right].$$

$\square$

**Remark 22.** *It follows from Proposition 21 that $\rho$, $h(s)$ and hence the tail-index $\alpha$ depends on $\eta$ and $\sigma^2$ only via its product $\eta\sigma^2$.*

We have seen in Theorem 1 that the iterates converge to a heavy-tailed distribution with tail-index $\alpha \in (0, \infty)$ provided that $\rho < 0$. When the data $a_i$ are i.i.d. in general, it is not easy to check whether $\rho < 0$ holds. For the Gaussian data (Assumption **(A1)**), it is possible to characterise the region of the parameters $\eta$, $b$, $d$, $\sigma^2$ in which $\rho < 0$. We first state the following result, which provides a sufficient (but not necessary) condition for $\rho < 0$.

**Proposition 23.** *$\rho < 0$ provided that*

$$\eta\sigma^2(d + b + 1) < 2b. \quad \text{(E.3)}$$

*Proof.* We recall from Proposition 21 that

$$\rho = \frac{1}{2}\mathbb{E}\log\left[1 - \frac{2\eta\sigma^2}{b}\sum_{i=1}^{b}z_{i1}^2 + \frac{\eta^2\sigma^4}{b^2}\sum_{i=1}^{b}\sum_{j=1}^{b}(z_{i1}z_{j1} + \cdots + z_{id}z_{jd})z_{i1}z_{j1}\right], \quad (E.4)$$

where $z_i = (z_{i1}, \ldots, z_{id}) \sim \mathcal{N}(0, I_d)$, $1 \le i \le b$. Note that the function $x \mapsto \log x$ is concave, and by Jensen's inequality, we have

$$\rho \le \frac{1}{2}\log\mathbb{E}\left[1 - \frac{2\eta\sigma^2}{b}\sum_{i=1}^{b}z_{i1}^2 + \frac{\eta^2\sigma^4}{b^2}\sum_{i=1}^{b}\sum_{j=1}^{b}(z_{i1}z_{j1} + \cdots + z_{id}z_{jd})z_{i1}z_{j1}\right]. \quad (E.5)$$

We can compute that

$$\mathbb{E}\left[\sum_{i=1}^{b}z_{i1}^2\right] = b\mathbb{E}[z_{11}^2] = b, \quad (E.6)$$

and

$$\mathbb{E}\left[\sum_{i=1}^{b}\sum_{j=1}^{b}(z_{i1}z_{j1} + \cdots + z_{id}z_{jd})z_{i1}z_{j1}\right]$$

$$= \mathbb{E}\left[\sum_{i=1}^{b}(z_{i1}^2 + \cdots + z_{id}^2)z_{i1}^2\right] + \mathbb{E}\left[\sum_{1 \le i \ne j \le b}(z_{i1}z_{j1} + \cdots + z_{id}z_{jd})z_{i1}z_{j1}\right]$$

$$= b\mathbb{E}\left[(z_{11}^2 + \cdots + z_{1d}^2)z_{11}^2\right] + b(b-1)\mathbb{E}\left[(z_{11}z_{21} + \cdots + z_{1d}z_{2d})z_{11}z_{21}\right]$$

$$= b\mathbb{E}[z_{11}^4] + b(d-1)\left(\mathbb{E}[z_{11}^2]\right)^2 + b(b-1)\mathbb{E}[z_{11}^2 z_{21}^2]$$

$$= 3b + b(d-1) + b(b-1) = b(d + b + 1),$$

where we used the property that $z_i = (z_{ij}, 1 \le j \le d)$ are i.i.d. and $z_{ij}$ are i.i.d. $N(0, 1)$ and

$$\mathbb{E}[z_{11}^2] = 1, \qquad \mathbb{E}[z_{11}^4] = 3. \quad (E.7)$$

Hence, we conclude that $\rho < 0$ provided that

$$1 - 2\eta\sigma^2 + \frac{\eta^2\sigma^4}{b}(d + b + 1) < 1, \quad (E.8)$$

which is equivalent to

$$\eta\sigma^2(d + b + 1) < 2b. \quad (E.9)$$

The proof is complete. $\qquad\square$

**Remark 24.** *It is worth pointing out that $\rho < 0$ does not hold for arbitrary model parameters. In particular, we can compute that*

$$\rho = \frac{1}{2}\mathbb{E}\log\left[\left(1 - \frac{\eta\sigma^2}{b}\sum_{i=1}^{b}z_{i1}^2\right)^2 + \frac{\eta^2\sigma^4}{b^2}\sum_{i=1}^{b}\sum_{j=1}^{b}z_{i1}z_{j1}\sum_{k=2}^{d}z_{ik}z_{jk}\right]$$

$$= \frac{1}{2}\mathbb{E}\log\left[\left(1 - \frac{\eta\sigma^2}{b}\sum_{i=1}^{b}z_{i1}^2\right)^2 + \frac{\eta^2\sigma^4}{b^2}\sum_{k=2}^{d}\left(\sum_{i=1}^{b}z_{i1}z_{ik}\right)^2\right]$$

$$\ge \frac{1}{2}\mathbb{E}\log\left[\frac{\eta^2\sigma^4}{b^2}\sum_{k=2}^{d}\left(\sum_{i=1}^{b}z_{i1}z_{ik}\right)^2\right]$$

$$= \log\left(\frac{\eta\sigma^2}{b}\right) + \frac{1}{2}\mathbb{E}\log\left[\sum_{k=2}^{d}\left(\sum_{i=1}^{b}z_{i1}z_{ik}\right)^2\right].$$

*Note that conditional on $z_{i1}$, $1 \leq i \leq b$,*

$$\sum_{i=1}^{b} z_{i1} z_{ik} \sim \mathcal{N}\left(0, \sum_{i=1}^{b} z_{i1}^2\right), \tag{E.10}$$

*are i.i.d. for $k = 2, \ldots, d$. Therefore,*

$$\frac{1}{2}\mathbb{E}\log\left[\sum_{k=2}^{d}\left(\sum_{i=1}^{b} z_{i1} z_{ik}\right)^2\right] = \frac{1}{2}\mathbb{E}\log[XY] = \frac{1}{2}\mathbb{E}\log X + \frac{1}{2}\log\mathbb{E}\log Y, \tag{E.11}$$

*where $X$ is a chi-square random variable with degree of freedom $b$ and $Y$ is a chi-square random variable with degree of freedom $(d-1)$. Therefore, we can compute that*

$$\frac{1}{2}\mathbb{E}\log X = \frac{1}{2}\int_0^{\infty}\log(x)\frac{x^{\frac{b}{2}-1}e^{-\frac{x}{2}}}{2^{\frac{b}{2}}\Gamma(\frac{b}{2})}dx$$

$$\geq \frac{1}{2^{\frac{b}{2}+1}\Gamma(\frac{b}{2})}\int_0^1 \log(x)x^{\frac{b}{2}-1}e^{-\frac{x}{2}}dx$$

$$\geq \frac{1}{2^{\frac{b}{2}+1}\Gamma(\frac{b}{2})}\int_0^1 \log(x)x^{\frac{b}{2}-1}dx = \frac{-1}{2^{\frac{b}{2}-1}b^2\Gamma(\frac{b}{2})}.$$

*Similarly, we can show that $\frac{1}{2}\mathbb{E}\log Y \geq \frac{-1}{2^{\frac{d-1}{2}-1}(d-1)^2\Gamma(\frac{d-1}{2})}$. Hence, we conclude that $\rho \geq 0$ provided that*

$$\eta\sigma^2 \geq b\exp\left\{\frac{1}{2^{\frac{b}{2}-1}b^2\Gamma(\frac{b}{2})} + \frac{1}{2^{\frac{d-1}{2}-1}(d-1)^2\Gamma(\frac{d-1}{2})}\right\}. \tag{E.12}$$

Next, we provide alternative formulas for $h(s)$ and $\rho$ for the Gaussian data (Assumption **(A1)**) which is used for some technical proofs.

**Lemma 25.** *For any $s > 0$,*

$$h(s) = \mathbb{E}\left[\left(\left(1 - \frac{\eta\sigma^2}{b}X\right)^2 + \frac{\eta^2\sigma^4}{b^2}XY\right)^{s/2}\right],$$

*and*

$$\rho = \frac{1}{2}\mathbb{E}\left[\log\left(\left(1 - \frac{\eta\sigma^2}{b}X\right)^2 + \frac{\eta^2\sigma^4}{b^2}XY\right)\right],$$

*where $X, Y$ are independent and $X$ is chi-square random variable with degree of freedom $b$ and $Y$ is a chi-square random variable with degree of freedom $(d-1)$.*

*Proof.* We can compute that

$$h(s) = \mathbb{E}\left[\left(1 - \frac{2\eta\sigma^2}{b}\sum_{i=1}^{b} z_{i1}^2 + \frac{\eta^2\sigma^4}{b^2}\sum_{i=1}^{b}\sum_{j=1}^{b}(z_{i1}z_{j1} + \cdots + z_{id}z_{jd})z_{i1}z_{j1}\right)^{s/2}\right]$$

$$= \mathbb{E}\left[\left(1 - \frac{2\eta\sigma^2}{b}\sum_{i=1}^{b} z_{i1}^2 + \frac{\eta^2\sigma^4}{b^2}\sum_{i=1}^{b}\sum_{j=1}^{b}\left(z_{i1}^2 z_{j1}^2 + z_{i1}z_{j1}\sum_{k=2}^{d} z_{ik}z_{jk}\right)\right)^{s/2}\right]$$

$$= \mathbb{E}\left[\left(1 - \frac{2\eta\sigma^2}{b}\sum_{i=1}^{b} z_{i1}^2 + \frac{\eta^2\sigma^4}{b^2}\left(\sum_{i=1}^{b} z_{i1}^2\right)^2 + \frac{\eta^2\sigma^4}{b^2}\sum_{k=2}^{d}\left(\sum_{i=1}^{b} z_{i1}z_{ik}\right)^2\right)^{s/2}\right]$$

$$= \mathbb{E}\left[\left(\left(1 - \frac{\eta\sigma^2}{b}\sum_{i=1}^{b} z_{i1}^2\right)^2 + \frac{\eta^2\sigma^4}{b^2}\sum_{k=2}^{d}\left(\sum_{i=1}^{b} z_{i1}z_{ik}\right)^2\right)^{s/2}\right].$$

Note that conditional on $z_{i1}$, $1 \leq i \leq b$,

$$\sum_{i=1}^{b} z_{i1} z_{ik} \sim \mathcal{N}\left(0, \sum_{i=1}^{b} z_{i1}^2\right), \tag{E.13}$$

are i.i.d. for $k = 2, \ldots, d$. Therefore, we have

$$h(s) = \mathbb{E}\left[\left(\left(1 - \frac{\eta\sigma^2}{b}\sum_{i=1}^{b} z_{i1}^2\right)^2 + \frac{\eta^2\sigma^4}{b^2}\sum_{k=2}^{d}\left(\sum_{i=1}^{b} z_{i1} z_{ik}\right)^2\right)^{s/2}\right]$$

$$= \mathbb{E}\left[\left(\left(1 - \frac{\eta\sigma^2}{b}\sum_{i=1}^{b} z_{i1}^2\right)^2 + \frac{\eta^2\sigma^4}{b^2}\sum_{i=1}^{b} z_{i1}^2 \sum_{k=2}^{d} x_k^2\right)^{s/2}\right],$$

where $x_k$ are i.i.d. $N(0,1)$ independent of $z_{i1}$, $i = 1, \ldots, b$. Hence, we have

$$h(s) = \mathbb{E}\left[\left(\left(1 - \frac{\eta\sigma^2}{b}\sum_{i=1}^{b} z_{i1}^2\right)^2 + \frac{\eta^2\sigma^4}{b^2}\sum_{i=1}^{b} z_{i1}^2 \sum_{k=2}^{d} x_k^2\right)^{s/2}\right]$$

$$= \mathbb{E}\left[\left(\left(1 - \frac{\eta\sigma^2}{b}X\right)^2 + \frac{\eta^2\sigma^4}{b^2}XY\right)^{s/2}\right],$$

where $X, Y$ are independent and $X$ is chi-square random variable with degree of freedom $b$ and $Y$ is a chi-square random variable with degree of freedom $(d-1)$.

Similarly, we can compute that

$$\rho = \frac{1}{2}\mathbb{E}\left[\log\left[\left(1 - \frac{\eta\sigma^2}{b}\sum_{i=1}^{b} z_{i1}^2\right)^2 + \frac{\eta^2\sigma^4}{b^2}\sum_{i=1}^{b}\sum_{j=1}^{b} z_{i1} z_{j1} \sum_{k=2}^{d} z_{ik} z_{jk}\right]\right]$$

$$= \frac{1}{2}\mathbb{E}\left[\log\left[\left(1 - \frac{\eta\sigma^2}{b}\sum_{i=1}^{b} z_{i1}^2\right)^2 + \frac{\eta^2\sigma^4}{b^2}\sum_{k=2}^{d}\left(\sum_{i=1}^{b} z_{i1} z_{ik}\right)^2\right]\right]$$

$$= \frac{1}{2}\mathbb{E}\left[\log\left(\left(1 - \frac{\eta\sigma^2}{b}X\right)^2 + \frac{\eta^2\sigma^4}{b^2}XY\right)\right],$$

where $X, Y$ are independent and $X$ is chi-square random variable with degree of freedom $b$ and $Y$ is a chi-square random variable with degree of freedom $(d-1)$. The proof is complete. $\qquad\square$

In Theorem 1, we showed the existence of the tail-index $\alpha$. For the Gaussian data (Assumption (**A1**)), we can provide some explicit bound on the tail-index $\alpha$ provided some explicit technical conditions hold. Next, we provide a technical condition under which the tail-index $\alpha \in (0, 4]$. (See also Proposition 3 for a technical condition under which the tail-index $\alpha \in (0, 2]$.)

**Proposition 26.** *There exists some $0 < \alpha \leq 4$ such that $h(\alpha) = 1$ provided that*

$$2b > \eta\sigma^2(d + b + 1)$$

$$\geq 2b - 2\eta\sigma^2(b+2) + \frac{2\eta^2\sigma^4}{b}(b+2)(b+d+3)$$

$$- \frac{1}{2}\frac{\eta^3\sigma^6}{b^2}(b+2)(b+4)(b+6) - \frac{1}{2}\frac{\eta^3\sigma^6}{b^2}(b+2)(d^2-1) - \frac{\eta^3\sigma^6}{b^2}(b+2)(b+4)(d-1).$$

*Proof.* It follows from Lemma 25 that

$$h(4) = \mathbb{E}\left[\left(\left(1 - \frac{\eta\sigma^2}{b}X\right)^2 + \frac{\eta^2\sigma^4}{b^2}XY\right)^2\right],$$

where $X, Y$ are independent and $X$ is chi-square random variable with degree of freedom $b$ and $Y$ is a chi-square random variable with degree of freedom $(d-1)$. We can further compute that

$$h(4) = \mathbb{E}\left[\left(1 - \frac{\eta\sigma^2}{b}X\right)^4\right] + \mathbb{E}\left[\frac{\eta^4\sigma^8}{b^4}X^2Y^2\right] + 2\mathbb{E}\left[\left(1 - \frac{\eta\sigma^2}{b}X\right)^2\frac{\eta^2\sigma^4}{b^2}XY\right]. \quad \text{(E.14)}$$

First, we can compute that

$$\mathbb{E}\left[\left(1 - \frac{\eta\sigma^2}{b}X\right)^4\right] = 1 - \frac{4\eta\sigma^2}{b}\mathbb{E}[X] + \frac{6\eta^2\sigma^4}{b^2}\mathbb{E}[X^2] - \frac{4\eta^3\sigma^6}{b^3}\mathbb{E}[X^3] + \frac{\eta^4\sigma^8}{b^4}\mathbb{E}[X^4]. \quad \text{(E.15)}$$

We recall the formula for the $m$-th moment of a chi-square distribution with degree of freedom $k$ given by $2^m\Gamma(m + \frac{k}{2})/\Gamma(\frac{k}{2})$, which is $2^m\frac{k}{2}\left(\frac{k}{2}+1\right)\cdots\left(\frac{k}{2}+(m-1)\right)$ when $m$ is a positive integer. Since $X$ is chi-square distributed with degree of freedom $b$, we have

$$\mathbb{E}[X] = 2 \cdot \frac{b}{2} = b,$$

$$\mathbb{E}[X^2] = 2^2 \cdot \frac{b}{2}\left(\frac{b}{2}+1\right) = b(b+2),$$

$$\mathbb{E}[X^3] = 2^3 \cdot \frac{b}{2}\left(\frac{b}{2}+1\right)\left(\frac{b}{2}+2\right) = b(b+2)(b+4),$$

$$\mathbb{E}[X^4] = 2^4 \cdot \frac{b}{2}\left(\frac{b}{2}+1\right)\left(\frac{b}{2}+2\right)\left(\frac{b}{2}+3\right) = b(b+2)(b+4)(b+6),$$

which implies that

$$\mathbb{E}\left[\left(1 - \frac{\eta\sigma^2}{b}X\right)^4\right] = 1 - 4\eta\sigma^2 + \frac{6\eta^2\sigma^4}{b}(b+2) - \frac{4\eta^3\sigma^6}{b^2}(b+2)(b+4) + \frac{\eta^4\sigma^8}{b^3}(b+2)(b+4)(b+6). \quad \text{(E.16)}$$

Second, we can compute that

$$\mathbb{E}\left[\frac{\eta^4\sigma^8}{b^4}X^2Y^2\right] = \frac{\eta^4\sigma^8}{b^4}\mathbb{E}\left[X^2\right]\mathbb{E}[Y^2] = \frac{\eta^4\sigma^8}{b^4}b(b+2)(d-1)(d+1) = \frac{\eta^4\sigma^8}{b^3}(b+2)(d^2-1), \quad \text{(E.17)}$$

where we used the fact that $Y$ is independent of $X$ and $Y$ is chi-square distributed with degree of freedom $d-1$ so that $\mathbb{E}[Y^2] = (d-1)(d+1)$.

Third, we can compute that

$$2\mathbb{E}\left[\left(1 - \frac{\eta\sigma^2}{b}X\right)^2\frac{\eta^2\sigma^4}{b^2}XY\right]$$

$$= 2\mathbb{E}\left[\frac{\eta^2\sigma^4}{b^2}XY\right] + 2\mathbb{E}\left[\frac{\eta^4\sigma^8}{b^4}X^3Y\right] - 4\mathbb{E}\left[\frac{\eta^3\sigma^6}{b^3}X^2Y\right]$$

$$= 2\frac{\eta^2\sigma^4}{b}(d-1) + 2\frac{\eta^4\sigma^8}{b^3}(b+2)(b+4)(d-1) - 4\frac{\eta^3\sigma^6}{b^2}(b+2)(d-1).$$

Putting everything together, we have

$$h(4) = 1 - 4\eta\sigma^2 + \frac{6\eta^2\sigma^4}{b}(b+2) - \frac{4\eta^3\sigma^6}{b^2}(b+2)(b+4) + \frac{\eta^4\sigma^8}{b^3}(b+2)(b+4)(b+6)$$

$$+ \frac{\eta^4\sigma^8}{b^3}(b+2)(d^2-1)$$

$$+ 2\frac{\eta^2\sigma^4}{b}(d-1) + 2\frac{\eta^4\sigma^8}{b^3}(b+2)(b+4)(d-1) - 4\frac{\eta^3\sigma^6}{b^2}(b+2)(d-1),$$

and $h(4) \geq 1$ if and only if

$$1 - 4\eta\sigma^2 + \frac{6\eta^2\sigma^4}{b}(b+2) - \frac{4\eta^3\sigma^6}{b^2}(b+2)(b+4) + \frac{\eta^4\sigma^8}{b^3}(b+2)(b+4)(b+6)$$
$$+ \frac{\eta^4\sigma^8}{b^3}(b+2)(d^2-1)$$
$$+ 2\frac{\eta^2\sigma^4}{b}(d-1) + 2\frac{\eta^4\sigma^8}{b^3}(b+2)(b+4)(d-1) - 4\frac{\eta^3\sigma^6}{b^2}(b+2)(d-1) \geq 1,$$

which is equivalent to

$$- 2b + \eta\sigma^2(3b+6) + \eta\sigma^2(d-1)$$
$$- \frac{2\eta^2\sigma^4}{b}(b+2)(b+4) + \frac{1}{2}\frac{\eta^3\sigma^6}{b^2}(b+2)(b+4)(b+6) + \frac{1}{2}\frac{\eta^3\sigma^6}{b^2}(b+2)(d^2-1)$$
$$+ \frac{\eta^3\sigma^6}{b^2}(b+2)(b+4)(d-1) - 2\frac{\eta^2\sigma^4}{b}(b+2)(d-1) \geq 0,$$

which is equivalent to

$$- 2b + \eta\sigma^2(d+b+1) \geq -2\eta\sigma^2(b+2)$$
$$+ \frac{2\eta^2\sigma^4}{b}(b+2)(b+4) - \frac{1}{2}\frac{\eta^3\sigma^6}{b^2}(b+2)(b+4)(b+6) - \frac{1}{2}\frac{\eta^3\sigma^6}{b^2}(b+2)(d^2-1)$$
$$- \frac{\eta^3\sigma^6}{b^2}(b+2)(b+4)(d-1) + 2\frac{\eta^2\sigma^4}{b}(b+2)(d-1).$$

Finally, recall that $\rho < 0$ if $2b > \eta\sigma^2(d+b+1)$ and $h(4) \geq 1$ implies there exists some $0 < \alpha \leq 4$ such that $h(\alpha) = 1$. The proof is complete. $\qquad\square$

Let us define $\mathcal{D}_{s,b}$ as the set consisting of $(\eta, \sigma)$ such that

$$\mathcal{D}_{s,b} := \left\{ (\eta, \sigma) : \eta\sigma^2(d+b+1) < 2b \text{ and } h(s) \geq 1 \right\}.$$

We have shown in Proposition 23 that $\rho < 0$ provided that $\eta\sigma^2(d+b+1) < 2b$. Therefore, if $(\eta, \sigma) \in \mathcal{D}_{s,b}$, then the tail-index $\alpha \in (0, s]$. In Proposition 26, we characterised the set $\mathcal{D}_{4,b}$. In the next proposition, we show that $\mathcal{D}_{4,b}$ is non-trivial, i.e., $\mathcal{D}_{4,b}$ is not an empty set.

**Proposition 27.** *$\mathcal{D}_{4,b}$ is not an empty set. In particular, it includes the pairs $(\eta, \sigma)$ such that*

$$\frac{a_c}{d+b+1} \leq \frac{\eta\sigma^2}{b} < \frac{2}{d+b+1}, \tag{E.18}$$

*for some $a_c \in (0, 2)$ such that $F(a_c) = 0$ and $F(a) \leq 0$ for any $a_c \leq a \leq 2$, where*

$$F(a) := 2 - a - \frac{2a}{d+b+1}(b+2) + \frac{2a^2}{(d+b+1)^2}(b+2)(b+d+3)$$
$$- \frac{1}{2}\frac{a^3}{(d+b+1)^3}(b+2)(b+4)(b+6) - \frac{1}{2}\frac{a^3}{(d+b+1)^3}(b+2)(d^2-1)$$
$$- \frac{a^3}{(d+b+1)^3}(b+2)(b+4)(d-1).$$

*Proof.* We aim to show that there exist some $\eta, \sigma$ such that

$$2 > \frac{\eta\sigma^2}{b}(d+b+1)$$
$$\geq 2 - \frac{2\eta\sigma^2}{b}(b+2) + \frac{2\eta^2\sigma^4}{b^2}(b+2)(b+d+3)$$
$$- \frac{1}{2}\frac{\eta^3\sigma^6}{b^3}(b+2)(b+4)(b+6) - \frac{1}{2}\frac{\eta^3\sigma^6}{b^3}(b+2)(d^2-1) - \frac{\eta^3\sigma^6}{b^3}(b+2)(b+4)(d-1).$$

Let $\frac{\eta\sigma^2}{b}(d+b+1) = a$. Then, it suffices to show that there exists some $a < 2$ such that

$$a \geq 2 - \frac{2a}{d+b+1}(b+2) + \frac{2a^2}{(d+b+1)^2}(b+2)(b+d+3)$$
$$- \frac{1}{2}\frac{a^3}{(d+b+1)^3}(b+2)(b+4)(b+6) - \frac{1}{2}\frac{a^3}{(d+b+1)^3}(b+2)(d^2-1)$$
$$- \frac{a^3}{(d+b+1)^3}(b+2)(b+4)(d-1).$$

Let us define

$$F(a) := 2 - a - \frac{2a}{d+b+1}(b+2) + \frac{2a^2}{(d+b+1)^2}(b+2)(b+d+3)$$
$$- \frac{1}{2}\frac{a^3}{(d+b+1)^3}(b+2)(b+4)(b+6) - \frac{1}{2}\frac{a^3}{(d+b+1)^3}(b+2)(d^2-1)$$
$$- \frac{a^3}{(d+b+1)^3}(b+2)(b+4)(d-1).$$

Then, we can check that $F(0) = 2 > 0$ and

$$F(2) = -\frac{4}{d+b+1}(b+2) + \frac{8}{(d+b+1)^2}(b+2)(b+d+3)$$
$$- \frac{4}{(d+b+1)^3}(b+2)(b+4)(b+6) - \frac{4}{(d+b+1)^3}(b+2)(d^2-1)$$
$$- \frac{8}{(d+b+1)^3}(b+2)(b+4)(d-1),$$

so that

$$\frac{(d+b+1)^3}{4(b+2)}F(2) = -(d+b+1)^2 + 2(d+b+1)(b+d+3)$$
$$- (b+4)(b+6) - (d^2-1) - 2(b+4)(d-1)$$
$$= d^2 + b^2 + 1 + 2d + 2b + 2bd + 4d + 4b + 4$$
$$- b^2 - 10b - 24 - d^2 + 1 - 2bd - 8d + 2b + 8$$
$$= -2d - 2b - 10 < 0.$$

Thus, $F(2) < 0$. Hence, we conclude that there exists some $0 < a_c < 2$ such that $F(a_c) = 0$ and $F(a) \leq 0$ for any $a_c \leq a \leq 2$. Then, for any

$$a_c \leq \frac{\eta\sigma^2}{b}(d+b+1) < 2, \tag{E.19}$$

we have $(\eta, \sigma) \in \mathcal{D}_{4,b}$. The proof is complete. $\qquad\square$

Recall that $\mathcal{D}_{2m,b}$ consists of $(\eta, \sigma)$ such that

$$\mathcal{D}_{2m,b} = \left\{ (\eta, \sigma) : \eta\sigma^2(d+b+1) < 2b \text{ and } h(2m) \geq 1 \right\}.$$

Since $\mathcal{D}_{4,b} \neq \emptyset$ and $\mathcal{D}_{4,b} \subseteq \mathcal{D}_{2m,b}$ for any $m \geq 2$. Indeed, we can characterises the set $\mathcal{D}_{2m,b}$ in the following proposition.

**Proposition 28.** *Given any $m \in \mathbb{N}$, there exists some $0 < \alpha \leq 2m$ such that $h(\alpha) = 1$ provided that $\eta\sigma^2(d+b+1) < 2b$ and*

$$\sum_{k=0}^{m}\sum_{j=0}^{2k}\binom{m}{k}\binom{2k}{j}\frac{\eta^{2(m-k)+j}\sigma^{4(m-k)+2j}}{b^{2(m-k)+j}}(-1)^j 2^{j+2(m-k)}$$
$$\cdot \frac{\Gamma(j+m-k+\frac{b}{2})}{\Gamma(\frac{b}{2})}\frac{\Gamma(m-k+\frac{d-1}{2})}{\Gamma(\frac{d-1}{2})} \geq 1.$$

*Proof.* By applying Lemma 25, we can compute that

$$
h(2m) = \mathbb{E}\left[\left(\left(1 - \frac{\eta\sigma^2}{b}X\right)^2 + \frac{\eta^2\sigma^4}{b^2}XY\right)^m\right]
$$

$$
= \sum_{k=0}^{m}\binom{m}{k}\frac{\eta^{2(m-k)}\sigma^{4(m-k)}}{b^{2(m-k)}}\mathbb{E}\left[\left(1 - \frac{\eta\sigma^2}{b}X\right)^{2k}X^{m-k}Y^{m-k}\right]
$$

$$
= \sum_{k=0}^{m}\binom{m}{k}\frac{\eta^{2(m-k)}\sigma^{4(m-k)}}{b^{2(m-k)}}\sum_{j=0}^{2k}\binom{2k}{j}(-1)^j\frac{\eta^j\sigma^{2j}}{b^j}\mathbb{E}[X^{j+m-k}Y^{m-k}]
$$

$$
= \sum_{k=0}^{m}\binom{m}{k}\frac{\eta^{2(m-k)}\sigma^{4(m-k)}}{b^{2(m-k)}}\sum_{j=0}^{2k}\binom{2k}{j}(-1)^j\frac{\eta^j\sigma^{2j}}{b^j}2^{j+m-k}
$$

$$
\cdot \frac{\Gamma(j+m-k+\frac{b}{2})}{\Gamma(\frac{b}{2})}2^{m-k}\frac{\Gamma(m-k+\frac{d-1}{2})}{\Gamma(\frac{d-1}{2})}
$$

$$
= \sum_{k=0}^{m}\sum_{j=0}^{2k}\binom{m}{k}\binom{2k}{j}\frac{\eta^{2(m-k)+j}\sigma^{4(m-k)+2j}}{b^{2(m-k)+j}}
$$

$$
\cdot (-1)^j 2^{j+2(m-k)}\frac{\Gamma(j+m-k+\frac{b}{2})}{\Gamma(\frac{b}{2})}\frac{\Gamma(m-k+\frac{d-1}{2})}{\Gamma(\frac{d-1}{2})},
$$

where we used the formula for the moments of chi-square distribution, that is, the $m$-th moment of a chi-square distribution with degree of freedom $k$ is given by $2^m\Gamma(m+\frac{k}{2})/\Gamma(\frac{k}{2})$. □

Previously, we have provided some upper bounds on the tail-index $\alpha$ under some technical conditions on the model parameters. Next, let us provide an upper bound for the tail-index $\alpha$ without relying on any additional technical conditions.

**Proposition 29.** *The tail-index $\alpha$ is upper bounded by:*

$$
\alpha \leq \max\left\{2, \frac{\mathbb{P}\left(-\frac{2\eta\sigma^2}{b}X + \frac{\eta^2\sigma^4}{b^2}X^2 + \frac{\eta^2\sigma^4}{b^2}XY \leq 0\right)}{\mathbb{E}\left[\left(-\frac{2\eta\sigma^2}{b}X + \frac{\eta^2\sigma^4}{b^2}X^2 + \frac{\eta^2\sigma^4}{b^2}XY\right)^+\right]}\right\}, \tag{E.20}
$$

*where $X, Y$ are independent and $X$ is a chi-square random variable with degree of freedom $b$ and $Y$ is a chi-square random variable with degree of freedom $(d-1)$.*

*Proof.* We recall that

$$
1 = h(\alpha) = \mathbb{E}\left[\left(1 - \frac{2\eta\sigma^2}{b}X + \frac{\eta^2\sigma^4}{b^2}X^2 + \frac{\eta^2\sigma^4}{b^2}XY\right)^{\frac{\alpha}{2}}\right], \tag{E.21}
$$

where $X, Y$ are independent and $X$ is a chi-square random variable with degree of freedom $b$ and $Y$ is a chi-square random variable with degree of freedom $(d-1)$. Note that for any $x \geq 0$ and $\alpha \geq 2$, $(1+x)^{\frac{\alpha}{2}} \geq 1 + \frac{\alpha}{2}x$. Therefore,

$$
1 \geq \mathbb{E}\left[\left(1 - \frac{2\eta\sigma^2}{b}X + \frac{\eta^2\sigma^4}{b^2}X^2 + \frac{\eta^2\sigma^4}{b^2}XY\right)^{\frac{\alpha}{2}}1_{-\frac{2\eta\sigma^2}{b}X+\frac{\eta^2\sigma^4}{b^2}X^2+\frac{\eta^2\sigma^4}{b^2}XY\geq 0}\right]
$$

$$
\geq \mathbb{E}\left[\left(1 + \frac{\alpha}{2}\left(-\frac{2\eta\sigma^2}{b}X + \frac{\eta^2\sigma^4}{b^2}X^2 + \frac{\eta^2\sigma^4}{b^2}XY\right)\right)1_{-\frac{2\eta\sigma^2}{b}X+\frac{\eta^2\sigma^4}{b^2}X^2+\frac{\eta^2\sigma^4}{b^2}XY\geq 0}\right]
$$

$$
= \mathbb{P}\left(-\frac{2\eta\sigma^2}{b}X + \frac{\eta^2\sigma^4}{b^2}X^2 + \frac{\eta^2\sigma^4}{b^2}XY \geq 0\right)
$$

$$
+ \frac{\alpha}{2}\mathbb{E}\left[\left(-\frac{2\eta\sigma^2}{b}X + \frac{\eta^2\sigma^4}{b^2}X^2 + \frac{\eta^2\sigma^4}{b^2}XY\right)^+\right],
$$

which yields the desired result. The proof is complete. □

**Remark 30.** *(i) Note that it follows from Lemma 25 that we have*

$$\rho = \frac{1}{2}\mathbb{E}\left[\log\left(\left(1 - \frac{\eta\sigma^2}{b}X\right)^2 + \frac{\eta^2\sigma^4}{b^2}XY\right)\right],$$

*where $X, Y$ are independent and $X$ is chi-square random variable with degree of freedom $b$ and $Y$ is a chi-square random variable with degree of freedom $(d-1)$. Therefore, we have*

$$\rho = \frac{1}{2}\int_0^\infty\int_0^\infty \log\left(\left(1 - \frac{\eta\sigma^2}{b}x\right)^2 + \frac{\eta^2\sigma^4}{b^2}xy\right)\frac{x^{\frac{b}{2}-1}e^{-\frac{x}{2}}}{2^{\frac{b}{2}}\Gamma(\frac{b}{2})}\frac{y^{\frac{d-1}{2}-1}e^{-\frac{y}{2}}}{2^{\frac{d-1}{2}}\Gamma(\frac{d-1}{2})}dxdy. \qquad \text{(E.22)}$$

*In particular, when $d = 1$, we have $Y \equiv 0$ and*

$$\rho = \int_0^\infty \log\left|1 - \frac{\eta\sigma^2}{b}x\right|\frac{x^{\frac{b}{2}-1}e^{-\frac{x}{2}}}{2^{\frac{b}{2}}\Gamma(\frac{b}{2})}dx. \qquad \text{(E.23)}$$

*(ii) Note that it follows from Lemma 25 that we have*

$$h(s) = \mathbb{E}\left[\left(\left(1 - \frac{\eta\sigma^2}{b}X\right)^2 + \frac{\eta^2\sigma^4}{b^2}XY\right)^{s/2}\right],$$

*where $X, Y$ are independent and $X$ is chi-square random variable with degree of freedom $b$ and $Y$ is a chi-square random variable with degree of freedom $(d-1)$. Therefore, we have*

$$h(s) = \int_0^\infty\int_0^\infty\left(\left(1 - \frac{\eta\sigma^2}{b}x\right)^2 + \frac{\eta^2\sigma^4}{b^2}xy\right)^{\frac{s}{2}}\frac{x^{\frac{b}{2}-1}e^{-\frac{x}{2}}}{2^{\frac{b}{2}}\Gamma(\frac{b}{2})}\frac{y^{\frac{d-1}{2}-1}e^{-\frac{y}{2}}}{2^{\frac{d-1}{2}}\Gamma(\frac{d-1}{2})}dxdy.$$

*In particular, when $d = 1$, we have $Y \equiv 0$ and*

$$h(s) = \int_0^\infty\left|1 - \frac{\eta\sigma^2}{b}x\right|^s\frac{x^{\frac{b}{2}-1}e^{-\frac{x}{2}}}{2^{\frac{b}{2}}\Gamma(\frac{b}{2})}dx. \qquad \text{(E.24)}$$

So far, we have studied various properties of the tail-index $\alpha$, including the monotonicity on stepsize, noise variance, batch size and the dimension, as well as some quantitative bounds. In general, there is no simple closed-form formula for the tail-index $\alpha$. Next, we will obtain some approximations for the tail-index $\alpha$ in various asymptotic regimes. First, we provide a rigorous first-order approximation for the tail-index $\alpha$ when it is less than and close to 2.

**Proposition 31.** *Let $a := \eta\sigma^2$ and $a_c := \frac{2b}{d+b+1}$. Then the tail-index satisfies:*

$$\alpha \sim 2 - \frac{4}{F_c}(a - a_c), \qquad \text{(E.25)}$$

*for any $a \downarrow a_c$, where*

$$F_c := \mathbb{E}\left[\left(1 - \frac{2a_c}{b}X + \frac{a_c^2}{b^2}X^2 + \frac{a_c^2}{b^2}XY\right)\log\left(1 - \frac{2a_c}{b}X + \frac{a_c^2}{b^2}X^2 + \frac{a_c^2}{b^2}XY\right)\right] > 0, \quad \text{(E.26)}$$

*where $X, Y$ are independent and $X$ is a chi-square random variable with degree of freedom $b$ and $Y$ is a chi-square random variable with degree of freedom $(d-1)$.*

*Proof of Proposition 31.* Let us define $a = \eta\sigma^2$. In Proposition 3, we showed that there exists some $\delta > 0$ such that for any $2b \le a(d+b+1) < 2b + \delta$, $\rho < 0$ and there exists some $0 < \alpha \le 2$, such that $h(\alpha) = 1$. In particular, when $a = a_c := \frac{2b}{d+b+1}$, the tail-index $\alpha = 2$. Consider the tail-index $\alpha = \alpha(a)$ as a function of $a$. Then, we have

$$1 = h(\alpha) = \mathbb{E}\left[\left(1 - \frac{2a}{b}X + \frac{a^2}{b^2}X^2 + \frac{a^2}{b^2}XY\right)^{\frac{\alpha}{2}}\right], \qquad \text{(E.27)}$$

where $X, Y$ are independent and $X$ is a chi-square random variable with degree of freedom $b$ and $Y$ is a chi-square random variable with degree of freedom $(d-1)$. By differentiating (E.27) w.r.t. $a$, we get

$$0 = \mathbb{E}\left[\frac{1}{2}\frac{\partial\alpha}{\partial a}\log\left(1 - \frac{2a}{b}X + \frac{a^2}{b^2}X^2 + \frac{a^2}{b^2}XY\right)e^{\frac{\alpha}{2}\log\left(1 - \frac{2a}{b}X + \frac{a^2}{b^2}X^2 + \frac{a^2}{b^2}XY\right)}\right.$$
$$\left. + \frac{\alpha}{2}\frac{-\frac{2}{b}X + \frac{2a}{b^2}X^2 + \frac{2a}{b^2}XY}{1 - \frac{2a}{b}X + \frac{a^2}{b^2}X^2 + \frac{a^2}{b^2}XY}\cdot e^{\frac{\alpha}{2}\log\left(1 - \frac{2a}{b}X + \frac{a^2}{b^2}X^2 + \frac{a^2}{b^2}XY\right)}\right],$$

which implies that

$$\left.\frac{\partial\alpha}{\partial a}\right|_{a=a_c} = \frac{-2\mathbb{E}\left[-\frac{2}{b}X + \frac{2a_c}{b^2}X^2 + \frac{2a_c}{b^2}XY\right]}{\mathbb{E}\left[\left(1 - \frac{2a_c}{b}X + \frac{a_c^2}{b^2}X^2 + \frac{a_c^2}{b^2}XY\right)\log\left(1 - \frac{2a_c}{b}X + \frac{a_c^2}{b^2}X^2 + \frac{a_c^2}{b^2}XY\right)\right]}, \quad \text{(E.28)}$$

where we used the fact that $\alpha = 2$ when $a = a_c$. Note that the function $x \mapsto x\log x$ is convex, and by Jensen's inequality and the fact that $1 - \frac{2a_c}{b}X + \frac{a_c^2}{b^2}X^2 + \frac{a_c^2}{b^2}XY$ is not constant, we have

$$\mathbb{E}\left[\left(1 - \frac{2a_c}{b}X + \frac{a_c^2}{b^2}X^2 + \frac{a_c^2}{b^2}XY\right)\log\left(1 - \frac{2a_c}{b}X + \frac{a_c^2}{b^2}X^2 + \frac{a_c^2}{b^2}XY\right)\right]$$
$$> \mathbb{E}\left[\left(1 - \frac{2a_c}{b}X + \frac{a_c^2}{b^2}X^2 + \frac{a_c^2}{b^2}XY\right)\right]\log\mathbb{E}\left[\left(1 - \frac{2a_c}{b}X + \frac{a_c^2}{b^2}X^2 + \frac{a_c^2}{b^2}XY\right)\right] = 0.$$

Moreover, we can compute that

$$-2\mathbb{E}\left[-\frac{2}{b}X + \frac{2a_c}{b^2}X^2 + \frac{2a_c}{b^2}XY\right] = -2\left[-2 + \frac{2a_c}{b}(b+2) + \frac{2a_c}{b}(d-1)\right] = -4.$$

Hence, we conclude that

$$\alpha \sim 2 - \frac{4}{F_c}(a - a_c), \quad \text{(E.29)}$$

for any $a \downarrow a_c$, where

$$F_c := \mathbb{E}\left[\left(1 - \frac{2a_c}{b}X + \frac{a_c^2}{b^2}X^2 + \frac{a_c^2}{b^2}XY\right)\log\left(1 - \frac{2a_c}{b}X + \frac{a_c^2}{b^2}X^2 + \frac{a_c^2}{b^2}XY\right)\right] > 0. \quad \text{(E.30)}$$

$\square$

Next, we derive an approximation for the tail-index $\alpha$ when it is close to zero.

**Proposition 32.** *Let $a := \eta\sigma^2$ and $\rho = \rho(a)$ emphasizing the dependence on $a$. Define $a_* := \inf\{a > 0 : \rho(a) = 0\}$. Then, we have*

$$\alpha \sim c_*(a_* - a), \quad \text{(E.31)}$$

*as $a \uparrow a_*$, where*

$$c_* := 4\mathbb{E}\left[\frac{-\frac{2}{b}X + \frac{2a_*}{b^2}X^2 + \frac{2a_*}{b^2}XY}{1 - \frac{2a_*}{b}X + \frac{a_*^2}{b^2}X^2 + \frac{a_*^2}{b^2}XY}\right]\cdot\left(\mathbb{E}\left[\left(\log\left(1 - \frac{2a_*}{b}X + \frac{a_*^2}{b^2}X^2 + \frac{a_*^2}{b^2}XY\right)\right)^2\right]\right)^{-1}, \quad \text{(E.32)}$$

*where $X, Y$ are defined in Proposition 31.*

*Proof of Proposition 32.* The tail-index $\alpha$ is uniquely determined by

$$1 = h(\alpha) = \mathbb{E}\left[\left(1 - \frac{2a}{b}X + \frac{a^2}{b^2}X^2 + \frac{a^2}{b^2}XY\right)^{\alpha/2}\right], \quad \text{(E.33)}$$

where $a = \eta\sigma^2$ and $X, Y$ are independent and $X$ is chi-square random variable with degree of freedom $b$ and $Y$ is a chi-square random variable with degree of freedom $(d-1)$. It is clear that $\alpha$ depends on $\eta$ and $\sigma$ only via $a := \eta\sigma^2$. In Proposition 3, we showed that there exists some $\delta > 0$

such that for any $2b \leq a(d+b+1) < 2b+\delta$, $\rho < 0$ and there exists some $0 < \alpha \leq 2$, such that $h(\alpha) = 1$. Let $\rho = \rho(a)$ with emphasis on the dependence of $\rho$ on $a = \eta\sigma^2$. In Proposition 23, we showed that $\rho < 0$ provided that $a(d+b+1) < 2b$. On the other hand, we showed in Remark 24 that $\rho \geq 0$ for any $a \geq b\exp\left\{ \frac{1}{2^{\frac{b}{2}-1}b^2\Gamma(\frac{b}{2})} + \frac{1}{2^{\frac{d-1}{2}-1}(d-1)^2\Gamma(\frac{d-1}{2})} \right\}$. Therefore, there exists some critical value $a_* > 0$ such that $\rho(a) < 0$ for every $a < a_*$ and $\rho(a_*) = 0$. It is clear that as $a \to a_*$, $\alpha \to 0$. We are interested in studying the tail-index $\alpha$ when $\alpha$ is close to zero. By differentiating (E.33) w.r.t. $a$, we get

$$
0 = \mathbb{E}\Bigg[ \Bigg( \frac{1}{2}\frac{\partial\alpha}{\partial a} \log\left( 1 - \frac{2a}{b}X + \frac{a^2}{b^2}X^2 + \frac{a^2}{b^2}XY \right)
$$
$$
+ \frac{\alpha}{2} \frac{-\frac{2}{b}X + \frac{2a}{b^2}X^2 + \frac{2a}{b^2}XY}{1 - \frac{2a}{b}X + \frac{a^2}{b^2}X^2 + \frac{a^2}{b^2}XY} \Bigg) e^{\frac{\alpha}{2}\log\left(1-\frac{2a}{b}X+\frac{a^2}{b^2}X^2+\frac{a^2}{b^2}XY\right)} \Bigg],
$$

and by differentiating w.r.t. $a$ again, we get

$$
0 = \mathbb{E}\Bigg[ \Bigg( \frac{1}{2}\frac{\partial\alpha}{\partial a} \log\left( 1 - \frac{2a}{b}X + \frac{a^2}{b^2}X^2 + \frac{a^2}{b^2}XY \right)
$$
$$
+ \frac{\alpha}{2} \frac{-\frac{2}{b}X + \frac{2a}{b^2}X^2 + \frac{2a}{b^2}XY}{1 - \frac{2a}{b}X + \frac{a^2}{b^2}X^2 + \frac{a^2}{b^2}XY} \Bigg)^2 e^{\frac{\alpha}{2}\log\left(1-\frac{2a}{b}X+\frac{a^2}{b^2}X^2+\frac{a^2}{b^2}XY\right)} \Bigg]
$$
$$
+ \mathbb{E}\Bigg[ \frac{1}{2}\frac{\partial^2\alpha}{\partial a^2}\log\left(1 - \frac{2a}{b}X + \frac{a^2}{b^2}X^2 + \frac{a^2}{b^2}XY\right) e^{\frac{\alpha}{2}\log\left(1-\frac{2a}{b}X+\frac{a^2}{b^2}X^2+\frac{a^2}{b^2}XY\right)} \Bigg]
$$
$$
+ \mathbb{E}\Bigg[ \frac{\partial\alpha}{\partial a} \frac{-\frac{2}{b}X + \frac{2a}{b^2}X^2 + \frac{2a}{b^2}XY}{1 - \frac{2a}{b}X + \frac{a^2}{b^2}X^2 + \frac{a^2}{b^2}XY} \cdot e^{\frac{\alpha}{2}\log\left(1-\frac{2a}{b}X+\frac{a^2}{b^2}X^2+\frac{a^2}{b^2}XY\right)} \Bigg]
$$
$$
+ \mathbb{E}\Bigg[ \frac{\alpha}{2}\frac{\partial}{\partial a}\left( \frac{-\frac{2}{b}X + \frac{2a}{b^2}X^2 + \frac{2a}{b^2}XY}{1 - \frac{2a}{b}X + \frac{a^2}{b^2}X^2 + \frac{a^2}{b^2}XY} \right) e^{\frac{\alpha}{2}\log\left(1-\frac{2a}{b}X+\frac{a^2}{b^2}X^2+\frac{a^2}{b^2}XY\right)} \Bigg].
$$

At $a = a_*$, $\alpha = 0$ and $\rho = \frac{1}{2}\mathbb{E}\left[\log\left(1 - \frac{2a_*}{b}X + \frac{a_*^2}{b^2}X^2 + \frac{a_*^2}{b^2}XY\right)\right] = 0$, which implies that

$$
0 = \frac{1}{4}(\alpha'(a_*))^2\mathbb{E}\left[ \left(\log\left(1 - \frac{2a_*}{b}X + \frac{a_*^2}{b^2}X^2 + \frac{a_*^2}{b^2}XY\right)\right)^2 \right]
$$
$$
+ \alpha'(a_*)\mathbb{E}\left[ \frac{-\frac{2}{b}X + \frac{2a_*}{b^2}X^2 + \frac{2a_*}{b^2}XY}{1 - \frac{2a_*}{b}X + \frac{a_*^2}{b^2}X^2 + \frac{a_*^2}{b^2}XY} \right],
$$

which implies that

$$
\alpha'(a_*) = \frac{-4\mathbb{E}\left[ \frac{-\frac{2}{b}X + \frac{2a_*}{b^2}X^2 + \frac{2a_*}{b^2}XY}{1 - \frac{2a_*}{b}X + \frac{a_*^2}{b^2}X^2 + \frac{a_*^2}{b^2}XY} \right]}{\mathbb{E}\left[ \left(\log\left(1 - \frac{2a_*}{b}X + \frac{a_*^2}{b^2}X^2 + \frac{a_*^2}{b^2}XY\right)\right)^2 \right]}, \tag{E.34}
$$

and therefore

$$
\alpha \sim \frac{4\mathbb{E}\left[ \frac{-\frac{2}{b}X + \frac{2a_*}{b^2}X^2 + \frac{2a_*}{b^2}XY}{1 - \frac{2a_*}{b}X + \frac{a_*^2}{b^2}X^2 + \frac{a_*^2}{b^2}XY} \right]}{\mathbb{E}\left[ \left(\log\left(1 - \frac{2a_*}{b}X + \frac{a_*^2}{b^2}X^2 + \frac{a_*^2}{b^2}XY\right)\right)^2 \right]} (a_* - a) \tag{E.35}
$$

as $a \uparrow a_*$. $\qquad\square$

When the dimension $d$ is large, we can use Proposition 31 and Proposition 32 to obtain a more explicit approximation for the tail-index $\alpha$ when it is between 0 and 2.

**Theorem 33.** *When the dimension $d$ is large, the tail-index satisfies:*

$$\alpha \sim 2 - \frac{d^3}{4b(b+2)}\left(\eta\sigma^2 - \frac{2b}{d+b+1}\right), \tag{E.36}$$

*for any $\frac{2b}{d+b+1} \leq \eta\sigma^2 < \frac{2b}{d+b+1} + \frac{8b(b+2)}{d^3}$.*

Note that in Theorem 33 the approximation $2 - \frac{d^3}{4b(b+2)}\left(\eta\sigma^2 - \frac{2b}{d+b+1}\right)$ decreases from $2$ to $0$ as $\eta\sigma^2$ increases from $\frac{2b}{d+b+1}$ to $\frac{2b}{d+b+1} + \frac{8b(b+2)}{d^3}$, which is an approximation of $a_*$. Moreover, the approximation $2 - \frac{d^3}{4b(b+2)}\left(\eta\sigma^2 - \frac{2b}{d+b+1}\right)$ is strictly decreasing in $d$, $\eta$, and $\sigma^2$ and strictly increasing in $b$. This is consistent with the monotonicity results we have shown before.

*Proof of Theorem 33.* When $\eta\sigma^2$ increases from $a_c = \frac{2b}{d+b+1}$ to $a_*$, where $a_* = \inf\{a > 0 : \rho(a) = 0\}$, the tail-index decreases from $2$ to $0$. When $d \uparrow \infty$, $a_* \to 0$ and $a_c \to 0$, and hence it suffices to show that

$$\alpha \sim 2 - \frac{d^3}{4b(b+2)}\left(\eta\sigma^2 - \frac{2b}{d+b+1}\right), \tag{E.37}$$

as $d \uparrow \infty$ and $\eta\sigma^2 \downarrow \frac{2b}{d+b+1}$, and $a_* \sim \frac{2b}{d+b+1} + \frac{8b(b+2)}{d^3}$, and when $d \uparrow \infty$ and $\eta\sigma^2 \uparrow \frac{2b}{d+b+1} + \frac{8b(b+2)}{d^3}$,

$$\alpha \sim 2 - \frac{d^3}{4b(b+2)}\left(\eta\sigma^2 - \frac{2b}{d+b+1}\right). \tag{E.38}$$

We will prove (E.37) in Corollary 34 which is a corollary of Proposition 31 when $d \uparrow \infty$ and we will prove (E.38) in Corollary 35 which is a corollary of Proposition 32 when $d \uparrow \infty$. $\qquad\square$

When the dimension $d$ is large, we have the following result as a corollary of Proposition 31.

**Corollary 34.** *The tail-index satisfies:*

$$\alpha \sim 2 - \frac{d^3}{4b(b+2)}\left(\eta\sigma^2 - \frac{2b}{d+b+1}\right), \tag{E.39}$$

*as $d \uparrow \infty$ and $\eta\sigma^2 \downarrow \frac{2b}{d+b+1}$.*

*Proof of Corollary 34.* As $d \uparrow \infty$, $a_c \downarrow 0$, we have $\frac{2a_c}{b}X + \frac{a_c^2}{b^2}X^2 + \frac{a_c^2}{b^2}XY \to 0$ in probability, and using $\log(1+x) \sim x - \frac{x^2}{2}$ so that $(1+x)\log(1+x) \sim (1+x)x - \frac{x^2}{2}$, we have

$$
\begin{aligned}
F_c &\sim \mathbb{E}\left[\left(1 - \frac{2a_c}{b}X + \frac{a_c^2}{b^2}X^2 + \frac{a_c^2}{b^2}XY\right)\left(-\frac{2a_c}{b}X + \frac{a_c^2}{b^2}X^2 + \frac{a_c^2}{b^2}XY\right)\right] \\
&\quad - \frac{1}{2}\mathbb{E}\left[\left(-\frac{2a_c}{b}X + \frac{a_c^2}{b^2}X^2 + \frac{a_c^2}{b^2}XY\right)^2\right] \\
&= \frac{1}{2}\mathbb{E}\left[\left(-\frac{2a_c}{b}X + \frac{a_c^2}{b^2}X^2 + \frac{a_c^2}{b^2}XY\right)^2\right].
\end{aligned}
$$

Next, we can compute that

$$
\mathbb{E}\left[\left(-\frac{2a_c}{b}X + \frac{a_c^2}{b^2}X^2 + \frac{a_c^2}{b^2}XY\right)^2\right]
$$

$$
= \frac{4a_c^2}{b^2}\mathbb{E}[X^2] + \frac{a_c^4}{b^4}\mathbb{E}[X^4] + \frac{a_c^4}{b^4}\mathbb{E}[X^2Y^2] - \frac{4a_c^3}{b^3}\mathbb{E}[X^3] + \frac{2a_c^4}{b^4}\mathbb{E}[X^3Y] - \frac{4a_c^3}{b^3}\mathbb{E}[X^2Y]
$$

$$
= \frac{4a_c^2}{b^2}b(b+2) + \frac{a_c^4}{b^4}b(b+2)(b+4)(b+6) + \frac{a_c^4}{b^4}b(b+2)(d^2-1)
$$

$$
\quad - \frac{4a_c^3}{b^3}b(b+2)(b+4) + \frac{2a_c^4}{b^4}b(b+2)(b+4)(d-1) - \frac{4a_c^3}{b^3}b(b+2)(d-1)
$$

$$
= \frac{16b(b+2)}{(d+b+1)^2} + \frac{16b(b+2)(b+4)(b+6)}{(d+b+1)^4} + \frac{16b(b+2)(d^2-1)}{(d+b+1)^4}
$$

$$
\quad - \frac{32b(b+2)(b+4)}{(d+b+1)^3} + \frac{32b(b+2)(b+4)(d-1)}{(d+b+1)^4} - \frac{32b(b+2)(d-1)}{(d+b+1)^3}
$$

$$
= \frac{16b(b+2)}{(d+b+1)^4}\left[(d+b+1)^2 + (b+4)(b+6) + d^2 - 1\right.
$$

$$
\quad \left. - 2(b+4)(d+b+1) + 2(b+4)(d-1) - 2(d+b+1)(d-1)\right]
$$

$$
\sim \frac{32b(b+2)}{d^3},
$$

as $d \uparrow \infty$, where we used the formulas for the moments of chi-square random variables, and $a_c \sim \frac{2b}{d}$ for $d \uparrow \infty$. Therefore, it follows from Proposition 31 that we have

$$
\alpha \sim 2 - \frac{d^3}{4b(b+2)}\left(\eta\sigma^2 - \frac{2b}{d+b+1}\right), \tag{E.40}
$$

as $d \uparrow \infty$ and $\eta\sigma^2 \downarrow \frac{2b}{d+b+1}$. The proof is complete. $\qquad\square$

When the dimension $d$ is large, we have the following result as a corollary of Proposition 32.

**Corollary 35.** *When $d \uparrow \infty$ and $\eta\sigma^2 \uparrow a_* \sim \frac{2b}{d+b+1} + \frac{8b(b+2)}{d^3}$, the tail-index satisfies*

$$
\alpha \sim 2 - \frac{d^3}{4b(b+2)}\left(\eta\sigma^2 - \frac{2b}{d+b+1}\right). \tag{E.41}
$$

*Proof of Corollary 35.* Note that by the definition of $a_*$,

$$
0 = \rho(a_*) = \frac{1}{2}\mathbb{E}\left[\log\left(1 - \frac{2a_*}{b}X + \frac{a_*^2}{b^2}X^2 + \frac{a_*^2}{b^2}XY\right)\right]. \tag{E.42}
$$

When the dimension $d$ is large, i.e. $d \uparrow \infty$, we have $Y \uparrow \infty$ in probability, and thus $a_* \to 0$. This implies that $-\frac{2a_*}{b}X + \frac{a_*^2}{b^2}X^2 \to 0$ in probability, and hence we must have $\frac{a_*^2}{b^2}XY \to 0$ as well. It follows that

$$
0 \sim \frac{1}{2}\mathbb{E}\left[-\frac{2a_*}{b}X + \frac{a_*^2}{b^2}X^2 + \frac{a_*^2}{b^2}XY\right], \tag{E.43}
$$

which implies that as $d \uparrow \infty$,

$$
a_* \sim a_c := \frac{2b}{d+b+1}, \tag{E.44}
$$

where we used $\mathbb{E}\left[-\frac{2a_c}{b}X + \frac{a_c^2}{b^2}X^2 + \frac{a_c^2}{b^2}XY\right] = 0$. Note that when $a = a_c$, $\alpha = 2$, which is not close to zero, and to get a finer approximation, using $\log(1 + x) \sim x - \frac{x^2}{2}$, we get

$$0 \sim \frac{1}{2}\mathbb{E}\left[-\frac{2a_*}{b}X + \frac{a_*^2}{b^2}X^2 + \frac{a_*^2}{b^2}XY\right] - \frac{1}{4}\mathbb{E}\left[\left(-\frac{2a_*}{b}X + \frac{a_*^2}{b^2}X^2 + \frac{a_*^2}{b^2}XY\right)^2\right]$$

$$\sim \frac{1}{2}\mathbb{E}\left[-\frac{2a_*}{b}X + \frac{a_*^2}{b^2}X^2 + \frac{a_*^2}{b^2}XY\right] - \frac{1}{4}\mathbb{E}\left[\left(-\frac{2a_c}{b}X + \frac{a_c^2}{b^2}X^2 + \frac{a_c^2}{b^2}XY\right)^2\right]$$

$$\sim \frac{1}{2}\mathbb{E}\left[-\frac{2a_*}{b}X + \frac{a_*^2}{b^2}X^2 + \frac{a_*^2}{b^2}XY\right] - \frac{8b(b+2)}{d^3},$$

where we used from the proof of Corollary 34 that

$$\mathbb{E}\left[\left(-\frac{2a_c}{b}X + \frac{a_c^2}{b^2}X^2 + \frac{a_c^2}{b^2}XY\right)^2\right] \sim \frac{32b(b+2)}{d^3},$$

as $d \uparrow \infty$. Let us write $a_* = a_c + \epsilon$, then as $\epsilon \to 0$, we have

$$0 \sim \frac{1}{2}\mathbb{E}\left[-\frac{2a_c}{b}X + \frac{a_c^2}{b^2}X^2 + \frac{a_c^2}{b^2}XY\right] + \frac{1}{2}\mathbb{E}\left[-\frac{2}{b}X + \frac{2a_c}{b^2}X^2 + \frac{2a_c}{b^2}XY\right]\epsilon - \frac{8b(b+2)}{d^3}, \quad \text{(E.45)}$$

and we can compute that

$$\mathbb{E}\left[-\frac{2}{b}X + \frac{2a_c}{b^2}X^2 + \frac{2a_c}{b^2}XY\right] = \left[\frac{-2}{b}b + \frac{2a_c}{b^2}b(b+2) + \frac{2a_c}{b^2}b(d-1)\right] = 2, \quad \text{(E.46)}$$

which implies that

$$\epsilon \sim \frac{8b(b+2)}{d^3}. \quad \text{(E.47)}$$

Hence, we have

$$\alpha \sim \frac{4\mathbb{E}\left[-\frac{2}{b}X + \frac{2a_*}{b^2}X^2 + \frac{2a_*}{b^2}XY\right]}{\mathbb{E}\left[\left(-\frac{2a_*}{b}X + \frac{a_*^2}{b^2}X^2 + \frac{a_*^2}{b^2}XY\right)^2\right]}\left(a_* - \eta\sigma^2\right)$$

$$\sim \frac{4\mathbb{E}\left[-\frac{2}{b}X + \frac{2a_c}{b^2}X^2 + \frac{2a_c}{b^2}XY\right]}{\mathbb{E}\left[\left(-\frac{2a_c}{b}X + \frac{a_c^2}{b^2}X^2 + \frac{a_c^2}{b^2}XY\right)^2\right]}\left(a_c + \epsilon - \eta\sigma^2\right).$$

We recall that

$$4\mathbb{E}\left[-\frac{2}{b}X + \frac{2a_c}{b^2}X^2 + \frac{2a_c}{b^2}XY\right] = 8, \quad \text{(E.48)}$$

and from the proof of Corollary 34, we have

$$\mathbb{E}\left[\left(-\frac{2a_c}{b}X + \frac{a_c^2}{b^2}X^2 + \frac{a_c^2}{b^2}XY\right)^2\right] \sim \frac{32b(b+2)}{d^3},$$

as $d \uparrow \infty$, where we used the formulas for the moments of chi-square random variables, and $a_c \sim \frac{2b}{d}$ for $d \uparrow \infty$. Hence, we conclude that

$$\alpha \sim \frac{8d^3}{32b(b+2)}\left(\frac{2b}{d+b+1} + \frac{8b(b+2)}{d^3} - \eta\sigma^2\right), \quad \text{(E.49)}$$

when $d \uparrow \infty$ is large and $\eta\sigma^2 \uparrow \frac{2b}{d+b+1} + \frac{8b(b+2)}{d^3}$. The proof is complete by noticing that

$$\frac{8d^3}{32b(b+2)}\left(\frac{2b}{d+b+1} + \frac{8b(b+2)}{d^3} - \eta\sigma^2\right) = 2 - \frac{d^3}{4b(b+2)}\left(\eta\sigma^2 - \frac{2b}{d+b+1}\right). \quad \text{(E.50)}$$

$\square$

**Remark 36.** *We have already obtained an approximation of $\alpha$ when $\alpha$ lies between $0$ and $2$ and the dimension $d$ is large (see Theorem 33). Fix the dimension $d$ and batch size $b$, when $a = \eta\sigma^2 \to 0$, the tail-index $\alpha \to \infty$. Let us derive an approximation for the tail-index $\alpha$ in this asymptotic regime. We recall that the tail-index $\alpha$ is uniquely determined by*

$$1 = h(\alpha) = \mathbb{E}\left[\left(1 - \frac{2a}{b}X + \frac{a^2}{b^2}X^2 + \frac{a^2}{b^2}XY\right)^{\alpha/2}\right], \tag{E.51}$$

*where $a = \eta\sigma^2$ and $X, Y$ are independent and $X$ is chi-square random variable with degree of freedom $b$ and $Y$ is a chi-square random variable with degree of freedom $(d-1)$. We apply the approximation $(1+x)^{\alpha/2} \sim 1 + \frac{\alpha}{2}x + \frac{\frac{\alpha}{2}(\frac{\alpha}{2}-1)}{2}x^2$ to get:*

$$1 \sim 1 + \frac{\alpha}{2}\mathbb{E}\left[\left(-\frac{2\eta\sigma^2}{b}X + \frac{\eta^2\sigma^4}{b^2}X^2 + \frac{\eta^2\sigma^4}{b^2}XY\right)\right]$$
$$+ \frac{\frac{\alpha}{2}(\frac{\alpha}{2}-1)}{2}\mathbb{E}\left[\left(-\frac{2\eta\sigma^2}{b}X + \frac{\eta^2\sigma^4}{b^2}X^2 + \frac{\eta^2\sigma^4}{b^2}XY\right)^2\right].$$

*Assume that $\eta\sigma^2$ is small and ignore the higher-order terms, we get*

$$\frac{\alpha}{2} \sim 1 + \frac{2\mathbb{E}\left[\frac{2\eta\sigma^2}{b}X\right] - 2\mathbb{E}\left[\frac{\eta^2\sigma^4}{b^2}X^2 + \frac{\eta^2\sigma^4}{b^2}XY\right]}{\mathbb{E}\left[\left(\frac{2\eta\sigma^2}{b}X\right)^2\right]}$$

$$= 1 + \frac{\frac{4\eta\sigma^2}{b}b - 2\frac{\eta^2\sigma^4}{b^2}(b^2+2b) - 2\frac{\eta^2\sigma^4}{b^2}b(d-1)}{\frac{4\eta^2\sigma^4}{b^2}(b^2+2b)} = \frac{b}{\eta\sigma^2(b+2)} + \frac{1}{2} - \frac{d-1}{2(b+2)}.$$

*Hence, we conclude that as $\eta\sigma^2 \to 0$, the tail-index satisfies*

$$\alpha \sim \frac{2b}{\eta\sigma^2(b+2)} + 1 - \frac{d-1}{b+2}. \tag{E.52}$$

*Note that the approximation $\frac{2b}{\eta\sigma^2(b+2)} + 1 - \frac{d-1}{b+2}$ is strictly increasing in $b$, and strictly decreasing in $\eta$, $\sigma^2$ and $d$, which is consistent with what we have shown before. If $\eta\sigma^2 = \frac{2b}{d+b+1}$, we know from Proposition 3 that the tail-index $\alpha = 2$. Indeed, by plugging $\eta\sigma^2 = \frac{2b}{d+b+1}$ into the right hand side of (E.52), we get*

$$\frac{2b}{\eta\sigma^2(b+2)} + 1 - \frac{d-1}{b+2} = \frac{2b(d+b+1)}{2b(b+2)} + 1 - \frac{d-1}{b+2} = 2, \tag{E.53}$$

*which is consistent with Proposition 3.*

**Remark 37.** *The approximation in (E.52) is good if $\eta\sigma^2$ is small, and every other model parameter is fixed. When $\eta\sigma^2$ is small, and dimension $d$ is large, a finer approximation is given by*

$$\frac{\alpha}{2} \sim 1 + \frac{2\mathbb{E}\left[\frac{2\eta\sigma^2}{b}X\right] - 2\mathbb{E}\left[\frac{\eta^2\sigma^4}{b^2}X^2 + \frac{\eta^2\sigma^4}{b^2}XY\right]}{\mathbb{E}\left[\left(\frac{-2\eta\sigma^2}{b}X + \frac{\eta^2\sigma^4}{b^2}X^2 + \frac{\eta\sigma^4}{b^2}XY\right)^2\right]}, \tag{E.54}$$

*and we can compute that*

$$\mathbb{E}\left[\left(\frac{-2\eta\sigma^2}{b}X + \frac{\eta^2\sigma^4}{b^2}X^2 + \frac{\eta\sigma^4}{b^2}XY\right)^2\right]$$

$$= \frac{4a^2}{b^2}b(b+2) + \frac{a^4}{b^4}b(b+2)(b+4)(b+6) + \frac{a^4}{b^4}b(b+2)(d^2-1)$$

$$- \frac{4a^3}{b^3}b(b+2)(b+4) + \frac{2a^4}{b^4}b(b+2)(b+4)(d-1) - \frac{4a^3}{b^3}b(b+2)(d-1)$$

$$\sim \frac{4a^2}{b^2}b(b+2) + \frac{a^4}{b^4}b(b+2)d^2 - \frac{4a^3}{b^3}b(b+2)d, \tag{E.55}$$

*where $a = \eta\sigma^2$. Hence, we obtain the approximation:*

$$\frac{\alpha}{2} \sim 1 + \frac{\frac{4\eta\sigma^2}{b}b - 2\frac{\eta^2\sigma^4}{b^2}(b^2 + 2b) - 2\frac{\eta^2\sigma^4}{b^2}b(d-1)}{\frac{4\eta^2\sigma^4}{b^2}b(b+2) + \frac{\eta^4\sigma^8}{b^4}b(b+2)d^2 - \frac{4\eta^3\sigma^6}{b^3}b(b+2)d}, \tag{E.56}$$

*which yields that the tail-index $\alpha$ can be approximated as:*

$$\alpha \sim 2 + \frac{2b}{\eta\sigma^2(b+2)(1 + \frac{\eta^2\sigma^4}{4b^2}d^2 - \frac{\eta\sigma^2}{b}d)} - \frac{4(d+b+1)}{(b+2)(4 + \frac{\eta^2\sigma^4}{b^2}d^2 - \frac{4\eta\sigma^2}{b}d)}. \tag{E.57}$$

