# OpenReview forum: "The Heavy-Tail Phenomenon in SGD"
_ICLR.cc/2021/Conference — Reject_

### Official Review · AnonReviewer1 · 2020-10-25

**Rating:** 5
**Confidence:** 4

**Review:**

Summary: This paper studies the heavy-tail phenomenon in SGD. In recent years, many works either empirically or theoretically showed that the heavy-tail property of SGD leads to a better generalization performance. But no rigorous theory has been developed to explain the cause of this phenomenon.

In this paper, the authors studied SGD iterates in solving a quadratic linear regression problem with i.i.d (Gaussian) data. In this setting, they proved the main result that shows that the tails of SGD iterates become monotonically heavier for increasing curvature, decreasing stepwise, or increasing b, hence this tends to provide a rigorous explanation of the heavy-tail phenomenon. The authors also developed other results on the law convergence and variance bounds.

Comments: Overall I think the paper is technically sound and develops rigorous proof. My main comment is that the problem studied in the paper is very idealized, I.e., a regression with iid Gaussian data. I understand that this makes the proof cleaner with compact results. However, this may not have direct implications on the heavy-tail phenomenon in deep learning. In fact, it may be more meaningful and motivated to study other simpler problems, such as binary classification with linearly separable data, in which the data structure is more practical than iid Gaussian.

Overall, I consider the results in this paper as rigorously exploring the heavy-tail of SGD in linear regression under iid Gaussian data. I suggest the authors consider other more practical problems, and the results need not be related to deep learning. Focusing on the heavy-tail phenomenon in standard machine learning problems is already an interesting problem.

---

> ### Author Response · Authors · 2020-11-17
> **Authors' Reply to AnonReviewer1**
>
> We thank the reviewer for their time and suggestions.
>
> We have addressed the reviewer’s comments in the general response, under the following items:
>
> 1) **Idealized Setting**
> 2) **Non-Gaussian Data**
> 3) **More General Loss Functions**
>
> In addition to the above-mentioned responses, we would like to add that, we have provided various experiments that suggest that the heavy-tail phenomenon emerges in deep learning as well (see also our **new experiments on VGG networks**). Hence, we believe that such strong empirical evidence supports our claim that our theory **does have** important implications on the heavy-tail phenomenon in deep learning, and opens up various future directions.

---

### Official Review · AnonReviewer3 · 2020-10-28
**Reviews for The Heavy-Tail Phenomenon in SGD**

**Rating:** 6
**Confidence:** 3

**Review:**

This paper studies the relations between the heavy tail phenomenon of SGD and the ‘flatness’ of the local minimum found by SGD and the ratio of the step size $\eta$ to the batch size $b$ for the quadratic and convex problem. They show that depending on the curvature, the step size, and the batch size, the iterates can converge to a heavy-tailed random variable.  They conduct experiments on both synthetic data and fully connected neural networks, and illustrate that the results would also apply to more general settings and hence provide new insights about the behavior of SGD in deep learning.
I do not have time to check the proofs and are not familiar with this topic. However the results seem to be novel and interesting.

pros: 1, They characterize the empirically observed heavy-tailed behavior of SGD with respect to the step size, mini batch size, dimensions, and the curvature, with explicit convergence rates.

cons:1, The theoretical analysis is only for the quadratic and convex function. The input data is Gaussian, which is quite restrictive.

After the rebuttal.

The authors addressed my concerns. I have read other reviewers' comments. I decide to remain the current score.

---

> ### Author Response · Authors · 2020-11-17
> **Authors' Reply to AnonReviewer3**
>
> We thank the reviewer for the positive feedback and finding our results novel and interesting.
>
> We have addressed the reviewer’s comments in the general response, under the following items:
>
> 1) **Idealized Setting**
> 2) **Non-Gaussian Data**
> 3) **More General Loss Functions**

---

### Official Review · AnonReviewer4 · 2020-11-01
**SGB iterations can result in heavy-tail random distribution on certain quadratic optimisation problems**

**Rating:** 5
**Confidence:** 3

**Review:**

The main theme of this work is to study conditions under which SGD iterations result in random variables with heavy-tail random distributions. Specifically they focus on the step size, batch size and problem dimension. First they show theoretical results showing how the tail-index of the distribution generated by SGD depends on the chosen step size, batch size and problem dimension.
These results make several assumptions limiting their scope of applicability. Namely, they are limited to quadratic optimisation problems, normally distributed data and one-pass regime of SGD. However, the authors argue that since even in this simplest setting SGD results in a heavy-tail distribution, the more complex problems should not be expected to behave otherwise.
The experiments verify the correctness of the theoretical results, but they are also of limited value. For example, the deep learning section studies a shallow fully connected network on MNIST and CIFAR10. It would be more interesting to see a model performing closer to the state of the art here.
While the main results of this work are interesting, I have a few reservations. First, as I wrote above the setting (even in experiments) is limited to the simplest case and hence is of limited value. Second, it is quite difficult to follow the theoretical results, mostly because all details are given in the Appendix, which is 30 pages long. Perhaps, this work is better suited for a journal publication, where the authors will have more space to explain their work. And finally I have some rather minor comments:
- the paper refers to equation (3) in several places of the article and the Appendix, but there is no equation (3). I had to guess to which equation they refer each time, which made reading the already technically heavy paper more difficult.
- In Theorem 4, it is not clear how "the p-th moment of the iterates are of exponentially decaying nature", since in both cases the coefficient of $\mathbb{E}\Vert q_1\Vert^p$ is larger than 1.  Also, in the second case, the coefficient's value depends on whether $\alpha<2$. What is $q_1$ here?

---

> ### Author Response · Authors · 2020-11-17
> **Authors' Reply to AnonReviewer4**
>
>
> We thank the reviewer for their time and efforts put in their review.
>
> We have addressed the reviewer’s major comments in the general response, under the following items:
>
> 1) **Idealized Setting**
> 2) **Non-Gaussian Data**
> 3) **More General Loss Functions**
> 4) **Extending Beyond Streaming Data Setting**
>
> Here are our responses to the remaining comments.
>
> **Additional Experiments**
>
> In our experiments, we gradually increased the complexity of the experiments and showed that our theory is applicable on multiple datasets with different sizes of neural networks. More precisely:
> 1. we first validated our theory in its exact setup (linear regression)
> 2. and then illustrated that the conclusions of our theory are still applicable in neural networks.
>
> In this sense, we disagree with the reviewer on the comment that our experiments “are of limited value” (also note that our experiments are perceived as “comprehensive” by Reviewer 2).
>
> Nevertheless, we have now run more experiments on a much more complicated network architecture, namely VGG11 (10 convolutional layers interlaced with max-pooling and ReLU, and a final linear layer) on the CIFAR10 dataset. This network contains *10 million* parameters and it is close to the state-of-the-art architectures in terms of complexity. We have used the implementation provided in https://github.com/chengyangfu/pytorch-vgg-cifar10 and combined it with our code, which was submitted in the supplementary material.
>
> We report our results in the updated version of the paper. Our results show that the heavy-tails clearly emerge in the middle layers of the network and the results bring up new theoretical questions on the investigation of the layer-wise tail behavior.
>
> In the next revision, we will replicate the same experiments with different initial conditions (as we did with the other experiments) and we will further evaluate our theory on VGG networks with 13layers, 16layers, and 19 layers to provide further support for our theory. We are confident that the new results will also support our claims.
>
> **Length and the Content of the Supplementary Document**
>
> We shall underline that all the main results of the paper are given in the main body of the paper. The proof of the main results are all given in Appendix C **(6 pages)** and the supporting lemmas and their proofs are given in Appendix D **(6 pages)**. That is 12 pages in total.
>
> The remainder of the Appendix is dedicated to additional technical results only for those enthusiastic readers who might be interested to know what other results are possible in our theoretical framework. We also noticed that some theory papers at ICLR, ICML, and NeurIPS in the past are quite long, and a 30-page Appendix is not that uncommon.
>
> **Minor Corrections**
>
> We thank the reviewer for noticing the typo. Due to a Latex problem, Equation (3.3) was printed as Equation (3). This issue has been fixed now.
>
> In Theorem 4, what we aimed to explain was that when $p<\alpha$ the upper bound on the p-th moment of the iterates converges exponentially to the p-th moment of $q_1$ when $\alpha\leq 1$ and a neighborhood of the p-th moment of $q_1$ when $\alpha>1$. Here, $q_1$ is defined in equation (3.3). We have now re-phrased this.

---

### Official Review · AnonReviewer2 · 2020-11-09
**An Insightful Paper to be accpeted**

**Rating:** 7
**Confidence:** 4

**Review:**

This paper gives a theoretical study of the tail behavior of the SGD in a quadratic optimization problem and explores its relationship with the curvature, step size and batch size. To prove their results, the authors approximate the SGD recursion by a linear stochastic recursion and analyze the statistical properties by the tools from implicit renew theory. Under this setting, they show that the law of the SGD iterates converge to a heavy-tailed stationary distribution depending on the Hessian structure of the loss function at the minimum and choices of the step size and batch size. They take a further step to clarify the relationship and study the moment bounds and convergence rate.

Overall, I vote for accepting. I think the study of heavy tail phenomenon in SGD is quite a novel field and full of interest.
1.	This paper is the first one to study the originating cause of heavy tail phenomenon in SGD and give a rigorous proof of the relationship between tail index and the choice of step size and batch size. This helps a better understanding of the generalization properties of SGD.
2.	The paper provides comprehensive experiments to support their result. Experiments are conducted on both synthetic data and neural networks. The design of experiments is reasonable, and the results of the experiment not only support the claim that the tail index is deeply linked to the curvature of the loss and the ratio of step size and batch size but also give an insight on more general cases besides the quadratic optimization.

However, I am still concerned about the setting in the theoretical frame. This paper completes its proof under the settings of quadratic optimization and infinite streaming data, which may limit the applicability of the theoretical result. Although these issues have been discussed in this paper, whether or why this extension is feasible remains questionable.

---

> ### Author Response · Authors · 2020-11-17
> **Authors' Reply to AnonReviewer2**
>
> We thank the reviewer for finding our paper an insightful paper to be accepted. We also thank the reviewer for very helpful feedback.
>
> We have addressed the reviewer’s comments in the general response, under the following items:
>
> 1) **More General Loss Functions**
> 2) **Extending Beyond Streaming Data Setting**

---

### Author Response · Authors · 2020-11-17
**General Comments - Part I**

We thank all the four reviewers for their valuable time. Here, we address the recurring comments, whereas specific comments will be addressed as individual responses to reviewers’ comments.

**Idealized Setting (Reviewers 1, 3, 4)**

We agree that developing a general theory that applies to a generic deep learning setting would be the ultimate goal and we acknowledge that the problem studied in this paper is an idealized version of the typical learning setup in deep learning, i.e. linear regression with Gaussian streaming data.

However, we must emphasize that **our work is the first step towards identifying such heavy-tailed behavior in machine learning** and our paper is **not built up on any machine learning study**. In this sense, (similar to most theoretical advances) achieving the ultimate task would require first developing the theory in more idealized settings, such as the one considered in our paper.

As our analysis illustrates, identifying the heavy-tails in SGD is already very challenging in our idealized scenario. Furthermore, we present strong empirical evidence that our theory extends to more complex settings **(see the updated paper for more experiments)**. Hence, we believe that our contributions will serve a crucial role for future developments in this emerging field, in terms of being the basis of more detailed theoretical analysis. Without this first step, developing a more advanced theory would be a much more challenging task.

Below, we will explain in detail, how the scope of our analysis can be extended to more general settings. We hope that Reviewers 1, 3, and 4 could re-evaluate our contributions in the light of these explanations.

**Non-Gaussian Data (Reviewers 1, 3, 4)**

Let us highlight and reiterate that the emergence of heavy tails in optimization is an **irregularity result**. As opposed to the classical optimization literature, where one is interested in convergence to a minimum, in our setup, assuming that the data is Gaussian **does not** make our task easier: on the contrary, the challenge is to prove that such an irregularity happens **even when** all the constituents of the problem are “regular”.

As we mentioned in the paper (Footnote in Page 4), choosing a much more complicated, heavy-tailed data model immediately results in heavy-tailed iterates, which is far less interesting.

On the other hand, having Gaussian data allows us to have explicit computations such as the explicit characterization of the stepsize threshold $\eta_{\text{crit}}$ in Proposition 3. However, while not explicitly discussed in the paper, **our results also generalize to the case when the data is not Gaussian but coming from another distribution with finite moments.**

In this case, we still have heavy tails but characterizing the tail index **exactly** is an NP-hard problem as it requires computing the Lyapunov exponent of the random recursion of SGD. That being said, our estimate “alpha” of the tail index in this paper, i.e. the unique solution to the non-linear equation $h(s) = 1$ will be a **lower bound** on the tail index, and **it will still satisfy monotonicity properties**. In other words, all our results will hold except that they will hold for a tractable approximation of the actual tail index. We updated our paper adding a remark about this generalization and we highlighted our changes in the text with a blue marker in the revised version. Note that this particular data model (which only requires finite moments) is very generic and covers the **vast majority** of the data models that have been investigated in the literature.

---

> ### Author Response · Authors · 2020-11-17
> **General Comments - Part II**
>
> **More General Loss Functions (Reviewers 1, 2, 3, 4)**
>
> Extending our results beyond quadratic optimization is possible in a straightforward manner, **if the gradients have asymptotic linear growth**. For example, consider the cost
>
> $f(x) =  E [ L( a^T x - y) ]$
>
> with loss function $L$. The choice of $L(z) = \|z\|^2 / 2$ is the standard linear regression setting where the gradient of $f$ is an affine function of $x$ and in this case $\|\| \nabla f(x) - Mx\|\|$ is bounded where $M = \mathbb{E} [aa^T]$. Our results will generalize as long as $\|\| \nabla f(x) - Mx\|\|$ stays bounded even if the function $L$ is not a quadratic function. The proof would use the proof technique of Theorem 1.4 of the paper Mirek (2020) titled “Heavy Tail Phenomenon and Convergence to Stable Laws for Iterated Lipschitz Maps”.
>
> This type of optimization problems arise for instance in robust regression where the objective is
>
> $f(x) = \mathbb{E} (a^T x - b)^2 + \lambda g(x)$
>
> with a penalty function $g(x)$ whose gradient is bounded and a tunable parameter $\lambda$. The boundedness of the gradient of $g(x)$ results in at-most linear growth of $g(x)$ and allows robustness to outliers where the parameter $\lambda$ can be used to adjust the robustness level desired. Examples for the choice of $g(x)$ include the *“The smoothly clipped absolute deviation”* (SCAD) penalty (see e.g. [Loh and Wainwright, JMLR 2015]), Huber function or the exponential squared loss when $g(x) =  1 − exp(−\|x\|^2 / c)$ where $c$ is a tuning parameter.
>
> We updated the paper accordingly and added a paragraph explaining such extensions. Our changes in the revised version are highlighted in the text with a blue marker.
>
> **Extending Beyond Streaming Data Setting (Reviewers 2, 4)**
>
> We have been already working on handling the offline (non-streaming) data setting where the data is sampled over multiple passes. Though that work is preliminary and unpublished, let us explain our current findings, which we hope to be convincing about why such an extension is indeed possible.
>
> In this setting, given a dataset $(a_i, y_i)$ for $i=1,2,...,n$; if we consider linear regression with SGD; we will still have a similar stochastic recursion:
>
> $x_k = M_k x_{k-1} + q_k$
>
> where $M_k$ are i.i.d. The difference is that the entries of $M_k$ will come from a finite set. Since the dataset is fixed, the noise we can introduce in the $M_k$ matrix will have a structure and the support of the noise will not be the whole Euclidean plane. For instance, in the toy example $a_1 = +1$, $a_2 =  -1$ and $y_1 = y_2 = 0$, one can show that there are choices of stepsizes which will lead the SGD iterates to stay on the set of integers in which case the stationary distribution will be degenerate without a continuous density. Then, characterizing the tail index becomes a much harder problem.
>
> Nevertheless, we can still view the data $(a_i, y_i)$ uniformly sampled from a fixed finite set. Consider batch size equal to 1 for simplicity and let $\eta$ be the stepsize. By the stochastic recursion theory, if this finite set produces data points that can lead to a “balanced” mixture of “expansion” and “contraction”, in the sense that $\rho=(1/n)\sum_{i=1}^n \log \| I- \eta a_ia_i^T\|<0$, i.e. the top Lyapunov exponent computed over the uniform distribution over the fixed finite set for the data is less than 0, then **we can still expect heavy-tail phenomenon**.
>
> This will be a future work to investigate and given the current density of our paper, it is not possible to include those extensions in a single submission. To the best of our knowledge, **heavy tails in the non-streaming setting for SGD (or its variance-reduced variants) have never been studied before**. We believe any results in this direction would be interesting and a significant breakthrough.

---

### Decision · Program_Chairs · 2021-01-10
**Final Decision**

**Decision:**

Reject

**Comment:**

This work seeks to describe the heavy-tail phenomenon observed for deep networks learned with SGD. The work presents proof of a relationship between curvature, step size, batch size, and a heavy-tail weight distribution. The proofs assume a quadratic optimization problem and the authors speculate that the results may also be relevant for non-convex deep learning settings. On the positive side the reviewers agreed that this work is one of the first, if not the first, to try to theoretically describe a poorly understood phenomenon in deep learning. On the less positive side, the reviewers believe that the proofs developed in this paper are for an idealized setting that is too different from the settings under which deep models are trained. As such, even though the authors provide some (somewhat mixed) experimental results to support the claim of relevance to deep learning, the reviewers were not convinced. Given that the stated goal of the work is to attempt to explain this phenomenon in deep models, the majority view is that this work, while promising, needs further development to convincingly claim some relevance to the original phenomenon being studied.